# Efficiency Ordering of Stochastic Gradient Descent

**Jie Hu**[1][*]    **Vishwaraj Doshi**[2][*]    **Do Young Eun**[1]

[1] Department of Electrical and Computer Engineering, North Carolina State University
[2] Data Science and Advanced Analytics, IQVIA
jhu29@ncsu.edu    vishwaraj.doshi@iqvia.com    dyeun@ncsu.edu

## Abstract

We consider the stochastic gradient descent (SGD) algorithm driven by a general stochastic sequence, including *i.i.d* noise and random walk on an arbitrary graph, among others; and analyze it in the asymptotic sense. Specifically, we employ the notion of 'efficiency ordering', a well-analyzed tool for comparing the performance of Markov Chain Monte Carlo (MCMC) samplers, for SGD algorithms in the form of Loewner ordering of covariance matrices associated with the scaled iterate errors in the long term. Using this ordering, we show that input sequences that are more efficient for MCMC sampling also lead to smaller covariance of the errors for SGD algorithms in the limit. This also suggests that an arbitrarily weighted MSE of SGD iterates in the limit becomes smaller when driven by more efficient chains. Our finding is of particular interest in applications such as decentralized optimization and swarm learning, where SGD is implemented in a random walk fashion on the underlying communication graph for cost issues and/or data privacy. We demonstrate how certain non-Markovian processes, for which typical mixing-time based non-asymptotic bounds are intractable, can outperform their Markovian counterparts in the sense of efficiency ordering for SGD. We show the utility of our method by applying it to gradient descent with shuffling and mini-batch gradient descent, reaffirming key results from existing literature under a unified framework. Empirically, we also observe efficiency ordering for variants of SGD such as accelerated SGD and Adam, open up the possibility of extending our notion of efficiency ordering to a broader family of stochastic optimization algorithms.

## 1   Introduction

Stochastic gradient descent (SGD) is widely used in machine learning, signal processing and other engineering fields to solve the optimization problem

$$\theta^* = \arg\min_{\theta \in \Theta} \left\{ f(\theta) \triangleq \frac{1}{n} \sum_{i=1}^{n} F(\theta, i) \right\}, \tag{1}$$

where $\Theta \subset \mathbb{R}^d$ is some closed and convex set, and $F(\cdot, i) : \mathbb{R}^d \to \mathbb{R}$ for $i \in [n] \triangleq \{1, \cdots, n\}$ are smooth functions on $\Theta$, not necessarily convex, such that their summation $f : \mathbb{R}^d \to \mathbb{R}$ exhibits a minimizer $\theta^* \in \Theta$ satisfying $\nabla f(\theta^*) = 0$. The update rule of the iterative SGD scheme is of the form

$$\theta_{t+1} = \text{Proj}_{\Theta} \left( \theta_t - \gamma_{t+1} \nabla_\theta F\left( \theta_t, X_{t+1} \right) \right), \tag{2}$$

where $\gamma_t$ is the step size that can be constant or diminishing as $t \to \infty$, $\text{Proj}_{\Theta}$ is a projection operator onto the constraint set $\Theta$, and $\{X_t\}_{t \geq 0}$ is some sequence taking values in $[n]$. This sequence is often generated in a stochastic manner, and samples can be drawn from temporally independent and identically distributed *(i.i.d)* random variables that are either uniformly distributed over $[n]$ [60, 54, 14], or leverage importance sampling techniques for variance reduction [52, 12, 27]. $\{X_t\}_{t \geq 0}$

---

[*]Equal contributors.

36th Conference on Neural Information Processing Systems (NeurIPS 2022).

can also be constructed by repeatedly shuffling over all possible states without repetition,[2] leading to faster convergence than stochastic counterparts drawing *i.i.d* samples from $[n]$ [64, 3, 32, 72].

**Random Walk Stochastic Gradient Descent (RWSGD):** Some applications observe restricted access to the state space, such as decentralized optimization [65, 71, 46], where communication occurs between nodes in a network to collaboratively solve the optimization problem (1). For instance, disease classification in confidential clinical swarm learning [69] considers peer-to-peer networks due to the highly private nature of medical data. In such a setting, the random sequence $\{X_t\}_{t \geq 0}$ is usually realized as a Markov chain on a general graph $\mathcal{G}(\mathcal{V}, \mathcal{E})$ that only samples local gradients of the nodes in $\mathcal{V} \triangleq [n]$ and traverses the network via edges connecting them without divulging the update history or its own gradient. The randomness of the communication path ensures that the compromised node can not easily leak the data of its neighbors [46].

Apart from the privacy concern, such dynamics are also employed in swarm learning/optimization in robotics [17] and wireless sensor networks [43] due to their low communication cost and asynchronous nature. The need for data privacy and demand for communication-efficient algorithms for decentralized optimization has spurred the study of RWSGD algorithms in recent years [57, 34, 67], with the underlying Markov chain in the form of Metropolis-Hasting random walk (MHRW) [48].

**Common analytical approach - Finite time bounds based on mixing time:** Most of the existing works analyzing iteration (2) provide so-called finite-time upper bounds on expected error in either the objective function $\mathbb{E}[f(\tilde{\theta}_t) - f(\theta^*)]$, where $\tilde{\theta}_t$ is some weighted average of the iterates, or its gradient $\mathbb{E}[\|\nabla f(\theta_t)\|_2^2]$; and are used to infer the convergence rate of the iterate sequence [57, 25, 67]. For diminishing step sizes $\gamma_t = t^{-\alpha}$ with $\alpha \in (0.5, 1)$,[3] the upper bound on $\mathbb{E}[\|\nabla f(\theta_t)\|_2^2]$ reads as

$$\mathbb{E}[\|\nabla f(\theta_t)\|_2^2] \leq O\left(\frac{\max\{M, 1/\log(1/\beta)\}}{t^{1-\alpha}}\right), \tag{3}$$

and a similar form for $\mathbb{E}[f(\tilde{\theta}_t) - f(\theta^*)]$ as well [67, 24, 7]. Here, $\beta \in (0, 1)$ is the second largest eigenvalue modulus (SLEM) of the underlying Markov chain's transition matrix and is related to its mixing time property, since smaller SLEM leads to faster mixing of the Markov chain [18, 44]. On the other hand, $M > 0$ is usually a quantity proportional to the local gradients evaluated at the minimizer, or their upper bound. Both the gradient information and the mixing time play a key role in quantifying the convergence rate derived from this upper bound, and the mixing time is especially important since it hints that convergence rate of the SGD algorithm can potentially be accelerated using faster mixing Markov chains for the input driving sequence. It has also been noted that the inherent correlation of the underlying random walk has to be addressed in any analysis concerning Markov-chain-driven gradient descent [67]. The mixing time technique, by capturing the rate at which the chain converges to its stationary distribution [18, 44], is one way of doing so.

**Alternative approach - Asymptotic analysis and efficiency ordering:** In addition to the aforementioned mixing time, another widely used metric for characterizing the second order properties of Markov chains is the asymptotic variance (AV). For any scalar valued function $g : [n] \rightarrow \mathbb{R}$, the estimator $\hat{\mu}_t(g) \triangleq \frac{1}{t} \sum_{i=1}^{t} g(X_i)$, associated with an irreducible Markov chain $\{X_t\}_{t \geq 0}$ with stationary distribution $\boldsymbol{\pi}$, is the average of the samples of $g(\cdot)$ obtained along the chain's sample path up to time $t > 0$. The AV of the Markov chain, denoted by $\sigma_X^2(g)$, is then defined as the the limiting variance of the estimator; that is,

$$\sigma_X^2(g) \triangleq \lim_{t \to \infty} t \cdot \text{Var}(\hat{\mu}_t(g)). \tag{4}$$

For all functions $g(\cdot)$ satisfying $\mathbb{E}_{\boldsymbol{\pi}}(g^2) < \infty$, the AV is associated with the Central Limit Theorem (CLT) for any Markovian kernel on a finite state space, as the variance of the normally distributed estimates in the limit [61, 35, 18]. More formally, we have

$$\sqrt{t} \cdot [\hat{\mu}_t(g) - \mathbb{E}_{\boldsymbol{\pi}}(g)] \xrightarrow[t \to \infty]{dist} \mathcal{N}(0, \sigma_X^2(g)). \tag{5}$$

A smaller AV means that fewer samples are required *post* mixing of the chain[4] in order to obtain a desired accuracy - in some sense quantifying the chain's *efficiency*.

---

[2]One complete pass over the entire set $[n]$ is typically called an epoch. Shuffling can refer to passing over $[n]$ in the same order for every epoch (single shuffling), or in a random order (random shuffling).

[3]We only need the step size to be $O(t^{-\alpha})$, but we omit the $O(\cdot)$ notation for simplicity. We also consider a slightly more general case, allowing for $\alpha = 1$ as well.

[4]Achieved by employing a burn-in period to get rid of the correlation with the initial state [29].

Both the AV and the mixing time of a Markov chain are very strongly related concepts[5]. In fact, the AV has an upper bound in terms of the SLEM, which decreases as the SLEM gets smaller (chain mixes faster) [50]. However, an ordering of the SLEM between two Markov chains does not imply an ordering of their AV, as we shall demonstrate later in Section 4 for a special case. Both of these second-order properties therefore lead to different notions of optimality; and the comparison of two chains based on their AV leads to the concept of *efficiency ordering* [50], where we say that a chain is more efficient than the other if it has a smaller AV, uniformly over all functions $g : [n] \to \mathbb{R}$.

As mentioned earlier, the common intuition asserted by finite time bounds such as (3) is that Markov chains with smaller SLEM lead to faster convergence of the SGD iteration (2) to the minimizer [67, 7]. We put this logic to test by simulating the RWSGD algorithm with three different reversible Markov chains (w.r.t. uniform stationary distribution) as the stochastic inputs - the MHRW, a modification of MHRW, which is also shown in Appendix I [1] to be more *efficient* than MHRW, and the so-called 'fastest mixing Markov chain' (FMMC) as defined in [16] as the Markov chain obtained by minimizing the SLEM over the entire class of reversible chains for a given graph topology. We employ RWSGD to minimize a quadratic objective function for two underlying graphs. The exact details of the setup are deferred to Appendix I [1], and our numerical results in Figure 1 show that even though the FMMC is theoretically guaranteed to have the smallest SLEM ($\beta_i$ for $i \in \{1, 2, 3\}$) of the three reversible chains simulated, it is the worst performing one with largest mean square error (MSE). Although MHRW and Modified-MHRW share the

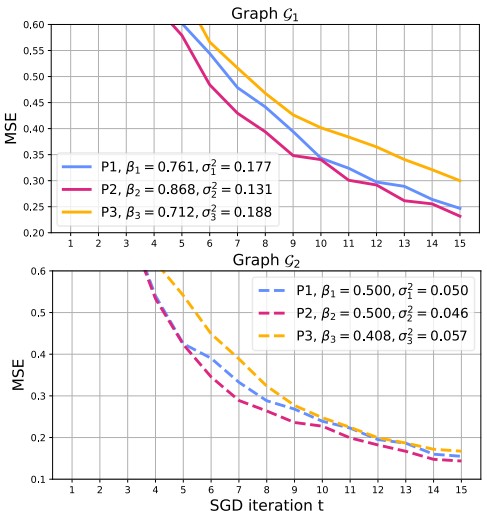

Figure 1: Comparison of MHRW (P1), Modified-MHRW (P2) and FMMC (P3) as stochastic inputs for RWSGD on two different graphs $\mathcal{G}_1$ and $\mathcal{G}_2$.

same SLEM in the lower plot of Figure 1, they still have performance differences. This is contradictory to the intuition derived from (3), and could be attributed to the finite time results providing upper bounds for *all* times $t > 0$, which may therefore not necessarily be tight. On the other hand, the performance of the chains seem to be ordered according to their AV ($\sigma_i^2$ for $i \in \{1, 2, 3\}$) evaluated for a test function. This lends credence to developing techniques based on AV, for judging the performance of different stochastic inputs for SGD, as possible alternatives to using SLEM as the sole performance metric.

The asymptotic variance also appears in the CLT for stochastic approximation (SA) algorithms [11, 21], though this time not directly as the variance in the limit, but as a component of the limiting covariance matrix of the scaled iterate errors. Recent works [21, 51] point out that the covariance matrix itself is of special interest, and typically contains more information than the non-asymptotic MSE bounds [51]. In the sense of SGD algorithms, we will show in Section 3 that it embeds explicit information of the exact vector-valued gradient evaluated at the optimizer as well as the entire spectrum of the transition matrix; as opposed to the upper bound $M$ of the gradients and only the second largest eigenvalue modulus commonly found in mixing time based non-asymptotic bounds. It has been suggested [20, 23], and also proved for the special case of linear SA [21], that the covariance matrix emerging out of the CLT dominates as a leading term of the finite-time MSE bounds. This also holds true for finite-time bounds on weighted MSE for any preferred weight; the weighted MSE being utilized in fields such as wireless MIMO [68] and process optimization [30]. Overall, while finite-time bounds have enjoyed great success in the literature, the potential for performance gains out of the asymptotic analysis of SGD algorithms have remained largely unexplored.

**Contributions:** We employ asymptotic analysis to propose a general framework that offers seamless connection between AV in the MCMC literature with efficiency ordering and covariance matrix in the SGD algorithms. Our framework can be used to design different random walk variants and also to systematically compare the existing sampling methods in the SGD iteration (2) with diminishing step

---

[5]For reversible Markov chains, the AV can be written explicitly as an increasing function of *every* eigenvalue of the transition matrix [18], while the mixing time is related to the SLEM as mentioned earlier.

size, not just limited to random walks. In particular, we show that any two random walks following an efficiency ordering have their covariance matrices Loewner ordered, including *non-Markovian* stochastic processes versus its Markovian counterpart, which defies any mixing-time (SLEM) based analysis. Such ordering can be harnessed into improving the accuracy of SGD iterates, which implies a reduction in the weighted MSE with arbitrary weights. Moreover, via a specific augmentation of the state space, we are able to analyze SGD for both single and random shuffling and show the efficiency of shuffling over *i.i.d* sampling for a set of objective functions that may not satisfy 'Polyak-Łojasiewicz inequality'. We further extend such comparison to mini-batch SGD algorithms. Lastly, we present numerical results where the efficiency ordering via asymptotic analysis tends to hold over all time periods and input sequences with higher efficiency have smaller errors in SGD.

## 2   Modeling setup

**Basic notations:** We use lower case, bold-faced letters to denote vectors ($\mathbf{v} \in \mathbb{R}^d$), and use upper case, bold-faced letters to denote matrices ($\mathbf{M} \in \mathbb{R}^{d \times d}$). $\| \cdot \|_2$ denotes the $l^2$ norm for vectors or 2-norm for matrices. We use $\nabla \mathbf{f}(\cdot)$ as Jacobian matrix of vector-valued function $\mathbf{f}(\cdot)$, and $\nabla^2 g(\cdot)$ as Hessian matrix of scalar-valued function $g(\cdot)$. We let $\nabla_\theta g(\theta, X)$ be the gradient of the scalar-valued function $g(\theta, X)$ with respect to $\theta$ and omit the subscript $\theta$ for simplicity. Loewner ordering of matrices is denote by '$\leq_L$' such that $\mathbf{A} \leq_L \mathbf{B} \iff \mathbf{x}^T(\mathbf{A} - \mathbf{B})\mathbf{x} \leq 0$ for any $\mathbf{x} \in \mathbb{R}^d$. The term $Tr(\mathbf{A})$ denotes the trace of matrix $\mathbf{A}$, and let $\mathbb{1}_{\{\cdot\}}$ be the indicator function. We write $\mathcal{N}(0, \mathbf{V})$ to represent a multivariate Gaussian distribution with zero mean and covariance matrix $\mathbf{V}$. For a connected and undirected graph $\mathcal{G}(\mathcal{V}, \mathcal{E})$ with node set $\mathcal{V}$ and edge set $\mathcal{E}$, we use $N(i)$ for the set of neighbors of node $i \in \mathcal{V}$ and $\mathbf{d} \triangleq [d_1, d_2, \cdots, d_n]^T$ for the degree vector where $d_i = |N(i)|$.

**SGD algorithm with arbitrary input sequence:** We consider random walks $\{X_t\}_{t \geq 0}$ for which the limit $\pi_i \triangleq \lim_{t \to \infty} \frac{1}{t} \sum_{k=1}^t \mathbb{1}_{\{X_k = i\}}$ exists almost surely and is positive for all $i \in [n]$, with $\boldsymbol{\pi} = [\pi_i]_{i \in [n]}$ denoting the limiting or *stationary* distribution. This is trivially satisfied via strong law of large numbers [26] when $X_t$ for each $t > 0$ are *i.i.d* random variables with distribution $\boldsymbol{\pi}$ over $[n]$, and via the ergodic theorem [18] when $\{X_t\}_{t \geq 0}$ is an irreducible, aperiodic and positive recurrent (ergodic) Markov chain. Note however that this way of defining the stationary distribution $\boldsymbol{\pi}$ allows for the input sequence $\{X_t\}_{t \geq 0}$ to be more general, possibly being non-Markov on $[n]$. Then, we can use $\boldsymbol{\pi}$ to rewrite the objective in (1) as

$$f(\theta) = \frac{1}{n} \sum_{i=1}^n F(\theta, i) = \mathbb{E}_{X \sim \boldsymbol{\pi}} \left[ G(\theta, X) \right], \tag{6}$$

where function $G(\theta, i) \triangleq \frac{1}{n\pi_i} F(\theta, i)$ for any $\theta \in \Theta, i \in [n]$. The generalized update rule then becomes

$$\theta_{t+1} = \text{Proj}_\Theta \left( \theta_t - \gamma_{t+1} \nabla G(\theta_t, X_{t+1}) \right). \tag{7}$$

This change of notation allows us to consider input sequences having possibly non-uniform stationary distributions, and is a version of importance sampling for RWSGD schemes, as in [7]. For example, the iteration (7) with the input sequence generated from a MHRW with uniform target distribution $\boldsymbol{\pi} = \mathbf{1}/n$ will reduce down to (2) with $G(\theta, i) = F(\theta, i)$ for all $\theta \in \Theta, i \in [n]$. If the input sequence is instead a simple random walk on a connected graph $\mathcal{G}(\mathcal{V}, \mathcal{E})$ with $\mathcal{V} = [n]$, we have $\boldsymbol{\pi} \propto \mathbf{d}$, and $G(\theta, i) = \frac{\mathbf{1}^T \mathbf{d}}{n d_i} F(\theta, i)$ for all $\theta \in \Theta, i \in \mathcal{V}$.[6]

**Asymptotic covariance matrix.** We now quickly review the multivariate CLT for Markov chains, since it is a natural way to introduce the *asymptotic covariance* matrix, used heavily throughout the paper. For any finite, irreducible Markov chain $\{X_t\}_{t \geq 0}$ with stationary distribution $\boldsymbol{\pi}$, its *estimator* is defined as $\hat{\mu}_t(\mathbf{g}) \triangleq \frac{1}{t} \sum_{k=1}^t \mathbf{g}(X_k)$ for any vector-valued function $\mathbf{g} : [n] \to \mathbb{R}^d$. Then, the ergodic theorem [18, 19] states that for any initial distribution and any $\mathbf{g}(\cdot)$ such that $\mathbb{E}_{\boldsymbol{\pi}}(\mathbf{g}) = \sum_{i \in [n]} \mathbf{g}(i)\pi_i < \infty$, we have $\hat{\mu}_t(\mathbf{g}) \xrightarrow[t \to \infty]{a.s.} \mathbb{E}_{\boldsymbol{\pi}}(\mathbf{g})$. Similarly to the asymptotic variance $\sigma_X^2(g)$ for a scalar-valued function $g(\cdot)$, we can also define the *asymptotic covariance* matrix $\boldsymbol{\Sigma}_X(\mathbf{g})$ for vector-valued function $\mathbf{g}(\cdot)$,

$$\boldsymbol{\Sigma}_X(\mathbf{g}) \triangleq \lim_{t \to \infty} t \cdot \text{Var}(\hat{\mu}_t(\mathbf{g})) = \lim_{t \to \infty} \frac{1}{t} \cdot \mathbb{E} \left\{ \Delta_t \Delta_t^T \right\}, \tag{8}$$

---

[6] In practice, knowing $\pi_i$ up to a multiplicative constant is enough to converge to the optimal point.

where $\Delta_t \triangleq \sum_{s=1}^t (\mathbf{g}(X_s) - \mathbb{E}_{\boldsymbol{\pi}}(\mathbf{g}))$. The associated multivariate CLT is then given as follows.

**Theorem 2.1** (Chapter 1 [19])**.** *For any function* $\mathbf{g} : [n] \to \mathbb{R}^d$ *that satisfies* $\mathbb{E}_{\boldsymbol{\pi}}(\mathbf{g}^2) < \infty$*, we have*

$$\sqrt{t} \cdot [\hat{\mu}_t(\mathbf{g}) - \mathbb{E}_{\boldsymbol{\pi}}(\mathbf{g})] \xrightarrow[t \to \infty]{dist} \mathcal{N}(0, \boldsymbol{\Sigma}_X(\mathbf{g})). \qquad \square$$

In the next section, we will show how the the asymptotic covariance matrix $\boldsymbol{\Sigma}_X(\cdot)$ also appears as part of the CLT result for SGD algorithms.

## 3 Efficiency Ordering of SGD Algorithms

In this section, we present our main result concerning the performance comparison of different SGD algorithms to solve (1). We first begin by stating our assumptions on the objective function and the stochastic input sequence, providing a CLT result for SGD algorithms, and analyzing the covariance matrix arising therein. We then introduce the notion of *efficiency ordering* of Markov chains in the context of MCMC sampling, and form the connection with covariance matrices as our main result in Theorem 3.6.

For the rest of this section we assume that the functions $F(\cdot, i)$ (possibly non-convex), the summands of the objective function in (1), and the input process $\{X_t\}_{t \geq 0}$ for the SGD iteration (7) satisfy:

(A1) The step size is given by $\gamma_t = t^{-\alpha}$ for $\alpha \in (1/2, 1]$;

(A2) There exists a unique minimizer $\theta^*$ in the interior of the compact set $\Theta$ with $\nabla f(\theta^*) = 0$, and matrix $\nabla^2 f(\theta^*)$ (resp. $\nabla^2 f(\theta^*) - \mathbf{I}/2$) is positive definite for $\alpha \in (1/2, 1)$ (resp. $\alpha = 1$);

(A3) Gradients are bounded in the compact set $\Theta$, that is, $\sup_{\theta \in \Theta} \sup_{i \in [n]} \|\nabla F(\theta, i)\|_2 < \infty$;

(A4) For every $z \in [n], \theta \in \mathbb{R}^d$, the solution $\tilde{F}(\theta, z) \in \mathbb{R}^d$ of the Poisson equation $\tilde{F}(\theta, z) - \mathbb{E}[\tilde{F}(\theta, X_{t+1}) \mid X_t = z] = \nabla F(\theta, z) - \nabla f(\theta)$ exists, and $\sup_{\theta \in \Theta, z \in [n]} \|\tilde{F}(\theta, z)\|_2 < \infty$;

(A5) The functions $F(\theta, i)$ are $L$-smooth for all $i \in [n]$, that is, $\forall \theta_1, \theta_2 \in \Theta, \forall i \in [n]$, we have $\|\nabla F(\theta_1, i) - \nabla F(\theta_2, i)\|_2 \leq L \|\theta_1 - \theta_2\|_2$.

We then have the following CLT result for SGD algorithms.

**Lemma 3.1.** *For iterates* $\{\theta_t\}_{t \geq 0}$ *of the SGD algorithm* (7) *satisfying (A1)–(A5), we have*

$$\theta_t \xrightarrow[t \to \infty]{a.s.} \theta^*, \quad and \quad (\theta_t - \theta^*)/\sqrt{\gamma_t} \xrightarrow[t \to \infty]{Dist} \mathcal{N}(0, \mathbf{V}_X), \tag{9}$$

*where covariance matrix* $\mathbf{V}_X$ *is the unique solution to the Lyapunov equation* $\boldsymbol{\Sigma}_X + \mathbf{K} \mathbf{V}_X + \mathbf{V}_X \mathbf{K}^T = \mathbf{0}$ *when* $\alpha \in (0.5, 1)$ *(resp.* $\boldsymbol{\Sigma}_X + (\mathbf{K} + \frac{\mathbf{I}}{2})\mathbf{V}_X + \mathbf{V}_X (\mathbf{K} + \frac{\mathbf{I}}{2})^T = \mathbf{0}$ *when* $\alpha = 1$*). Here,* $\boldsymbol{\Sigma}_X \triangleq \boldsymbol{\Sigma}_X(\nabla G(\theta^*, \cdot))$ *is the asymptotic covariance matrix[7] as in* (8)*, and* $\mathbf{K} \triangleq \nabla^2 f(\theta^*)$*.*

*Additionally, for the averaged iterates* $\{\bar{\theta}_t\}_{t \geq 0}$ *where* $\bar{\theta}_t \triangleq \frac{1}{t} \sum_{i=0}^{t-1} \theta_t$*, we have*

$$\bar{\theta}_t \xrightarrow[t \to \infty]{a.s.} \theta^*, \quad and \quad \sqrt{t}(\bar{\theta}_t - \theta^*) \xrightarrow[t \to \infty]{Dist} \mathcal{N}(0, \mathbf{V}_X'), \tag{10}$$

*where* $\mathbf{V}_X' = \mathbf{K}^{-1} \boldsymbol{\Sigma}_X (\mathbf{K}^{-1})^T$ *with the same matrices* $\mathbf{K}$ *and* $\boldsymbol{\Sigma}_X$ *as in the non-averaged case.* $\square$

**Remark 3.2.** Lemma 3.1 is itself a special case of the more general CLT result for SA algorithms provided in Appendix A [1], and as proved in Appendix B [1]. $\square$

**Remark 3.3.** While (A2) may appear to be too strict at first, it can be relaxed to the setting of the objective function $f(\cdot)$ having multiple minimizers, by leveraging more general CLT results from SA literature, such as Theorem 2.1 in [28]. However, this comes at a cost of cumbersome notation, requiring conditioning of iterates converging to one of the minimizers, potentially making the mathematical parts harder to follow. We also show in Appendix C [1] that (A2) is no stricter than the Polyak-Łojasiewicz inequality – a popularly adopted weak assumption in recent SGD literature studying non-convex objective functions [36, 47, 70, 72]. $\square$

---

[7]We slightly abuse the notation and shorten $\boldsymbol{\Sigma}_X(\nabla G(\theta^*, \cdot))$, that is, the asymptotic covariance matrix evaluated at $\nabla G(\theta^*, \cdot)$, to $\boldsymbol{\Sigma}_X$ for better readability. In this paper, $\boldsymbol{\Sigma}_X(\nabla G(\theta^*, \cdot))$ and $\boldsymbol{\Sigma}_X$ are equivalent.

**Remark 3.4.** Assumptions (A3) and (A5) are widely seen in the RWSGD literature [57, 34, 67], while (A4) is automatically satisfied for any ergodic Markov chain (see [49, 21] for details), a common assumption for the stochastic noise sequence [34, 25, 7]. The compactness in (A3) can also be relaxed, given assumptions on the objective function in [37], such that the estimator $\theta_t$ generated by Markov-driven sequences can still be 'locked in' a compact set after a sufficiently long time. □

Lemma 3.1 implicitly indicates that the asymptotic convergence rate (in distribution) for $\theta_t - \theta^*$ (resp. $\bar{\theta}_t - \theta^*$) is $O(\sqrt{\gamma_t})$ (resp. $O(1/\sqrt{t})$). While this does not necessarily translate to $O(\sqrt{\gamma_t})$ convergence rate for $\mathbb{E}[\|\theta_t - \theta^*\|_2]$ ($O(1/\sqrt{t})$ for $\mathbb{E}[\|\bar{\theta}_t - \theta^*\|_2]$), it has been suggested [20, 23], and is in fact true for cases such as quadratic objective functions since they satisfy the linear stochastic approximation in [21], which is of the form

$$\theta_{t+1} = \theta_t - \gamma_{t+1}(\mathbf{A}\theta_t - \mathbf{b}(X_{t+1})), \tag{11}$$

for which the connection between finite-time MSE and covariance matrix $\mathbf{V}_X$ has been established [21]. This is also true for arbitrarily weighted MSE, which can be obtained as a weighted sum of diagonal entries of the covariance matrices $\mathbf{V}_X$ and $\mathbf{V}'_X$.

In addition to the apparent connection to MSE, the covariance matrix plays a wider role in SGD performance. Given any vector of weights $\mathbf{w} \in \mathbb{R}^d$, from Lemma 3.1 we also have that the weighted sum of errors $\mathbf{w}^T(\theta_t - \theta^*)$ converges to zero almost surely, and that $\mathbf{w}^T(\theta_t - \theta^*)/\sqrt{\gamma_t} \xrightarrow[t\to\infty]{Dist} \mathcal{N}(0, \mathbf{w}^T\mathbf{V}_X\mathbf{w})$. This means that, for sufficiently large $t$, we can estimate

$$P\left(\frac{\mathbf{w}^T(\theta_t - \theta^*)}{\sqrt{\gamma_t \mathbf{w}^T\mathbf{V}_X\mathbf{w}}} > \alpha\right) \approx \frac{1}{2\pi}\int_\alpha^\infty e^{-x^2/2}dx,$$

such that, for instance, the 95% confidence interval for $\mathbf{w}^T\theta_t$ is approximately $\mathbf{w}^T\theta^* \pm 2\sqrt{\gamma_t \mathbf{w}^T\mathbf{V}_X\mathbf{w}}$. In other words, smaller $\mathbf{w}^T\mathbf{V}_X\mathbf{w}$ leads to narrower confidence interval and higher accuracy. The form $\mathbf{w}^T\mathbf{V}_X\mathbf{w}$ for any vector $\mathbf{w} \in \mathbb{R}^d$ naturally implies that Loewner ordering should come into play when concerning the performance of SGD algorithms.

To proceed, we first employ the widely used notion of *efficiency ordering* of Markov chains. The efficiency of different chains is compared by ordering them using their respective AV as follows.

**Definition 3.5** (**Efficiency Ordering** [50]). For two random walks $\{X_t\}_{t\geq0}$ and $\{Y_t\}_{t\geq0}$ with the same stationary distribution $\boldsymbol{\pi}$, we say $\{X_t\}_{t\geq0}$ is more *efficient* than $\{Y_t\}_{t\geq0}$, which we write as $X \geq_E Y$, if and only if $\sigma_X^2(g) \leq \sigma_Y^2(g)$ for any $g: [n] \to \mathbb{R}$. □

We are now ready to state our main result. We first extend the efficiency ordering of Markov chains by proving the equivalence of comparing their scalar-valued AVs, to comparing their asymptotic covariance matrices via Loewner ordering. We then use this extension to show that more efficient inputs $\{X_t\}_{t\geq0}$ (as in Definition 3.5) to the SGD algorithm lead to performance improvements in the form of smaller covariance matrices in the Loewner ordering sense.

**Theorem 3.6.** *Consider the SGD iteration (7) with two random walks $\{X_t\}_{t\geq0}$ and $\{Y_t\}_{t\geq0}$ as input sequences, with the same stationary distribution $\boldsymbol{\pi}$, satisfying (A1)–(A5). Then,*

  *(i) $X \geq_E Y$ if and only if $\boldsymbol{\Sigma}_X(\mathbf{g}) \leq_L \boldsymbol{\Sigma}_Y(\mathbf{g})$ for any vector-valued function $\mathbf{g}$;*

  *(ii) If $\boldsymbol{\Sigma}_X(\nabla G(\theta^*, \cdot)) \leq_L \boldsymbol{\Sigma}_Y(\nabla G(\theta^*, \cdot))$, then $\mathbf{V}_X \leq_L \mathbf{V}_Y$ ($\mathbf{V}'_X \leq_L \mathbf{V}'_Y$ for the case of averaged iterates);*

*where function $\nabla G(\theta^*, \cdot): [n] \to \mathbb{R}^d$ is defined in the SGD iteration (7), $\mathbf{V}_X$ and $\mathbf{V}'_X$ (resp. $\mathbf{V}_Y$ and $\mathbf{V}'_Y$) are the covariance matrices from Lemma 3.1, corresponding to $\{X_t\}_{t\geq0}$ (resp. $\{Y_t\}_{t\geq0}$) as the stochastic input sequence.* □

Theorem 3.6 enables us to provide a sense of *efficiency ordering of SGD algorithms* which are driven by different stochastic inputs. Since this is achieved via Loewner ordering, it also leads to smaller confidence intervals in the long run as mentioned earlier, as well as potentially smaller MSE[8] depending on the objective function.

---

[8]The mean square error can be retrieved as the trace of the covariance matrix (weighted sum of its diagonal entries in case of weighted MSE). Loosely speaking, an iterate having a smaller covariance matrix in the Loewner ordering will then also have a smaller MSE (weighted MSE).

**Remark 3.7.** In addition to the CLT result for SGD algorithms with diminishing step size described in Lemma 3.1, we include in Appendix E [1] similar results for constant step sizes and quadratic objective functions, where the statement of Theorem 3.6 still holds. $\qquad\square$

## 4 Applications: Towards More Efficient SGD

In this section, we present some SGD variants and compare them in terms of efficiency ordering of SGD. Specifically, we first show that a certain class of non-Markov random walks can provide a better input sequence than its Markovian counterpart. We then analyze shuffling-based gradient descent and compare it to the SGD with *i.i.d* input in terms of efficiency ordering for SGD algorithm. We also extend our approach to a more general mini-batch version, the discussion for which is deferred to Appendix H.3 [1].

**High-Order Efficient Random Walk for SGD:** The simple random walk (SRW) is a popular Markov chain that has been extensively studied in the literature [62, 58, 29]. Several recent works have focused on the non-backtracking random walk (NBRW) on a connected undirected graph $\mathcal{G}(\mathcal{V}, \mathcal{E})$ in the MCMC literature, which is an extension of SRW with the same limiting distribution $\boldsymbol{\pi} = \mathbf{d}/\mathbf{1}^T\mathbf{d}$ [53, 5, 41, 38, 10]. Intuitively speaking, NBRW is a random walk that selects one of its neighbors uniformly at random *except* the one it just came/transitioned from. Specifically, the NBRW $\{Y_t\}_{t\geq 0}$ is a second-order non-reversible Markov chain (i.e., it is non-Markov on $\mathcal{V} = [n]$) with its transition probability given by

$$P(Y_{t+1} = j | Y_t = i, Y_{t-1} = k) = \begin{cases} \frac{1}{d_i-1} & \text{if } j \neq k, j \in N(i), d_i > 1, \\ 1 & \text{if } d_i = 1, j \in N(i), \\ 0 & \text{otherwise.} \end{cases} \tag{12}$$

Since the limiting distributions of NBRW and SRW are the same, NBRW can be used as the input for SGD iterations (7) with the same re-weighted local functions $G(\theta^*, i)$ as that of SRW for all $i \in [n]$ whenever the applications call for random-walk type of inputs. Let $\boldsymbol{\Sigma}_Y(\nabla G(\theta^*, \cdot))$ be the asymptotic covariance matrix of this NBRW $\{Y_t\}_{t\geq 0}$, as defined in (8). One of the main results in [41] concerns the efficiency ordering of NBRW and SRW. They show that NBRW has a smaller AV, or equivalently, from our Theorem 3.6 (i), a smaller asymptotic covariance in terms of Loewner ordering. Our next result forms the necessary connection between the asymptotic covariance matrix arising in the CLT result and $\boldsymbol{\Sigma}_Y(\nabla G(\theta^*, \cdot))$.

**Proposition 4.1.** *Consider the SGD iteration* (7) *with two input sequences SRW* $\{X_t\}_{t\geq 0}$ *and NBRW* $\{Y_t\}_{t\geq 0}$ *respectively. Then, both the respective estimators* $\theta_t^X, \theta_t^Y \xrightarrow[t\to\infty]{a.s.} \theta^*$, *and* $\mathbf{V}_Y \leq_L \mathbf{V}_X$, *that is, NBRW is more efficient than SRW in the SGD algorithm.* $\qquad\square$

By augmenting the state space, we can represent NBRW as a Markov chain $Z_t = (Y_{t-1}, Y_t) \in \mathcal{V} \times \mathcal{V}$, as was done in [53, 41]. This transformation then allows us to build CLT for an SGD iteration with $\{Z_t\}_{t\geq 0}$ as the input. The subtlety here is to prove that the asymptotic covariance matrix arising out of the CLT with respected to the augmented process $\{Z_t\}_{t\geq 0}$ is indeed equal to $\boldsymbol{\Sigma}_Y(\nabla G(\theta^*, \cdot))$. This is shown by cultivating the relationship between the stationary distribution of $\{Z_t\}_{t\geq 0}$ on the augmented state space $\mathcal{V} \times \mathcal{V}$ and $\{Y_t\}_{t\geq 0}$ on the node space $\mathcal{V}$, as provided in [53].

Thus, our Theorem 3.6 together with the existing works on efficiency ordering of NBRW versus SRW in the MCMC literature [53, 41] enable us to show that NBRW is a more efficient input sequence than SRW for the SGD iteration (7). Interestingly, it has been shown that non-backtracking walks mix faster when the underlying graph is $d-$regular [5]. In this case, a faster convergence rate is also suggested by mixing time based non-asymptotic bounds prevalent in RWSGD literature. However, no such results concerning mixing time and SLEM exists for NBRW on a general graph. Thus, in the form of Proposition 4.1, we demonstrate the utility of our approach in settings where mixing time based comparisons are unavailable.

**Shuffling versus *i.i.d* Input Sequence:** Shuffling-based methods have been widely used in machine learning applications [13]. They work by repeatedly passing over the entire state space $[n]$ without repetition, each complete pass forming an *epoch*. *Random shuffling* and *single shuffling* are two versions therein and differ in the order in which they pass over $[n]$. Random shuffling, as the name suggests, makes the pass in a randomly chosen order in each epoch, while single shuffling maintains the same predetermined order (often randomly chosen once at the beginning) for all

epochs. Shuffling-based methods are known to show better empirical performance than *i.i.d* input [15], although intense theoretical analysis for shuffling-based gradient descent has only emerged in recent years [66, 33, 64, 3, 32]. In what follows, we use our results from Section 3 to compare shuffling-based gradient descent to SGD with *i.i.d* input. To do so, we first analyze the asymptotic covariance matrix for shuffling-based methods.

**Lemma 4.2.** *Let the input process* $\{X_t\}_{t\geq 0}$ *be single or random shuffling. Then, for any vector-valued function* $\mathbf{g} : [n] \to \mathbb{R}^d$, $\boldsymbol{\Sigma}_X(\mathbf{g}) = \mathbf{0}$*, where* $\boldsymbol{\Sigma}_X(\mathbf{g})$ *is defined in* (8). $\qquad\square$

For *i.i.d* input sequence with distribution $\hat{\boldsymbol{\pi}}$, the asymptotic covariance from Lemma 3.1 reduces to

$$\boldsymbol{\Sigma}_X(\nabla G(\theta^*, \cdot)) \triangleq \text{Var}_{X_0 \sim \hat{\boldsymbol{\pi}}}\left(\nabla G(\theta^*, X_0)\right) \tag{13}$$

following its definition in (8), and thus, trivially, $\boldsymbol{\Sigma}_X(\nabla G(\theta^*, \cdot)) \geq_L \mathbf{0}$. Lemma 4.2 shows that shuffling-based methods are more efficient than *i.i.d* input sequence due to a smaller asymptotic covariance matrix in Loewner ordering. Next, we show that they also outperform *i.i.d* input when used for driving the input sequence of SGD algorithms.

**Proposition 4.3.** *Consider the SGD iteration* (7) *with stochastic inputs single/random shuffling* $\{X_t\}_{t\geq 0}$ *and i.i.d sampling* $\{Y_t\}_{t\geq 0}$*, we have* $\theta_t^X, \theta_t^Y \xrightarrow[t\to\infty]{a.s.} \theta^*$ *and* $\mathbf{V}_X = \mathbf{0} \leq_L \mathbf{V}_Y$. $\qquad\square$

Though it may seem so at first, Proposition 4.3 is not a simple application of Theorem 3.6, especially for random shuffling because it is hard to check if random shuffling, formulated as a time-inhomogeneous Markov chain, indeed satisfies (A4). To overcome this difficulty, in Appendix H [1] we come up with a non-trivial augmentation to a much higher dimensional state space ($[n]^{n+1}$) to make random shuffling a time-homogeneous periodic Markov chain in order to show that both single shuffling and random shuffling satisfy (A4) and thus apply Theorem 3.6.

The case of shuffling versus *i.i.d* inputs is an example of a setting where the sequence with larger SLEM[9] is more efficient than one with smaller SLEM[9] as an input sequence to the SGD iteration (7). For quadratic objective functions that satisfy the linear SA iteration in [21], it also attains a faster convergence speed in terms of MSE than *i.i.d* inputs to SGD algorithms. Although some recent works provide more informative finite-time error bounds on the MSE of the objective function for shuffling-based methods, by studying a special case of the matrix norm AM-GM inequality and proving faster convergence rate than *i.i.d* inputs [56, 3, 32], our result is not a subset of theirs. To be precise, we show in Appendix C [1] that our assumption (A2) on the objective function is no less general than their most general setting based on the Polyak-Łojasiewicz inequality.

**Remark 4.4.** Mini-batch gradient descent is another popular gradient descent variant and is widely used in the machine learning tools [22, 2, 55] to accelerate the learning process when compared to SGD. In Appendix H.3 [1] we show how our framework can be applied to study min-batch based SGD algorithms, and include the asymptotic analysis on mini-batch gradient descent with shuffling.

Besides mini-batch gradient descent, there are other SGD variants, e.g., momentum SGD, Nesterov accelerated SGD and ADAM, that have been studied in the SGD literature for non-asymptotic analysis [39, 59, 6]. However, asymptotic analysis on the SGD variants is not well studied in the literature, with only very recent results on the CLT for *i.i.d* input sequences [42, 8, 9, 45]. Asymptotic analysis and CLT for variants more general Markovian input sequences, which would be a prerequisite for Theorem 3.6, remains an open problem. We defer the discussion on related works to Appendix I.3 [1], where we also empirically test the SGD variants and find that the efficiency ordering result still holds for these SGD variants - opening up the possibility for further theoretical analysis. $\qquad\square$

## 5 Numerical Experiments

In this section, we empirically validate our theoretical analysis. We select two convex objective functions as follows.

$$\tilde{f}(\theta) = \frac{1}{n}\sum_{i=1}^{n}\log(1+\exp(-y_i\mathbf{x}_i^T\theta)) + \frac{1}{2}\|\theta\|_2^2 \,, \quad \hat{f}(\theta) = \frac{1}{n}\sum_{i=1}^{n}\theta^T(\mathbf{a}_i\mathbf{a}_i^T + \mathbf{D}_i)\theta + \mathbf{b}^T\theta. \tag{14}$$

For $l_2$-regularized logistic regression $\tilde{f}(\theta)$, we choose the dataset CIFAR-10 [40] where $n$ is the total number of data points. Here, $\mathbf{x}_i \in \mathbb{R}^{108}$ is the vector flattened from the cropped image $i$ with

---

[9]The single shuffling when realized as a periodic Markov chain has SLEM = 1 (transition matrix is unitary), while the *i.i.d* input sequence has SLEM = 0 (transition matrix is rank one).

shape $(6, 6, 3)$, and $y_i \in \mathbb{R}$ is the label. For sum-of-non-convex functions $\hat{f}(\theta)$, which is based on the experiment setup in [31, 4], we generate random vectors $\mathbf{a}_i, \mathbf{b}$ and matrices $\mathbf{D}_i$ which ensure the invertibility of matrix $\sum_{i=1}^{n} \mathbf{a}_i \mathbf{a}_i^T$ and $\sum_{i=1}^{n} \mathbf{D}_i = \mathbf{0}$ (details are deferred to Appendix I.2 [1]). For both experiments, we assign a data point to each node $i$ on the general graph 'Dolphins' (62 nodes) [63]. We set the step size in the SGD algorithm as $1/t^{0.9}$, and use MSE $\mathbb{E}[\|\theta_t - \theta^*\|_2^2]$ to measure the relative performance of different inputs. We also employ the scaled MSE $\mathbb{E}[\|\theta_t - \theta^*\|_2^2]/\gamma_t$ to empirically show its relationship to the CLT result (9). Due to space constraints, additional simulation results which support our efficiency ordering result in the setting of large graphs, and non-convex functions are deferred to Appendix I.3 in [1]. Therein, via numerical simulations, we also observe the efficiency ordering for other SGD variants such as Nesterov accelerated SGD and ADAM when comparing their iterations under efficiency ordered noise sequences.

In Figure 2 we compare NBRW and SRW as input sequences on the graph 'Dolphins' for two objective functions in (14). We also compare uniform sampling, random shuffling and single shuffling, assuming that they can access any node on the graph in each iteration. We can see in Figure 2a and 2c that NBRW always falls below SRW throughout all time periods, which indicates that NBRW tends to have smaller MSE than SRW. Single and random shuffling are both better than uniform sampling in terms of smaller MSE. The oscillation of single shuffling comes from a predetermined fixed data sampling sequence, while random shuffling changes the permutation whenever traversing all nodes. Such oscillation is not visible in Figure 2c and Figure 2d because it is small on the current y-axis scale. The curves of single and random shuffling in Figure 2b and 2d fall below that of uniform sampling and still decrease in the linear rate because eventually their covariance matrices will be zero matrix, as indicated in Proposition 4.3. Figure 2b shows that the scaled MSEs of NBRW, SRW and uniform sampling approach some constants after some time, which is consistent with the CLT result (9). The curves of NBRW are still below that of SRW, showing that the input with smaller scaled MSE tends to have higher efficiency, which supports Proposition 4.1. We can see from Figure 2d that the curves NBRW, SRW are still increasing andd they have not yet entered the regime where the covariance matrix becomes the main factor. On the other hand, uniform sampling and both shuffling methods are just entering this regime.

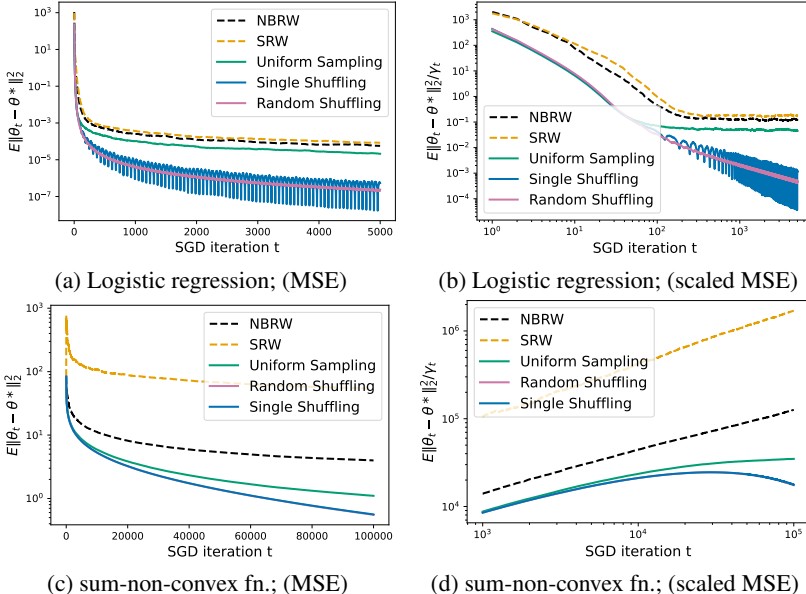

(a) Logistic regression; (MSE)   (b) Logistic regression; (scaled MSE)

(c) sum-non-convex fn.; (MSE)   (d) sum-non-convex fn.; (scaled MSE)

Figure 2: Performance comparison of different stochastic inputs on the graph 'Dolphins'.

# 6 Conclusion

We have introduced the notion of efficiency ordering of SGD algorithms, and shown that processes with smaller asymptotic covariance are always more efficient as input sequences for SGD algorithms. Furthermore, we numerically observe that this sense of efficiency ordering is also seen SGD variants such as Nesterov accelerated SGD and ADAM. Since the asymptotic analysis of such SGD variants is not well-established under general stochastic inputs, establishing theoretical results on efficiency ordering remain an open problem.

## 7 Acknowledgments and Disclosure of Funding

We thank the anonymous reviewers for their constructive comments. The research was conducted while Vishwaraj Doshi was with the Operations Research Graduate Program, North Carolina State University. This work was supported in part by National Science Foundation under Grant Nos. CNS-2007423, IIS-1910749, and CNS-1824518.

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
