# Efficiency Ordering of Stochastic Gradient Descent – Supplementary Material

**Jie Hu**[1][*]     **Vishwaraj Doshi**[2][*]     **Do Young Eun**[1]

[1] Department of Electrical and Computer Engineering, North Carolina State University
[2] Data Science and Advanced Analytics, IQVIA

jhu29@ncsu.edu     vishwaraj.doshi@iqvia.com     dyeun@ncsu.edu

## Contents

---

[*]Equal contributors.

36th Conference on Neural Information Processing Systems (NeurIPS 2022).

In this supplementary material, any theorems, lemmas, and propositions that are indexed without citation are referred to our paper [17].

## A    CLT for Stochastic Approximation Algorithms

The existing central limit theorem (CLT) for stochastic approximation (SA) with Markovian dynamics [4, 11, 14] usually studied a general Markov process $\{X_t\}_{t\geq 0}$ on the finite state space $\mathcal{V}$ and its transition kernel $P_\theta$ dependent on $\theta$ such that $P(X_{t+1} \in A | X_t = x, \theta_t = \theta) = P_\theta(x, A)$ for any subset $A \subseteq \mathcal{V}$. Denote $\pi_\theta$ the stationary distribution of $P_\theta$. Define $P_\theta v_\theta(x) \triangleq \sum_{l \in \mathcal{V}} [P_\theta]_{x,l} \cdot v_\theta(l)$. The general SA algorithm is of the form

$$\theta_{t+1} = \text{Proj}_\Theta \left( \theta_t + \gamma_{t+1} H(\theta_t, X_{t+1}) \right), \tag{15}$$

where $\Theta \subset \mathbb{R}^d$ is a closed and convex set. The main goal is to find the root $\theta^*$ of function

$$h(\theta) \triangleq \mathbb{E}_{X \sim \pi_\theta}[H(\theta, X)] \text{ i.e., } h(\theta^*) = 0.$$

As mentioned in [4] p.332 Theorem 13, and [11] p.31 Theorem 15, p.59 Theorem 25, the usual assumptions are given as

(B1)  Function $h : \Theta \to \mathbb{R}^d$ is continuous on $\Theta$, there exists a non-negative $C^1$ function $V$ such that $\langle \nabla V(\theta), h(\theta) \rangle \leq 0, \forall \theta \in \Theta$ and the set $\mathcal{S} = \{\theta; \langle \nabla V(\theta), h(\theta) \rangle = 0\}$ is such that $V(\mathcal{S})$ has empty interior. Also, $V(\theta)$ tends to $+\infty$ if $\theta \to \partial\Theta$, where $\partial\Theta$ is the boundary of $\Theta$, or $\|\theta\|_2 \to \infty$. There exists a compact set $\mathcal{K} \subset \Theta$ such that $\langle \nabla V(\theta), h(\theta) \rangle < 0$ if $\theta \notin \mathcal{K}$;

(B2)  For every $\theta$, there exist a function $v_\theta(x)$ such that the Poisson equation

$$v_\theta(x) - \mathbb{E}[v_\theta(X_{t+1}) | X_t = x, \theta_t = \theta] = H(\theta, X) - h(\theta). \tag{16}$$

For any compact set $\mathcal{C} \subset \Theta$,

$$\sup_{\theta \in \mathcal{C}, x \in \mathcal{V}} \|H(\theta, x)\|_2 + \|v_\theta(x)\|_2 < \infty. \tag{17}$$

There exists a continuous function $\phi_\mathcal{C}, \phi_\mathcal{C}(0) = 0$, such that for any $\theta, \theta' \in \mathcal{C}$,

$$\sup_{X \in \mathcal{V}} \|P_\theta v_\theta(x) - P_{\theta'} v_{\theta'}(x)\|_2 \leq \phi_\mathcal{C}\left( \|\theta - \theta'\|_2 \right). \tag{18}$$

(B3)  The step size follows $\gamma_t \geq 0, \sum_{t \geq 1} \gamma_t = \infty, \sum_{t \geq 1} \gamma_t^2 < \infty$ and $\sum_{t \geq 1} |\gamma_{t+1} - \gamma_t| < \infty$.

(B4)  Assume $\theta_t$ converges to some limit $\theta^* \in \mathcal{S}$. Function $h$ is $C^1$ in some neighborhood of $\theta^*$ with first derivatives Lipschitz, and matrix $\nabla h(\theta^*)$ has all its eigenvalues with negative real part.

Then, we have the following convergence and CLT result.

**Theorem A.1.**  *[4, 11, 14] Assume $\theta_t$ is given by the SA iteration (15) that satisfies assumptions (B1) – (B3) above, then iterate $\theta_t$ converges almost surely to the set $\mathcal{S}$ defined in (B1). Moreover, with additional assumption (B4), we have*

$$\frac{1}{\sqrt{\gamma_t}} \cdot (\theta_t - \theta^*) \xrightarrow[t \to \infty]{Dist} \mathcal{N}(0, \mathbf{V}_X), \tag{19}$$

*where covariance matrix $\mathbf{V}_X$ is the unique solution to the following Lyapunov equation:*

$$\begin{cases} \mathbf{\Sigma}_X + \mathbf{K}\mathbf{V}_X + \mathbf{V}_X \mathbf{K}^T = \mathbf{0} & \text{if } \alpha \in (\frac{1}{2}, 1), \\ \mathbf{\Sigma}_X + \left(\mathbf{K} + \frac{\mathbf{I}}{2}\right)\mathbf{V}_X + \mathbf{V}_X \left(\mathbf{K} + \frac{\mathbf{I}}{2}\right)^T = \mathbf{0} & \text{if } \alpha = 1. \end{cases} \tag{20}$$

*Here, $\mathbf{K} \triangleq \nabla h(\theta^*)$ and $\mathbf{\Sigma}_X \triangleq \mathbf{\Sigma}_X(H(\theta^*, \cdot))$ is the asymptotic covariance matrix as in (8), evaluated at function $H(\theta^*, \cdot)$.*

*In addition, for averaged iterates $\bar{\theta}_t \triangleq \frac{1}{t} \sum_{i=0}^{t-1} \theta_t$, we still have $\bar{\theta}_t \xrightarrow[t \to \infty]{a.s.} \theta^*$, and*

$$\sqrt{t} \cdot (\bar{\theta}_t - \theta^*) \xrightarrow[t \to \infty]{Dist} \mathcal{N}(0, \mathbf{V}_X'), \tag{21}$$

*where $\mathbf{V}_X' = \mathbf{K}^{-1}\mathbf{\Sigma}_X(\mathbf{K}^{-1})^T$ with the same matrices $\mathbf{K}$ and $\mathbf{\Sigma}_X$ as in (20).*  $\square$

# B   Proof of Lemma 3.1

To prove Lemma 3.1 with existing Theorem A.1, we need to show that (A1) – (A5) is a special case of (B1) – (B4). We list (A1) – (A5) here for self-contained purpose.

(A1) The step size is given by $\gamma_t = t^{-\alpha}$ for $\alpha \in (1/2, 1]$;

(A2) There exists a unique minimizer $\theta^*$ in the interior of the compact set $\Theta$ with $\nabla f(\theta^*) = 0$, and matrix $\nabla^2 f(\theta^*)$ (resp. $\nabla^2 f(\theta^*) - \mathbf{I}/2$) is positive definite for $a \in (1/2, 1)$ (resp. $a = 1$);

(A3) Gradients are bounded in the compact set $\Theta$, that is, $\sup_{\theta \in \Theta} \sup_{i \in [n]} \|\nabla F(\theta, i)\|_2 < \infty$;

(A4) For every $z \in [n], \theta \in \mathbb{R}^d$, the solution $\tilde{F}(\theta, z) \in \mathbb{R}^d$ of the Poisson equation

$$\tilde{F}(\theta, z) - \mathbb{E}[\tilde{F}(\theta, X_{t+1}) \mid X_t = z] = \nabla F(\theta, z) - \nabla f(\theta) \tag{22}$$

exists, and $\sup_{\theta \in \Theta, z \in [n]} \|\tilde{F}(\theta, z)\|_2 < \infty$;

(A5) The functions $F(\theta, i)$ are $L$-smooth for all $i \in [n]$, that is, $\forall \theta_1, \theta_2 \in \Theta, \forall i \in [n]$, we have $\|\nabla F(\theta_1, i) - \nabla F(\theta_2, i)\|_2 \le L\|\theta_1 - \theta_2\|_2$.

Let $H(\theta, X) \triangleq -\nabla F(\theta, X)$ for function $F(\theta, X)$ defined in (1). Then, we have $h(\theta) \triangleq \mathbb{E}_{X \sim \pi}[H(\theta, X)] = -\nabla f(\theta)$. By choosing $V(\theta) \triangleq f(\theta)$, we know $\langle \nabla V(\theta), h(\theta) \rangle = -\nabla f(\theta)^2 \le 0$. From (A2) we know $\theta^*$ is the unique minimizer of function $f$, by letting $\mathcal{K} = \{\theta^*\}$, we have $\langle \nabla V(\theta), h(\theta) \rangle < 0$ when $\theta \notin \mathcal{K}$. Therefore, **(B1) is satisfied**.

Now we need to check assumption (B2). Assumption (A4) is a direct translation to (16) in (B2), and $\sup_{\theta \in \Theta, z \in [n]} \|\tilde{F}(\theta, z)\|_2 < \infty$, as well as assumption (A3), implies (17). We still need to show (18). By assuming an $n$-state ergodic Markov chain $\{X_t\}_{t \ge 0}$ ($\theta$-independent) with transition kernel $\mathbf{P} \in \mathbb{R}^{n \times n}$ and stationary distribution $\boldsymbol{\pi}$, the solution $\tilde{F}(\theta, z)$ to the Poisson equation (22) in (A4) exists and is given as follows.[2]

$$\tilde{F}(\theta, z) = \nabla F(\theta, z) - \nabla f(\theta) + \sum_{l=1}^{n} \mathbf{P}_{z,l}(\nabla F(\theta, l) - \nabla f(\theta)) + \sum_{l=1}^{n} \left[\mathbf{P}^2\right]_{z,l}(\nabla F(\theta, l) - \nabla f(\theta)) + \cdots . \tag{23}$$

Next, we can rewrite $\tilde{F}(\theta, z)$ in the closed form and show that it is Lipschitz continuous and satisfies (20) in assumption (B2). Note that by definition of expectation and Chapman–Kolmogorov equation ($\sum_{k=1}^{n} \sum_{l=1}^{n} \mathbf{P}_{z,k}\mathbf{P}_{k,l} = \sum_{l=1}^{n}[\mathbf{P}^2]_{z,l}$), we have

$$\mathbb{E}[\tilde{F}(\theta, X_{t+1}) \mid X_t = z] = \sum_{l=1}^{n} \mathbf{P}_{z,l}(\nabla F(\theta, l) - \nabla f(\theta)) + \sum_{l=1}^{n} \left[\mathbf{P}^2\right]_{z,l}(\nabla F(\theta, l) - \nabla f(\theta))$$
$$+ \sum_{l=1}^{n} \left[\mathbf{P}^3\right]_{z,l}(\nabla F(\theta, l) - \nabla f(\theta)) + \cdots . \tag{24}$$

Then, from (23) and (24) we have $\tilde{F}(\theta, z) - \mathbb{E}[\tilde{F}(\theta, X_{t+1}) \mid X_t = z] = \nabla F(\theta, z) - \nabla f(\theta)$, which is exactly (22) in (A4). Moreover, since $\mathbf{1}$ and $\boldsymbol{\pi}$ are the right and left eigenvectors of $\mathbf{P}$ respectively with eigenvalue 1, by induction we know

$$\mathbf{P}^k - \mathbf{1}\boldsymbol{\pi}^T = (\mathbf{P} - \mathbf{1}\boldsymbol{\pi}^T)^k, \forall k \in \mathbb{Z}, \ k \ge 1. \tag{25}$$

Along with the fact that

$$\nabla f(\theta) = \sum_{l=1}^{n} \pi_l \nabla F(\theta, l) = \sum_{l=1}^{n} [\mathbf{1}\boldsymbol{\pi}^T]_{z,l} \nabla F(\theta, l), \tag{26}$$

---

[2]In this paper, we only consider $\theta$-independent Markovian inputs. The more general conditions of the $\theta$-dependent Markov chain under which the solution of (22) in (A4) exists are referred to [11] p.71 Theorem 35 or [4] p.217.

we can further simplify and get a closed form of (23), which is given below.

$$
\begin{aligned}
\tilde{F}(\theta, z) &= \nabla F(\theta, z) - \nabla f(\theta) + \sum_{l=1}^{n} \mathbf{P}_{z,l}(\nabla F(\theta, l) - \nabla f(\theta)) + \sum_{l=1}^{n} \left[\mathbf{P}^2\right]_{z,l}(\nabla F(\theta, l) - \nabla f(\theta)) + \cdots \\
&= \sum_{l=1}^{n} \left[\mathbf{P}^0 - \mathbf{1}\boldsymbol{\pi}^T\right]_{z,l} \nabla F(\theta, l) + \sum_{l=1}^{n} \left[\mathbf{P}^1 - \mathbf{1}\boldsymbol{\pi}^T\right]_{z,l} \nabla F(\theta, l) + \cdots \\
&= \left(\sum_{k=0}^{\infty} \sum_{l=1}^{n} \left[\mathbf{P}^k - \mathbf{1}\boldsymbol{\pi}^T\right]_{z,l} \nabla F(\theta, l)\right) - \nabla f(\theta) \\
&= \left(\sum_{l=1}^{n} \left[\sum_{k=0}^{\infty} \left[\mathbf{P}^k - \mathbf{1}\boldsymbol{\pi}^T\right]_{z,l}\right] \nabla F(\theta, l)\right) - \nabla f(\theta) \\
&= \left(\sum_{l=1}^{n} \left[\sum_{k=0}^{\infty} \left[(\mathbf{P} - \mathbf{1}\boldsymbol{\pi}^T)^k\right]_{z,l}\right] \nabla F(\theta, l)\right) - \nabla f(\theta) \\
&= \sum_{l=1}^{n} \left[(\mathbf{I} - \mathbf{P} + \mathbf{1}\boldsymbol{\pi}^T)^{-1}\right]_{z,l} \nabla F(\theta, l) - \nabla f(\theta),
\end{aligned}
\tag{27}
$$

where the second equality comes from (26) and the fifth equality is from (25). Recall the definition $\mathbf{P}F(\theta, z) \triangleq \sum_{l=1}^{n} \mathbf{P}_{z,l} F(\theta, l)$, we can show that

$$
\begin{aligned}
&\|\mathbf{P}\tilde{F}(\theta, z) - \mathbf{P}\tilde{F}(\theta', z)\|_2 \\
&= \left\|\sum_{l=1}^{n} \mathbf{P}_{z,l}\left(\tilde{F}(\theta, l) - \tilde{F}(\theta', l)\right)\right\|_2 \\
&\leq \sum_{l=1}^{n} \mathbf{P}_{z,l}\|\tilde{F}(\theta, l) - \tilde{F}(\theta', l)\|_2 \\
&\leq \sup_{z \in \mathcal{V}} \|\tilde{F}(\theta, z) - \tilde{F}(\theta', z)\|_2 \\
&\leq \sup_{z \in \mathcal{V}} \left\|\sum_{l=1}^{n} \left[(\mathbf{I} - \mathbf{P} + \mathbf{1}\boldsymbol{\pi}^T)^{-1}\right]_{z,l}(\nabla F(\theta, l) - \nabla F(\theta', l))\right\|_2 + \|\nabla f(\theta) - \nabla f(\theta')\|_2 \\
&\leq C \sup_{z \in \mathcal{V}} \|\nabla F(\theta, z) - \nabla F(\theta', z)\|_2 + \|\nabla f(\theta) - \nabla f(\theta')\|_2 \\
&\leq (C+1)L\|\theta - \theta'\|_2
\end{aligned}
\tag{28}
$$

for some constant $C$ related to matrix $(\mathbf{I} - \mathbf{P} + \mathbf{1}\boldsymbol{\pi}^T)^{-1}$, where the first and the third inequalities are from triangular inequality and the last inequality comes from assumption (A5). Note that we have

$$
\|\nabla f(\theta) - \nabla f(\theta')\|_2 = \left\|\sum_{i=1}^{n} \pi_i (\nabla F(\theta, i) - \nabla F(\theta', i))\right\|_2 \leq \sup_{z \in \mathcal{V}} \|\nabla F(\theta, i) - \nabla F(\theta', i)\|_2 \leq L\|\theta - \theta'\|_2
$$

in the last inequality of (28). So (18) is shown and **(B2) is satisfied**.

For assumption (A1) with respect to the conditions on the step size, we know for $a \in (1/2, 1]$, $\sum_{t \geq 1} 1/t^a = \infty$ and $\sum_{t \geq 0} 1/t^{2a} < \infty$. Besides,

$$
\gamma_t - \gamma_{t+1} = \frac{1}{t^a} - \frac{1}{(t+1)^a} = \frac{(t+1)^a - t^a}{t^a(t+1)^a} \leq \frac{(t+1)^a - t^a}{t^{2a}} \leq \frac{1}{t^{2a}}
$$

where the second inequality comes from $(t+1)^a - t^a$ monotone decreasing in $t$ for $a \in (1/2, 1]$. Then, we have $\sum_{t \geq 1} |\gamma_t - \gamma_{t+1}| \leq \sum_{t \geq 1} 1/t^{2a} < \infty$. Then, **(B3) is satisfied**.

Since (B1) – (B3) are satisfied and (A2) assumes unique minimizer such that $\mathcal{K} = \{\theta^*\}$, from Theorem A.1 we know $\theta_t \xrightarrow[t \to \infty]{a.s.} \theta^*$. Along with assumption (A2) on the positive definite matrix $\nabla^2 f(\theta^*)$, **(B4) is satisfied**.

Therefore, (A1)-(A5) implies (B1) – (B4) and all the results from Theorem A.1 can be carried over to Lemma 3.1.

## C  Discussion on Polyak-Lojasiewicz Inequality and Positive Definite Matrix $\nabla^2 f(\theta^*)$

In this part, we discuss the strictness of the condition on the objective function $f$ between Polyak-Lojasiewicz (P-L) inequality and our assumption (A2) - positive definite matrix $\nabla^2 f(\theta^*)$. We say that if a scalar-valued function $f$ satisfies $\mu$-P-L inequality, then for any $\theta \in \mathbb{R}^d$, the following condition holds:

$$\frac{1}{2}\|\nabla f(\theta)\|_2^2 \geq \mu(f(\theta) - f(\theta^*)), \tag{29}$$

where $\nabla f(\theta) \in \mathbb{R}^d$, $f(\theta^*) = \min_{\theta \in \mathbb{R}^d} f(\theta)$ and the minimizer $\theta^*$ belongs to a non-empty solution set. We define a new function

$$g(\theta) \triangleq \frac{1}{2}\|\nabla f(\theta)\|_2^2 - \mu(f(\theta) - f^*).$$

Then, (29) is equivalent to saying $\min_\theta g(\theta) \geq 0$, and the necessary condition to ensure that $\theta^*$ is the local minimizer is $\nabla^2 g(\theta^*) \geq_L \mathbf{0}$ (e.g., Chapter 1.2 [26]). We have

$$\nabla^2 g(\theta) = \left(\nabla^2 f(\theta) - \mu \mathbf{I}\right)\nabla^2 f(\theta) + \mathbf{M} \otimes \nabla f(\theta),$$

where matrix $\mathbf{M}$ is a 3D matrix with dimension $d \times d \times d$ and $\otimes$ is the tensor product. Since $\nabla f(\theta^*) = 0$, we have $\mathbf{M} \otimes (\nabla f(\theta^*)) = 0$. Then, $\nabla^2 g(\theta^*) \geq_L \mathbf{0}$ implies $(\nabla^2 f(\theta^*))^2 \geq_L \mu \nabla^2 f(\theta^*)$. Denote $\lambda_i \geq 0, i = 1, 2, \cdots, d$ the eigenvalues of matrix $\nabla^2 f(\theta^*)$, by spectral decomposition we need $\lambda_i \geq \mu$ or $\lambda_i = 0$ for each $i$. If all the eigenvalues of $\nabla^2 f(\theta^*)$ are no smaller than $\mu$, then $\nabla^2 f(\theta^*)$ is a positive definite matrix by definition. For example, $\mu$-strongly convex objective function $f$ satisfies both P-L inequality and $\nabla^2 f(\theta^*)$ being positive definite. If there exists at least one eigenvalue with zero value, then $\nabla^2 f(\theta^*)$ is no longer positive definite.

On the other hand, positive definite matrix $\nabla^2 f(\theta^*)$ does not necessarily imply P-L inequality. We give a toy example of objective function $f$ that satisfies positive definite matrix $\nabla^2 f(\theta^*)$ while fails to satisfy P-L inequality. For some smooth convex function

$$f(\theta) = \sqrt{\|\theta\|_2^2 + 1} \geq 1,$$

we know

$$f'(\theta) = \frac{\theta}{\sqrt{\|\theta\|_2^2 + 1}}, \quad f''(\theta) = \frac{1}{\sqrt{\|\theta\|_2^2 + 1}}\mathbf{I} - \frac{1}{(\|\theta\|_2^2 + 1)^{3/2}}\theta\theta^T$$

for any $\theta \in \mathbb{R}^d$. Since $\theta^* = \mathbf{0}$, $f(\theta^*) = 1$ and $f''(\theta^*) = \mathbf{I}$ is a positive definite matrix such that this objective function satisfies our assumption (A2). However, for any $\theta \in \mathbb{R}^d$, there always exists a constant $\epsilon > 0$ such that

$$\epsilon(f(\theta) - f(\theta^*)) \geq \|f'(\theta)\|_2^2,$$

which fails to satisfy (29). Therefore, there is no inclusive relationship between P-L inequality and positive definite matrix $\nabla^2 f(\theta^*)$. Both of the conditions can cover different types of functions.

## D  Proof of Theorem 3.6

### D.1  Proof of Theorem 3.6 (i)

We first prove the direction that efficiency ordering implies Loewner ordering. For any vector $\mathbf{v} \triangleq [v_1, v_2, \cdots, v_d]^T \in \mathbb{R}^d \backslash \{\mathbf{0}\}$ and vector-valued function $\mathbf{f}(X) \triangleq [f_1(X), f_2(X), \cdots, f_d(X)]^T$,

with $\mathbf{\Sigma}(\mathbf{f})$ defined in (8) we can get

$$
\begin{aligned}
\mathbf{v}^T\mathbf{\Sigma}(\mathbf{f})\mathbf{v} &= \lim_{t\to\infty}\mathbf{v}^T\mathbf{\Sigma}(\mathbf{f},t)\mathbf{v} \\
&= \lim_{t\to\infty}\frac{1}{t}\mathbb{E}\left\{\mathbf{v}^T\left[\sum_{s=1}^t\left(\mathbf{f}(X_s)-\mathbb{E}_{\boldsymbol{\pi}}\left[\mathbf{f}(X)\right]\right)\right]\left[\sum_{s=1}^t\left(\mathbf{f}(X_s)-\mathbb{E}_{\boldsymbol{\pi}}\left[\mathbf{f}(X)\right]\right)\right]^T\mathbf{v}\right\} \\
&= \lim_{t\to\infty}\frac{1}{t}\mathbb{E}\left\{\left[\sum_{s=1}^t\left(g_{\mathbf{v},\mathbf{f}}(X_s)-\mathbb{E}_{\boldsymbol{\pi}}\left[g_{\mathbf{v},\mathbf{f}}(X)\right]\right)\right]^2\right\} \\
&= \sigma^2(g_{\mathbf{v},\mathbf{f}}),
\end{aligned}
\tag{30}
$$

where function

$$
g_{\mathbf{v},\mathbf{f}}(X) \triangleq v_1 f_1(X) + v_2 f_2(X) + \cdots + v_d f_d(X)
$$

is a linear combination of $f_i(X)$. For two random processes with efficiency ordering and an arbitrary vector-valued function $\mathbf{f}$, $\sigma_X^2(g_{\mathbf{v},\mathbf{f}}) \leq \sigma_Y^2(g_{\mathbf{v},\mathbf{f}})$ for any vector $\mathbf{v}$, which is exactly $\mathbf{v}^T\mathbf{\Sigma}_X(\mathbf{f})\mathbf{v} \leq \mathbf{v}^T\mathbf{\Sigma}_Y(\mathbf{f})\mathbf{v}$. Then, by definition of Loewner ordering, we have $\mathbf{\Sigma}_X(\mathbf{f}) \leq_L \mathbf{\Sigma}_Y(\mathbf{f})$ for any vector-valued function $\mathbf{f}$.

On the other direction, let $\mathbf{v} = [1, 0, \cdots, 0]^T$ and vector-valued function $\mathbf{f}(X) = [g(X), 0, \cdots, 0]^T$, where $g$ can be any scalar-valued function. Then, (30) can be written as $\mathbf{v}^T\mathbf{\Sigma}(\mathbf{f})\mathbf{v} = \sigma^2(g)$ and it holds for any scalar-valued function $g$. For two Markov chains $\{X_t\}, \{Y_t\}$ with $\mathbf{\Sigma}_X(\mathbf{f}) \leq_L \mathbf{\Sigma}_Y(\mathbf{f})$ for any vector-valued function $\mathbf{f}$, we have $\mathbf{v}^T\mathbf{\Sigma}_X(\mathbf{f})\mathbf{v} \leq \mathbf{v}^T\mathbf{\Sigma}_Y(\mathbf{f})\mathbf{v}$ for any vector $\mathbf{v}$. Then, with $\mathbf{v} = [1, 0, \cdots, 0]^T$ we show that $\sigma_X^2(g) \leq \sigma_Y^2(g)$ for any scalar-valued function $g$, which proves the efficiency ordering.

### D.2 Proof of Theorem 3.6 (ii)

We first introduce the closed form of the solution $\mathbf{V}$ to the Lyapunov equation in Lemma 3.1 and the useful lemma on Loewner ordering.

**Lemma D.1** ([8] Theorem 3.16 and (3.160)). *If all the eigenvalues of matrix $\mathbf{K}$ have negative real part, then for every positive-definite matrix $\mathbf{U}$ there exists a unique positive-definite matrix $\mathbf{V}$ satisfying $\mathbf{U} + \mathbf{K}\mathbf{V} + \mathbf{V}\mathbf{K}^T = \mathbf{0}$. The explicit solution $\mathbf{V}$ is given as*

$$
\mathbf{V} = \int_0^\infty e^{\mathbf{K}t}\mathbf{U}e^{(\mathbf{K}^T)t}dt.
\tag{31}
$$

**Lemma D.2** ([29] Theorem 8.2.7). *If two real matrix $\mathbf{A}, \mathbf{B} \in \mathbb{R}^{m\times m}$ are Loewner ordered $\mathbf{A} \leq_L \mathbf{B}$, then $\mathbf{C}\mathbf{A}\mathbf{C}^T \leq_L \mathbf{C}\mathbf{B}\mathbf{C}^T$ for any real matrix $\mathbf{C} \in \mathbb{R}^{m\times m}$.*

From Theorem 3.6 (i), we know efficiency ordering $\sigma_X^2(g) \leq \sigma_Y^2(g)$ for any scalar-valued function $g$ leads to Loewner ordering $\mathbf{\Sigma}_X(\mathbf{f}) \leq_L \mathbf{\Sigma}_Y(\mathbf{f})$ for any vector-valued function $\mathbf{f}$. Consider two random process $\{X_t\}_{t\geq 0}, \{Y_t\}_{t\geq 0}$ with efficiency ordering $X \geq_E Y$, we have $\mathbf{\Sigma}_X \leq_L \mathbf{\Sigma}_Y$. By Lemma D.2 and (20), for any $t$ in (31), we have $e^{\mathbf{K}t}\mathbf{\Sigma}_X e^{(\mathbf{K}^T)t} \leq_L e^{\mathbf{K}t}\mathbf{\Sigma}_Y e^{(\mathbf{K}^T)t}$. Then, for any vector $\mathbf{v} \in \mathbb{R}^d\backslash\{\mathbf{0}\}$, we have

$$
\mathbf{v}^T\mathbf{V}_X\mathbf{v} = \int_0^\infty \mathbf{v}^T e^{\mathbf{K}t}\mathbf{\Sigma}_X e^{(\mathbf{K}^T)t}\mathbf{v}dt \leq \int_0^\infty \mathbf{v}^T e^{\mathbf{K}t}\mathbf{\Sigma}_Y e^{(\mathbf{K}^T)t}\mathbf{v}dt = \mathbf{v}^T\mathbf{V}_Y\mathbf{v},
$$

such that $\mathbf{V}_X \leq_L \mathbf{V}_Y$ by definition of Loewner ordering. Similarly, for averaged iterates, we have $\mathbf{V}'_X \leq_L \mathbf{V}'_Y$ immediately from Lemma D.2 because $\mathbf{\Sigma}_X \leq_L \mathbf{\Sigma}_Y$ and $\mathbf{V}'_X = \mathbf{K}^{-1}\mathbf{\Sigma}_X(\mathbf{K}^{-1})^T$, $\mathbf{V}'_Y = \mathbf{K}^{-1}\mathbf{\Sigma}_Y(\mathbf{K}^{-1})^T$.

## E  Additional Convergence and CLT results for SGD with Constant Step Size and Quadratic Objective Function

Lemma 3.1 has shown the CLT result for general SGD iteration (7) with diminishing step size. A natural question would be if any CLT result exists for the same SGD iteration with constant step size $\gamma$. For the *i.i.d* inputs and a special case of the iteration

$$
\theta_{t+1} = \theta_t - \gamma(\mathbf{A}(X_{t+1})\theta_t - b(X_{t+1})),
\tag{32}
$$

which is usually called linear stochastic approximation in the stochastic approximation literature, it has been studied in [13, 30] that $\theta_t$ forms a Markov chain and its time-averaged iterate $\bar{\theta}_t = \frac{1}{t}\sum_{i=0}^{t-1}\theta_i$ converges to the minimizer $\theta^*$ almost surely and a CLT result is given in Theorem 1 [30]. However, for Markovian inputs $\{X_t\}_{t\geq 0}$, V. Borkar and S. Meyn mentioned in [5] that the behavior of $(\theta_t, X_t)$ itself is still an open problem under the SGD iteration (7). In this part, we propose Lemma E.1 that studies the special case of the SGD iteration [9] (which studied the diminishing step size) and complements the CLT result for constant step size with time-averaged iterates.

We consider a quadratic objective function

$$f(\theta) = \frac{1}{n}\sum_{i=1}^{n}\left(\frac{1}{2}\theta^T\mathbf{A}\theta - \theta^T\mathbf{b}(i)\right),$$

where matrix $\mathbf{A} \in \mathbb{R}^{d\times d}$ is positive definite and vector $\mathbf{b}(X) \in \mathbb{R}^d$ only depends on the state $X \in \mathcal{V}$ of the Markovian input. Then, the SGD iteration studied in [9] is given as

$$\theta_{t+1} = \theta_t - \gamma_{t+1}\left(\mathbf{A}\theta_t - \mathbf{b}(X_{t+1})\right). \tag{33}$$

Here, we study the constant step size $\gamma_t = \gamma$, $\forall t \geq 0$. Define $\bar{\mathbf{b}} \triangleq \sum_{i\in[n]}\mathbf{b}(X_i)\pi_i$. The minimizer is given by $\theta^* = \mathbf{A}^{-1}\bar{\mathbf{b}}$ such that $\nabla f(\theta^*) = 0$. Then, we have the following CLT result for the SGD update rule (33) with constant step size and Markovian input $\{X_t\}_{t\geq 0}$.

**Lemma E.1.** *Consider the update rule (33) with positive definite matrix $\mathbf{A}$ and constant step size $\gamma$ such that $0 < \gamma < 2/\|\mathbf{A}\|_2$. Then, for averaged iterates $\bar{\theta}_t = \frac{1}{t}\sum_{i=0}^{t-1}\theta_i$, we have*

$$\bar{\theta}_t \xrightarrow[t\to\infty]{a.s.} \theta^*, \quad and \quad \sqrt{t}(\bar{\theta}_t - \theta^*) \xrightarrow[t\to\infty]{Dist} \mathcal{N}(0, \mathbf{V}_X), \tag{34}$$

*where $\mathbf{V}_X = \mathbf{A}^{-1}\mathbf{\Sigma}_X(\mathbf{A}^{-1})^T$ and $\mathbf{\Sigma}_X = \lim_{t\to\infty}\frac{1}{t}\mathbb{E}[B_tB_t^T]$, $B_t \triangleq \sum_{s=1}^{t}(\mathbf{b}(X_s) - \bar{\mathbf{b}})$.*

*Proof.* Let $\tilde{\theta}_t = \theta_t - \theta^*$ and recall $\theta^* = \mathbf{A}^{-1}\bar{\mathbf{b}}$, we can rewrite (33) as

$$\tilde{\theta}_{t+1} = \tilde{\theta}_t - \gamma(\mathbf{A}\tilde{\theta}_t - \mathbf{b}(X_{t+1}) + \bar{\mathbf{b}}). \tag{35}$$

Recursively solving (35) gives

$$\tilde{\theta}_t = (\mathbf{I} - \gamma\mathbf{A})^t\tilde{\theta}_0 - \gamma\sum_{i=1}^{t}(\mathbf{I} - \gamma\mathbf{A})^{t-i}(\mathbf{b}(X_i) - \bar{\mathbf{b}}). \tag{36}$$

For averaged iterates $\bar{\theta}_t = \frac{1}{t}\sum_{i=0}^{t-1}\theta_i$, (36) gives

$$
\begin{aligned}
\bar{\theta}_t - \theta^* &= \frac{1}{t}\sum_{i=0}^{t-1}\tilde{\theta}_t \\
&= \frac{1}{t}\sum_{i=0}^{t-1}(\mathbf{I} - \gamma\mathbf{A})^i\tilde{\theta}_0 - \frac{\gamma}{t}\sum_{i=1}^{t-1}\sum_{j=1}^{i}(\mathbf{I} - \gamma\mathbf{A})^{i-j}(\mathbf{b}(X_j) - \bar{\mathbf{b}}) \\
&= \frac{1}{t}\sum_{i=0}^{t-1}(\mathbf{I} - \gamma\mathbf{A})^i\tilde{\theta}_0 - \frac{\gamma}{t}\sum_{i=1}^{t-1}\left[\sum_{j=0}^{t-i-1}(\mathbf{I} - \gamma\mathbf{A})^j\right](\mathbf{b}(X_i) - \bar{\mathbf{b}}) \\
&= \frac{1}{t}(\gamma\mathbf{A})^{-1}(\mathbf{I} - (\mathbf{I} - \gamma\mathbf{A})^t)\tilde{\theta}_0 - \frac{\gamma}{t}\sum_{i=1}^{t-1}(\gamma\mathbf{A})^{-1}(\mathbf{I} - (\mathbf{I} - \gamma\mathbf{A})^{t-i})(\mathbf{b}(X_i) - \bar{\mathbf{b}}),
\end{aligned}
\tag{37}
$$

where the third equality comes from rearranging the summation order in the second term on the RHS. The fourth equality comes from the fact that $\sum_{i=0}^{t-1}(\mathbf{I} - \gamma\mathbf{A})^i = (\gamma\mathbf{A})^{-1}(\mathbf{I} - (\mathbf{I} - \gamma\mathbf{A})^t)$.

Next we want to show $\lim_{t\to\infty}(\mathbf{I} - \gamma\mathbf{A})^t = \mathbf{0}$. Since we assume $0 < \gamma < 2/\|\mathbf{A}\|_2$, we have $\|\mathbf{I} - \gamma\mathbf{A}\|_2 = \max_{i=1,2,\cdots,n}|1 - \gamma\lambda_i(\mathbf{A})| < 1$, where $\lambda_i(\mathbf{A}) > 0$ is the $i$-the eigenvalue of the positive definite matrix $\mathbf{A}$. Then, by submultiplicative property, $\|(\mathbf{I} - \gamma\mathbf{A})^t\|_2 \leq \|\mathbf{I} - \gamma\mathbf{A}\|_2^t$ such that $\lim_{t\to\infty}\|(\mathbf{I} - \gamma\mathbf{A})^t\|_2 \leq \lim_{t\to\infty}\|\mathbf{I} - \gamma\mathbf{A}\|_2^t = 0$, which implies that $\lim_{t\to\infty}(\mathbf{I} - \gamma\mathbf{A})^t = \mathbf{0}$.

Now we want to show $\lim_{t\to\infty} \|\sum_{i=1}^{t-1}(\mathbf{I}-\gamma\mathbf{A})^{t-i}(\mathbf{b}(X_i)-\bar{\mathbf{b}})\|_2 < \infty$. Since vector-valued function $\mathbf{b}(\cdot)$ is defined on the finite state space $\mathcal{V}$, it is safe to assume $\|\mathbf{b}(X)-\bar{\mathbf{b}}\|_2 \leq C$ for some constant $C$. Then,

$$
\begin{aligned}
\lim_{t\to\infty} \left\|\sum_{i=1}^{t-1}(\mathbf{I}-\gamma\mathbf{A})^{t-i}(\mathbf{b}(X_i)-\bar{\mathbf{b}})\right\|_2 &\leq \lim_{t\to\infty}\sum_{i=1}^{t-1}\|(\mathbf{I}-\gamma\mathbf{A})^{t-i}\|_2\|(\mathbf{b}(X_i)-\bar{\mathbf{b}})\|_2 \\
&\leq C\lim_{t\to\infty}\sum_{i=1}^{t-1}\|\mathbf{I}-\gamma\mathbf{A}\|_2^{t-i} \\
&= C\lim_{t\to\infty}\sum_{i=1}^{t-1}\|\mathbf{I}-\gamma\mathbf{A}\|_2^{i} < \infty,
\end{aligned}
\tag{38}
$$

where the first inequality comes from submultiplicative property and triangular inequality, the first equality is by rewriting the index inside the summation, and the third inequality comes from the fact that $\|\mathbf{I}-\gamma\mathbf{A}\|_2 < 1$. Then, we have

$$
\lim_{t\to\infty}\frac{1}{t}\sum_{i=1}^{t-1}(\mathbf{I}-\gamma\mathbf{A})^{t-i}(\mathbf{b}(X_i)-\bar{\mathbf{b}}) = \mathbf{0},
\tag{39}
$$

and

$$
\lim_{t\to\infty}\frac{1}{\sqrt{t}}\sum_{i=1}^{t-1}(\mathbf{I}-\gamma\mathbf{A})^{t-i}(\mathbf{b}(X_i)-\bar{\mathbf{b}}) = \mathbf{0}.
\tag{40}
$$

With $\lim_{t\to\infty}(\mathbf{I}-\gamma\mathbf{A})^t = \mathbf{0}$ and (39), we have from (37) that

$$
\lim_{t\to\infty}\bar{\theta}_t - \theta^* = \lim_{t\to\infty} -\frac{\gamma}{t}\sum_{i=1}^{t-1}(\gamma\mathbf{A})^{-1}(\mathbf{b}(X_i)-\bar{\mathbf{b}}) = -\mathbf{A}^{-1}\lim_{t\to\infty}\frac{1}{t}\sum_{i=1}^{t-1}(\mathbf{b}(X_i)-\bar{\mathbf{b}}).
\tag{41}
$$

From the ergodic theorem for Markov chains ([7] Theorem 3.3.2), we have $\lim_{t\to\infty}\frac{1}{t}\sum_{i=1}^{t-1}(\mathbf{b}(X_i)-\bar{\mathbf{b}}) = \mathbf{0}$ and therefore $\lim_{t\to\infty}\bar{\theta}_t = \theta^*$.

To get the CLT result in (34), we first scale $\bar{\theta}_t - \theta^*$ from (37), along with (40), such that

$$
\begin{aligned}
\lim_{t\to\infty}\sqrt{t}(\bar{\theta}_t - \theta^*) &= \lim_{t\to\infty} -\frac{\gamma}{\sqrt{t}}\sum_{i=1}^{t-1}(\gamma\mathbf{A})^{-1}(\mathbf{b}(X_i)-\bar{\mathbf{b}}) \\
&= -\mathbf{A}^{-1}\lim_{t\to\infty}\frac{\sqrt{t-1}}{\sqrt{t}}\left(\frac{1}{\sqrt{t-1}}\sum_{i=1}^{t-1}(\mathbf{b}(X_i)-\bar{\mathbf{b}})\right).
\end{aligned}
\tag{42}
$$

From the CLT of Markov chain in Theorem 2.1, we know $\frac{1}{\sqrt{t}}\sum_{i=1}^{t}(\mathbf{b}(X_i)-\bar{\mathbf{b}}) \xrightarrow[t\to\infty]{dist} \mathcal{N}(0,\boldsymbol{\Sigma}_X)$, where $\boldsymbol{\Sigma}_X = \lim_{t\to\infty}\frac{1}{t}\mathbb{E}[(\sum_{s=1}^{t}(b(X_i)-\bar{b}))(\sum_{s=1}^{t}(b(X_i)-\bar{b}))^T]$. This result shows that time-averaged iterate $\bar{\theta}_t$ will guarantee the convergence to the exact solution and we have CLT result for $\sqrt{t}(\bar{\theta}_t - \theta^*)$ too.

Finally, we need to quantify the covariance matrix in the CLT result to $\sqrt{t}(\bar{\theta}_t - \theta^*)$. We will look at $\lim_{t\to\infty} t\mathbb{E}[(\bar{\theta}_t - \theta^*)(\bar{\theta}_t - \theta^*)^T]$. Note that the second term in (37) is bounded (see (38) for the proof) such that the cross term in the outer-product of $\bar{\theta}_t - \theta^*$ will vanish when $t\to\infty$. Then, we have

$$
\begin{aligned}
&\lim_{t\to\infty} t(\bar{\theta}_t - \theta^*)(\bar{\theta}_t - \theta^*)^T \\
&= \lim_{t\to\infty}\frac{t-1}{t}\left(-\frac{1}{\sqrt{t-1}}\mathbf{A}^{-1}\sum_{i=1}^{t-1}(\mathbf{b}(X_i)-\bar{\mathbf{b}})\right)\left(-\frac{1}{\sqrt{t-1}}\mathbf{A}^{-1}\sum_{i=1}^{t-1}(\mathbf{b}(X_i)-\bar{\mathbf{b}})\right)^T \\
&= \mathbf{A}^{-1}\lim_{t\to\infty}\frac{1}{t}\left(\sum_{i=1}^{t-1}(\mathbf{b}(X_i)-\bar{\mathbf{b}})\right)\left(\sum_{i=1}^{t-1}(\mathbf{b}(X_i)-\bar{\mathbf{b}})\right)^T(\mathbf{A}^{-1})^T.
\end{aligned}
\tag{43}
$$

Taking the expectation of (43) gives

$$\lim_{t\to\infty} t\mathbb{E}[(\bar{\theta}_t - \theta^*)(\bar{\theta}_t - \theta^*)^T]$$

$$=\mathbf{A}^{-1}\lim_{t\to\infty}\frac{t-1}{t}\mathbb{E}\left[\frac{1}{t-1}\left(\sum_{i=1}^{t-1}(\mathbf{b}(X_i)-\bar{\mathbf{b}})\right)\left(\sum_{i=1}^{t-1}(\mathbf{b}(X_i)-\bar{\mathbf{b}})\right)^T\right](\mathbf{A}^{-1})^T \quad (44)$$

$$=\mathbf{A}^{-1}\mathbf{\Sigma}_X(\mathbf{A}^{-1})^T.$$

Therefore, we have

$$\sqrt{t}(\bar{\theta}_t - \theta^*)\xrightarrow[t\to\infty]{dist}\mathcal{N}(0, \mathbf{A}^{-1}\mathbf{\Sigma}_X(\mathbf{A}^{-1})^T).$$

$\square$

In Lemma E.1, $\mathbf{A} = \nabla^2 f(\theta)$ and $\mathbf{\Sigma}_X$, by definition (8), is an asymptotic covariance matrix of the Markov chain $\{X_t\}_{t\geq 0}$ for vector-valued function $\mathbf{b}(\cdot)$. Therefore, (34) shares a similar form to (21) in Lemma 3.1. Our Theorem 3.6 can be carried over to Lemma E.1, which enables us to compare the efficiency ordering of SGD algorithms driven by different stochastic inputs under the update rule (33) and constant step size.

## F Proof of Proposition 4.1

[31, 19] proposed the guidance by modifying a reversible random walk into a non-Markovian random walk to achieve higher sampling efficiency and it was applied to other applications to improve sampling efficiency (e.g., [23, 25]). Specifically speaking, consider a reversible random walk $\{X_t\}_{t\geq 0}$ (e.g., SRW) with transition matrix $\mathbf{P}$ and stationary distribution $\pi$. Let its counterpart (e.g., NBRW) on the augmented state space be given by $\{Z_t\}_{t\geq 0} \triangleq \{(Y_{t-1}, Y_t)\}_{t\geq 0}$, where $Y_{t-1}, Y_t \in \mathcal{V}$ and $Z_0 = (Y_0, Y_0)$. Additionally, $\{Z_t\}_{t\geq 0}$ is a Markov chain on the augmented state space

$$\mathcal{E} \triangleq \{(i,j) : i,j \in \mathcal{V} \ s.t. \ P(i,j) > 0\} \subseteq \mathcal{V} \times \mathcal{V}$$

with stationary distribution $\pi'$. For notation simplicity, we use $e_{ij}$ to represent edge $(i,j)$. Note that by definition $e_{ij} \neq e_{ji}$ and we allow $i = j$ if $P(i,j) > 0$, which is a bit different from the edge set that does not include edge $(i,i)$. As proved in [19], the properties of NBRW $\{Z_t\}_{t\geq 0}$ are detailed in the following theorem.

**Theorem F.1** ([31] Theorem 2). *Suppose that $\{X_t\}$ is an irreducible, reversible Markov chain on the state space $\mathcal{V} = \{1, 2, \cdots, n\}$ with transition matrix $\mathbf{P} = \{P(i,j)\}$ and stationary distribution $\pi$. Construct a Markov chain $\{Z_t\}$ on the augmented state space $\mathcal{E}$ with transition matrix $\mathbf{P}' = \{P'(e_{ij}, e_{lk})\}$ in which the transition probabilities $P'(e_{ij}, e_{lk})$ satisfy the following two conditions: for all $e_{ij}, e_{ji}, e_{jk}, e_{kj} \in \mathcal{E}$ with $i \neq k$,*

$$P(j,i)P'(e_{ij}, e_{jk}) = P(j,k)P'(e_{kj}, e_{ji}), \quad (45a)$$

$$P'(e_{ij}, e_{jk}) \geq P(j,k). \quad (45b)$$

*Then, the Markov chain $\{Z_t\}_{t\geq 0}$ is irreducible and non-reversible with a unique stationary distribution $\pi'$ in which*

$$\pi'(e_{ij}) = \pi_i P(i,j) = \pi_j P(j,i), \ \ e_{ij} \in \mathcal{E}. \quad (46)$$

*Also, for any scalar-valued function $g$, the asymptotic variance $\sigma_Z^2(g) \leq \sigma_X^2(g)$.*

Now, we show how the non-Markov random walk with properties in Theorem F.1 can be included in Lemma 3.1. For the original function $G : \mathbb{R}^d \times \mathcal{V} \to \mathbb{R}^d$, we define another function $\Phi : \mathbb{R}^d \times \mathcal{E} \to \mathbb{R}^d$ such that $\Phi(\theta, e_{ij}) = G(\theta, j)$. Then, the SGD update rule (7) becomes $\theta_{t+1} = Proj_\Theta(\theta_t - \gamma_{t+1}\nabla\Phi(\theta_t, Z_{t+1}))$. From (46), we have for any $\theta \in \mathbb{R}^d$,

$$\phi(\theta) \triangleq \mathbb{E}_{Z\sim\pi'}\Phi(\theta, Z) = \sum_{e_{ij}\in\mathcal{E}}\Phi(\theta, e_{ij})\pi'(e_{ij}) = \sum_{i,j\in\mathcal{V}}G(\theta, j)\pi_j P(j,i) = \sum_{j\in\mathcal{V}}\frac{1}{n\pi_j}F(\theta, j)\pi_j = f(\theta),$$

$$(47)$$

showing the mean-field function $\phi(\theta)$ for $\Phi(\theta, Z)$ is the same as the objective function $f(\theta)$.

Next, we show assumptions (A1)-(A5) of Lemma 3.1 still hold for $\{Z_t\}_{t\geq 0}$ on the augmented state space $\mathcal{E}$ and function $\Phi$. Assumption (A1), (A5) and (A3) hold for function $\Phi(\theta, Z)$ because of our definition $\Phi(\theta, e_{ij}) = G(\theta, j)$. Assumption (A4) is satisfied because from Theorem F.1, $\{Z_t\}_{t\geq 0}$ is an irreducible and non-reversible Markov chain on the augmented state space $\mathcal{E}$ and there always exists a solution (27) to (22). Assumption (A2) holds because matrix $\mathbf{K} = \nabla^2 \phi(\theta^*) = \nabla^2 f(\theta^*)$ by (47). Therefore, we can say the Markov chain $\{Z_t\}_{t\geq 0}$ on the augmented state space $\mathcal{E}$, along with the newly defined function $\Phi$, can apply Lemma 3.1. The asymptotic covariance matrix $\boldsymbol{\Sigma}_Z \triangleq \boldsymbol{\Sigma}_Z(\nabla\Phi(\theta^*, \cdot))$ is given as

$$
\begin{aligned}
\boldsymbol{\Sigma}_Z &= \mathrm{Var}_{\boldsymbol{\pi}'}(\nabla\Phi(\theta^*, Z_0)) + \sum_{k\geq 1}\mathrm{Cov}_{\boldsymbol{\pi}'}(\nabla\Phi(\theta^*, Z_0), \nabla\Phi(\theta^*, Z_k))\mathrm{Cov}_{\boldsymbol{\pi}'}(\nabla\Phi(\theta^*, Z_0), \nabla\Phi(\theta^*, Z_k))^T \\
&= \lim_{t\to\infty}\frac{1}{t}\mathbb{E}\left\{\left[\sum_{s=1}^{t}(\nabla\Phi(\theta^*, Z_s) - \mathbb{E}_{\boldsymbol{\pi}'}(\nabla\Phi(\theta^*, \cdot)))\right]\left[\sum_{s=1}^{t}(\nabla\Phi(\theta^*, Z_s) - \mathbb{E}_{\boldsymbol{\pi}'}(\nabla\Phi(\theta^*, \cdot)))\right]^T\right\} \\
&= \lim_{t\to\infty}\frac{1}{t}\mathbb{E}\left\{\left[\sum_{s=1}^{t}(\nabla G(\theta^*, Y_s) - \mathbb{E}_{\boldsymbol{\pi}}(\nabla G(\theta^*, \cdot)))\right]\left[\sum_{s=1}^{t}(\nabla G(\theta^*, Y_s) - \mathbb{E}_{\boldsymbol{\pi}}(\nabla G(\theta^*, \cdot)))\right]^T\right\} \\
&= \boldsymbol{\Sigma}_Y(\nabla G(\theta^*, \cdot)),
\end{aligned}
\tag{48}
$$

where the third equality comes from (46) because

$$
\mathbb{E}_{\boldsymbol{\pi}'}[(\nabla\Phi(\theta^*, Z))] = \nabla f(\theta^*) = \sum_{j\in\mathcal{V}}\frac{1}{n\pi_j}\nabla F(\theta, j)\pi_j = \sum_{j\in\mathcal{V}}\pi_j\nabla G(\theta, j) = \mathbb{E}_{\boldsymbol{\pi}}(\nabla G(\theta^*, \cdot)).
$$

$\{Y_t\}_{t\geq 0}$ on the node space $\mathcal{V}$ is the trajectory generated by $\{Z_t\}_{t\geq 0}$ on the augmented state space $\mathcal{E}$. Let $\mathbf{V}_Z$ be the covariance matrix generated by the SGD algorithm driven by $\{Z_t\}_{t\geq 0}$. Denote $\boldsymbol{\Sigma}_X \triangleq \boldsymbol{\Sigma}_X(\nabla G(\theta^*, \cdot))$ the asymptotic covariance matrix and $\mathbf{V}_X$ the covariance matrix in (20) from the original Markov chain $\{X_t\}$. Then, from Theorem F.1 we know the asymptotic variances of NBRW and SRW are ordered for any scalar-valued function. Then, with Theorem 3.6 (i) we know that the asymptotic covariance matrices of NBRW and SRW are Loewner ordered for any vector-valued function such that $\boldsymbol{\Sigma}_Y(\nabla G(\theta^*, \cdot)) \leq_L \boldsymbol{\Sigma}_X(\nabla G(\theta^*, \cdot))$. From (48) we have $\boldsymbol{\Sigma}_Z \leq_L \boldsymbol{\Sigma}_X$. By applying Theorem 3.6 (ii), we have $\mathbf{V}_Z \leq_L \mathbf{V}_X$.

## G  Proof of Lemma 4.2

Assume we have a vector-valued function $\mathbf{g}: [n] \to \mathbb{R}^d$. For shuffling without replacement, which traverses every node in each epoch with length $n$, we group all the terms in each $k$-th epoch (shown in Figure 3) and analyze the term $\sum_{i=(k-1)n}^{kn-1}(\mathbf{g}(X_i) - \mathbb{E}_{\boldsymbol{\pi}}(\mathbf{g})) < \infty$ for $k \in \mathbb{Z}_+$. Note that we have

$$
\sum_{i=(k-1)n}^{kn-1}\mathbf{g}(X_i) = \sum_{j=1}^{n}\mathbf{g}(j)
\tag{49}
$$

by definition of shuffling without replacement. By (49) and $\mathbb{E}_{\boldsymbol{\pi}}(\mathbf{g}) = \frac{1}{n}\sum_{i=1}^{n}\mathbf{g}(i)$, we have

$$
\sum_{i=(k-1)n}^{kn-1}(\mathbf{g}(X_i) - \mathbb{E}_{\boldsymbol{\pi}}(\mathbf{g})) = \sum_{j=1}^{n}\mathbf{g}(j) - n\sum_{i=1}^{n}\frac{1}{n}\mathbf{g}(i) = 0.
\tag{50}
$$

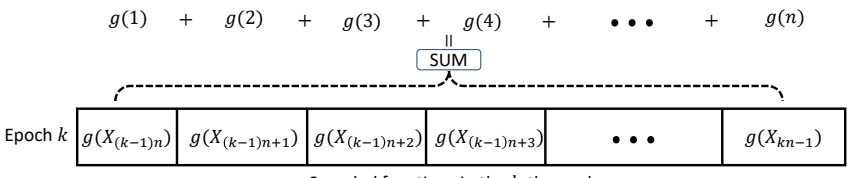

Figure 3: Diagram of the sampled functions in each epoch.

With the definition of the asymptotic covariance matrix, we have

$$
\begin{aligned}
\mathbf{\Sigma}_X(\mathbf{g}) &= \lim_{t \to \infty} \frac{1}{t} \mathbb{E} \left\{ \left( \sum_{i=1}^{t} (\mathbf{g}(X_s) - \mathbb{E}_{\boldsymbol{\pi}}(\mathbf{g})) \right) \left( \sum_{i=1}^{t} (\mathbf{g}(X_s) - \mathbb{E}_{\boldsymbol{\pi}}(\mathbf{g})) \right)^T \right\} \\
&= \lim_{t \to \infty} \frac{1}{t} \mathbb{E} \left\{ \left[ \sum_{i=1}^{s} (\mathbf{g}(X_i) - \mathbb{E}_{\boldsymbol{\pi}}(\mathbf{g})) + \sum_{i=s+1}^{t} (\mathbf{g}(X_i) - \mathbb{E}_{\boldsymbol{\pi}}(\mathbf{g})) \right] \right. \\
&\qquad \left. \cdot \left[ \sum_{i=1}^{s} (\mathbf{g}(X_i) - \mathbb{E}_{\boldsymbol{\pi}}(\mathbf{g})) + \sum_{i=s+1}^{t} (\mathbf{g}(X_i) - \mathbb{E}_{\boldsymbol{\pi}}(\mathbf{g})) \right]^T \right\} \\
&= \lim_{t \to \infty} \frac{1}{t} \mathbb{E} \left\{ \left[ \sum_{i=s+1}^{t} (\mathbf{g}(X_i) - \mathbb{E}_{\boldsymbol{\pi}}(\mathbf{g})) \right] \left[ \sum_{i=s+1}^{t} (\mathbf{g}(X_i) - \mathbb{E}_{\boldsymbol{\pi}}(\mathbf{g})) \right]^T \right\},
\end{aligned}
\tag{51}
$$

where $\boldsymbol{\pi}$ is the uniform stationary distribution, $s \triangleq t - (t \bmod n)$ is the time at which the previous epoch ended before time $t$ and we let $\sum_{i=t+1}^{t} (\mathbf{g}(X_i) - \mathbb{E}_{\boldsymbol{\pi}}(\mathbf{g})) = 0$ by default. The third equality in (51) comes from (50). Note that there always exists a constant $D$ such that $\|\mathbf{g}(i) - \mathbb{E}_{\boldsymbol{\pi}}\mathbf{g}\|_2 < D$ for any $i \in [n]$ because of the boundedness of function $\mathbf{g}$. Then, we have for any $t$,

$$
\left\| \sum_{i=s+1}^{t} (\mathbf{g}(X_i) - \mathbb{E}_{\boldsymbol{\pi}}(\mathbf{g})) \right\|_2 \leq \sum_{i=s+1}^{t} \|\mathbf{g}(X_i) - \mathbb{E}_{\boldsymbol{\pi}}(\mathbf{g})\|_2 < (t - s - 1)D < nD < \infty,
$$

where the second last inequality holds since $t - s - 1 = (t \bmod n) - 1 < n$. Back to (51), we have

$$
\|\mathbf{\Sigma}_X(\mathbf{g})\|_2 \leq \lim_{t \to \infty} \frac{1}{t} \mathbb{E} \left[ \left\| \sum_{i=s+1}^{t} (\mathbf{g}(X_i) - \mathbb{E}_{\boldsymbol{\pi}}(\mathbf{g})) \right\|_2^2 \right] \leq \lim_{t \to \infty} \frac{nD}{t} = 0,
$$

where the first inequality comes from Jensen's inequality. Finally, we have $\mathbf{\Sigma}_X(\mathbf{g}) = \mathbf{0}$ such that the asymptotic covariance matrices for both random and single shuffling are zero for any vector-valued function $\mathbf{g}$.

## H  Proof of Proposition 4.3 and Extension to Mini-batch Gradient Descent

### H.1  Single Shuffling in SGD CLT Analysis

Single shuffling is seen as a time-homogeneous, irreducible, periodic Markov chain and we know (A4) is the only requirement for Markov chain in the CLT result. As mentioned in [15] and [28] Chapter 17, the necessary condition to ensure the existence of function $\tilde{F}$ as in (27) is that the inverse $(\mathbf{I} - \mathbf{P} + \mathbf{1}\boldsymbol{\pi}^T)^{-1}$ exists. This is true for periodic Markov chain, and is shown in the following lemma.

**Lemma H.1.** *The solution* (27) *to the Poisson equation* (22) *exists for an underlying finite, irreducible periodic Markov chain with transition matrix* $\mathbf{P} \in \mathbb{R}^{m \times m}$, *stationary distribution* $\boldsymbol{\pi}$ *and period* $n \leq m$.

*Proof.* From Perron–Frobenius theorem for irreducible, non-negative stochastic matrices [34], we know there are $n$ complex eigenvalues uniformly distributed on the unit circle, including the *unique*

eigenvalue with value 1. Other $m - n$ eigenvalues fall inside the unit circle, but still uniformly distributed on some circles with absolute value strictly smaller than 1 because transition matrix $\mathbf{P}$ is similar to $e^{i\omega}\mathbf{P}$ where $i = \sqrt{-1}$ and $\omega = 2\pi/n$.

Denote $\lambda_1, \lambda_2, \cdots, \lambda_m = 1$ the eigenvalues of the transition matrix $\mathbf{P}$ and let $\mathbf{J}$ be the Jordan norm form. There exists an invertible matrix $\mathbf{Q} = [\mathbf{u}_1, \mathbf{u}_2, \cdots, \mathbf{u}_m]^T$ and $\mathbf{Q}^{-1} = [\mathbf{v}_1, \mathbf{v}_2, \cdots, \mathbf{v}_m]$ such that $\mathbf{P} = \mathbf{Q}\mathbf{J}\mathbf{Q}^{-1}$ and $\mathbf{u}_i^T \mathbf{v}_i = 1$ for all $i \in [m]$. In particular, $\mathbf{u}_m = \boldsymbol{\pi}$ and $\mathbf{v}_m = \mathbf{1}$. Now, $(\lambda_m = 1, \mathbf{u}_m, \mathbf{v}_m)$ is also the PF eigenpair for the matrix $\boldsymbol{\Pi}$, which enables us to write down $\mathbf{1}\boldsymbol{\pi}^T = \mathbf{Q}\boldsymbol{\Lambda}\mathbf{Q}^{-1}$, where $\boldsymbol{\Lambda} = \text{diag}(0, 0, \cdots, 1)$. Therefore, $\mathbf{I} - \mathbf{P} + \mathbf{1}\boldsymbol{\pi}^T = \mathbf{Q}(\mathbf{I} - \mathbf{J} + \boldsymbol{\Lambda})\mathbf{Q}^{-1}$. Note that $\mathbf{I} - \mathbf{J} + \boldsymbol{\Lambda}$ is a new Jordan norm form with **non-zero** entries on the main diagonal. Assume $\mathbf{J}_i = \lambda_i \mathbf{I} + \mathbf{N}$ is one of its Jordan block with nilpotent matrix $\mathbf{N}$ such that $\mathbf{N}^{p_i} = 0$ for some $p_i \geq 2$, then

$$\mathbf{J}_i^{-1} = \lambda_i^{-1}(\mathbf{I} + \lambda_i^{-1}\mathbf{N})^{-1} = \lambda_i^{-1}(\mathbf{I} - \lambda_i^{-1}\mathbf{N} + \cdots + (\lambda_i)^{-p_i+1}\mathbf{N}^{p_i-1}),$$

showing that $\mathbf{J}_i^{-1}$ exists for all Jordan blocks in the new Jordan norm form $\mathbf{I} - \mathbf{J} + \boldsymbol{\Lambda}$ because all $\lambda_i$ are non-zero. Therefore, $(\mathbf{I} - \mathbf{P} + \mathbf{1}\boldsymbol{\pi}^T)^{-1}$ exists and (27) holds. $\qquad\square$

With Lemma H.1, single shuffling can indeed be included in the SGD CLT result, and its covariance matrix is the zero matrix $\mathbf{0}$ (from Lemma 4.2). Random shuffling is a *time-inhomogeneous* Markov chain due to its nature of reshuffling at the beginning of each epoch. Before providing our main proof, we first present the augmentation of the random shuffling sequence which transforms it into a *time-homogeneous* periodic Markov chain on the augmented state space.

## H.2 Augmentation of Random Shuffling for CLT Analysis

By the definition of random shuffling, in each epoch of length $n$, the sampler traverses one permutation sequence drawn uniformly at random from the permutation sequence set with size $n!$. Due to its random nature across each epoch, random shuffling is not a Markov chain on state space $[n]$. In order to include random shuffling in the SGD CLT result, Lemma 3.1, we need to transform it into a Markov chain on a augmented state space.

Let $\{X_t\}_{t\geq 0}$ be the sequence generated by random shuffling. We first define an augmented state space $\mathcal{S}$, where for each state $s_t \triangleq \{\{A_j^{(t)}\}_{j\in[n]}, c_t\} \in \mathcal{S}$, the sequence $\{A_j^{(t)}\}_{j\in[n]} \triangleq \{X_{t-n+1}, X_{t-n+2}, \cdots, X_t\}$ is of length $n$, and records the history of past $n$ indices until time $t$. The integer $c_t \in \{1, 2, \cdots, n\}$ is the time spent in current epoch at time $t$. For examples, consider the state space to be $\mathcal{S} = \{1, 2, \cdots, 6\}$ in total and assume the sequence of visited states until $t = 8$ is $\{3, 6, 2, 1, 5, 4, 2, 5\}$. Here, $\{3, 6, 2, 1, 5, 4\}$ is one complete permutation sequence in the first epoch and $\{2, 5\}$ are in the second epoch. At time $t = 8$, the sampler is at index 5 and the sequence of past 6 indices is $\{2, 1, 5, 4, 2, 5\}$ and $c_t = 2$, such that $s_8 = \{\{2, 1, 5, 4, 2, 5\}, 2\}$. In the next iteration $t = 9$, the sequence will be $\{1, 5, 4, 2, 5, X\}$ and $c_t = 3$, where $X$ is the index that can be chosen from $\{1, 3, 4, 6\}$ uniformly at random because $\{2, 5\}$ have been chosen in the current epoch. Then, we have

$$s_9 = \begin{cases} \{\{1, 5, 4, 2, 5, 1\}, 3\} & \text{w.p } 1/4, \\ \{\{1, 5, 4, 2, 5, 3\}, 3\} & \text{w.p } 1/4, \\ \{\{1, 5, 4, 2, 5, 4\}, 3\} & \text{w.p } 1/4, \\ \{\{1, 5, 4, 2, 5, 6\}, 3\} & \text{w.p } 1/4. \end{cases} \tag{52}$$

Assume $s_{12} = \{2, 5, 6, 1, 3, 4, 6\}$ at $t = 12$, the next state $s_{13} = \{\{5, 6, 1, 3, 4, X\}, 1\}$ and $X$ is chosen from $\{1, 2, \cdots, 6\}$ uniformly at random.

Note that

- We only include *proper combination* of sequence $\{A_j\}_{j\in[n]}$ in the augmented state space, where 'proper' means the sequence is possible to appear with the current value of $c_t$. For instance, $\{\{2, 1, 5, 4, 2, 2\}, 2\}$ or $\{\{2, 1, 5, 1, 2, 5\}, 2\}$ is inproper because $\{2, 2\}$ or $\{2, 1, 5, 1\}$ doesn't exist in the permutation sequence in one epoch.

- Transition probability $P(s_t, s_{t+1})$ is possibly non-zero **only** when $c_{t+1} = c_t + 1$ for $c_t \leq n - 1$, or $c_{t+1} = 1$ when $c_t = n$.

Next, we show the proposition that will be used later to show that random shuffling can also be fitted into the CLT result.

**Proposition H.2.** $\{s_t\}_{t\geq 0}$ *forms a finite, irreducible and periodic Markov chain with period $n$.*

*Proof.* By our construction, the size of choice of $\{A_j\}_{j\in[n]}$ with $c = i$ is $(C_n^i)^2 i!(n-i)!$, because the first $i$ indices has $C_n^i i!$ choices and remaining sequence has $C_n^{n-i}(n-i)!$ choices. The size of the augmented state space is $\sum_{i=1}^n (C_n^i)^2 i!(n-i)!$ and is still *finite*.

The *irreducibility* can be shown by $P(s_{t+2n} = s'|s_t = s) > 0$ because we can always construct two permutation sequences in two epochs; one including first $i$ indices of $\{A'_j\}_{j\in[n]}$ in state $s'$ and the other including the remaining sequences.

For *periodicity*, if the sequence $\{A_j^{(t)}\}_{j\in[n]}$ in the current state $s_t = s \in \mathcal{S}$ includes repeated index at time $t$, e.g., index $i \in \{X_{t-n+1}, \cdots, X_m\}$ (in the $\frac{m}{n}$-th epoch) and $i \in \{X_{m+1}, \cdots, X_t\}$ (in the $(\frac{m}{n}+1)$-th epoch), then $i \notin \{X_{t+1}, \cdots, X_{m+n}\}$ due to the nature of shuffling without replacement in an epoch, which leads to $P(s_{t+n} = s|s_t = s) = 0$. In addition, for $k \geq 2$ we can always construct intermediate sequences $\{X_{t+1}, \cdots, X_{t+(k-1)n}\}$ such that $\{X_{t-n+1}, \cdots, X_m\} = \{X_{t+(k-1)n+1}, \cdots, X_{m+kn}\}$ and $\{X_{m+1}, \cdots, X_t\} = \{X_{m+kn+1}, \cdots, X_{t+kn}\}$, implying that $P(s_{t+kn} = s|s_t = s) > 0$ for $k \geq 2$. On the other hand, if $\{A_j^{(t)}\}_{j\in[n]}$ does not include repeated index, $P(s_{t+kn} = s|s_t = s) > 0$ for $k \in \mathbb{N}$. We also note that $P(s_{t+j} = s|s_t = s) = 0$ for $s \in \mathcal{S}$, $j \neq kn$ and $k \in \mathbb{N}$. Since $\{c_t\}$ by its definition is a periodic sequence $\{1, 2, \cdots, n, 1, 2, \cdots, n, 1, 2, \cdots\}$ with period of length $n$, we know that $c_t = c_{t+j}$ holds only when $j = kn$ for $k \in \mathbb{N}$. Then, for $j \neq kn$, we have $c_{t+j} \neq c_t$ such that $s_{t+j} \neq s_t$, which leads to $P(s_{t+j} = s|s_t = s) = 0$ for $j \neq kn$ and $k \in \mathbb{N}$. Therefore, by definition of periodicity, the Markov chain is of period $n$. $\square$

Together with Lemma H.1 and Proposition H.2, we can see random shuffling can also be include in the SGD CLT.

## H.3 Extension to Mini-batch Gradient Descent

Mini-batch gradient descent is another popular gradient descent variant and is widely used in the machine learning tools [10, 1, 32] to accelerate the learning process when compared to SGD. Instead of sampling a single element, mini-batch gradient descent samples multiple elements from $[n]$ in each iteration that form a batch.

To incorporate the notion of mini-batches in our SGD framework, we provide a reformulation of the general SGD iteration based on a similar formulation in [16] for the general analysis of SGD with *i.i.d* inputs. Consider a stochastic process $\{B_t\}_{t\geq 0}$ as the driving sequence, which randomly samples batches of size $S$ (without replacement) from the state space $[n]$, that is $B_t \subset [n]$ and $|B_t| = S$ for all $t \geq 0$. Here we assume $[n] \bmod S = 0$ for simplicity. $B_t$ will therefore refer to the batch chosen at any time $t > 0$. We assume that $B_t$ for all $t > 0$ are *i.i.d* random variables drawn from a distribution $\mathcal{P}$, such that $\mathcal{P}(B) > 0$ is the probability with which a batch $B \subset [n]$ is picked. We associate with any batch $B$, $\mathbf{v}(B) \triangleq \left[\sum_{i\in B} \mathbf{e}_i\right] / \binom{N}{S}\mathcal{P}(B)$, where $\mathbf{e}_i$ is the $i$'th vector of the canonical basis of $\mathbb{R}^d$. We then denote $\mathbf{F}(\theta) \triangleq [F(\theta, 1), \cdots, F(\theta, n)]^T$, and $\nabla\mathbf{F}(\theta) \triangleq [\nabla F(\theta, 1), \cdots, \nabla F(\theta, n)]^T$ for all $\theta \in \Theta$. With this notation, we can rewrite the general update rule for mini-batch SGD as

$$\theta_{t+1} = \text{Proj}_\Theta \left(\theta_t - \gamma_{t+1} \nabla\mathbf{F}(\theta_t)^T \mathbf{v}(B_{t+1})\right). \tag{53}$$

Note that this way of defining the mini-batch based random input ensures that $\mathbb{E}_\mathcal{P}[\mathbf{F}(\theta)^T \mathbf{v}(\cdot)] = f(\theta)$ for all $\theta \in \Theta$, maintaining the same objective function irrespective of the distribution from which batches are sampled.

With $X_t = B_t$ for all $t \geq 0$, and $\nabla G(\theta_t, X_{t+1}) = \nabla\mathbf{F}(\theta_t)^T \mathbf{v}(B_{t+1})$, the iteration (53) can still be written in the form of (7) with *i.i.d* input sequence $\{X_t\}_{t\geq 0}$. We can thus apply the CLT for SGD algorithms to the mini-batch SGD with *i.i.d* input, and in a similar fashion as (13) derive the explicit form of the asymptotic covariance matrix of (53), that is,

$$\mathbf{\Sigma}_B(\nabla\mathbf{F}(\theta^*)^T \mathbf{v}(\cdot)) \triangleq \text{Var}_{B_0\sim\mathcal{P}}(\nabla\mathbf{F}(\theta^*)^T \mathbf{v}(B_0)). \tag{54}$$

In practice, mini-batch gradient descent with shuffling is more widely used than $i.i.d$ sampling [1], in which $B_t$ is generated by shuffling-based method instead of independent drawn from a distribution.[3] At the beginning of each epoch, *Mini-batch gradient descent with random shuffling* shuffles the whole dataset $[n]$ and split it into small batches. On the other hand, *mini-batch gradient descent with single shuffling* only shuffles the dataset $[n]$ once before dividing it into batches, sticking to a predetermined sequence of batches for all epochs of the training process. As pointed out by [36], there is still a gap between practical implementation and theoretical analysis for mini-batch gradient descent with shuffling. Nevertheless, by extrapolating the analysis from Proposition 4.3, we are able to analyze the efficiency ordering of shuffling and *i.i.d* sampling in the mini-batch version, as stated next.

**Proposition H.3.** *Consider the mini-batch gradient descent* (53) *with stochastic inputs single/random shuffling* $\{X_t\}_{t\geq 0}$ *and i.i.d sampling* $\{Y_t\}_{t\geq 0}$, *we have* $\theta_t^X, \theta_t^Y \xrightarrow[t\to\infty]{a.s.} \theta^*$ *and* $\mathbf{V}_X = \mathbf{0} \leq_L \mathbf{V}_Y$.

*Proof.* Let $l \triangleq n/B \in \mathbb{N}$. We first give the following corollary.

**Corollary H.4.** *Mini-batch gradient descent with single shuffling is an irreducible, periodic Markov chain with period* $l$.

*Proof.* For single shuffling version, we divide the whole dataset $[n]$ into $B(1), B(2), \cdots, B(l)$ and the corresponding sampling vector will be $\mathbf{v}(1), \mathbf{v}(2), \cdots, \mathbf{v}(l)$. We shuffle the indices once, denoted by $a(1), a(2), \cdots, a(l)$, and stick to this sequence all the time. Then, in each epoch, sampler will update $\theta_t$ according to the sequence $\mathbf{v}(a(1)), \mathbf{v}(a(2)), \cdots, \mathbf{v}(a(l))$, where $\{\mathbf{v}_t\}_{t\geq 0}$ forms the finite, irreducible periodic Markov chain with period $l$. $\square$

Together with Corollary H.4 and Lemma H.1, mini-batch gradient descent with single shuffling can be included in the SGD CLT analysis and Theorem 3.6. Then, by Lemma H.1 and 4.2, mini-batch SGD with single shuffling can be applied to CLT result, which gives the asymptotic covariance matrix $\mathbf{\Sigma_w}(\nabla \mathbf{F}(\theta^*)^T \mathbf{v}(\cdot)) = \mathbf{0}$ and thus the covariance matrix in the CLT result is also zero.

For random shuffling version, we can use similar method as in Appendix H.2 to augment the state space, which forms the corollary as follows.

**Corollary H.5.** $\{x_t\}_{t\geq 0}$ *forms a finite, irreducible and periodic Markov chain with period* $l$.

*Proof.* Let $\mathcal{X}$ be the augmented space, where state $x_t \triangleq \{\{W_j^{(t)}\}_{j\in[l]}, c_t\}$. Sequence $\{W_j^{(t)}\}_{j\in[l]} = \{\mathbf{v}_{t-l+1}, \mathbf{v}_{t-l+2}, \cdots, \mathbf{v}_t\}$ records the last $l$ selected batches and $c_t \in \{1, 2, \cdots, l\}$ is the relative position of the batch in the current epoch at time $t$. The only difference for mini-batch version to the single element version is that we sample one batch of size $B$ without replacement according to the indices yet to be chosen in the current epoch. Similar to the proof in Proposition H.2, $\{X_t\}_{t\geq 0}$ is also a finite, irreducible and periodic Markov chain with period $l$. $\square$

Corollary H.5 and Lemma H.1 show that mini-batch gradient descent with random shuffling can also be included in the SGD CLT analysis. However, we still need to check the form of asymptotic covariance matrix due to the augmentation. We follow the same idea from Appendix F and define a function $\Phi(\theta, x_t) \triangleq \mathbf{F}(\theta)^T \mathbf{v}(B_t)$. Then, from Lemma 4.2, asymptotic covariance matrix $\mathbf{\Sigma}_x$ is

---

[3]The reformulation (53) enables us to analyze mini-batch gradient descent with various stochastic processes that samples $B_t$, not just i.i.d input and shuffling. However, discussing general processes $\{B_t\}_{t\geq 0}$ is beyond the scope of this paper.

given as

$$\boldsymbol{\Sigma}_x = \text{Var}_{\boldsymbol{\pi}}(\nabla\Phi(\theta^*, x_0)) + \sum_{k\geq 1}\text{Cov}_{\boldsymbol{\pi}}(\nabla\Phi(\theta^*, x_0), \nabla\Phi(\theta^*, x_k))\text{Cov}_{\boldsymbol{\pi}}(\nabla\Phi(\theta^*, x_0), \nabla\Phi(\theta^*, x_k))^T$$

$$= \lim_{t\to\infty}\frac{1}{t}\mathbb{E}\left\{\left[\sum_{s=1}^{t}(\nabla\Phi(\theta^*, x_s) - \mathbb{E}_{\boldsymbol{\pi}}(\nabla\Phi(\theta^*, \cdot)))\right]\left[\sum_{s=1}^{t}(\nabla\Phi(\theta^*, x_s) - \mathbb{E}_{\boldsymbol{\pi}}(\nabla\Phi(\theta^*, \cdot)))\right]^T\right\}$$

$$= \lim_{t\to\infty}\frac{1}{t}\mathbb{E}\left\{\left[\sum_{s=1}^{t}(\nabla\mathbf{F}(\theta^*)^T\mathbf{v}(B_s) - \nabla f(\theta^*))\right]\left[\sum_{s=1}^{t}(\nabla\mathbf{F}(\theta^*)^T\mathbf{v}(B_s) - \nabla f(\theta^*))\right]^T\right\}$$

$$= \boldsymbol{\Sigma}_x(\nabla\mathbf{F}(\theta^*)^T\mathbf{v}(\cdot)) = \mathbf{0}, \tag{55}$$

where the third equality comes from the limiting distribution of random shuffling that is uniform. Thus, the covariance matrix of random shuffling in the CLT result is also zero.

Above results show that both single shuffling and random shuffling in mini-batch SGD have higher efficiency than mini-batch SGD with i.i.d sampling. $\qquad\square$

Proposition H.3 generalizes Proposition 4.3 (special case with mini-batch of size $S = 1$) in that the same efficiency ordering between shuffling and *i.i.d* input holds true even with mini-batches.

# I  Simulation

In Appendix I.1, we give the details of our simulation setup for Figure 1, involving three reversible Markov chains - the Metropolis-Hasting random walk (MHRW), a modification of MHRW (Modified-MHRW) and fastest mixing Markov chain (FMMC), each having the uniform distribution as their stationary measure. In Appendix I.2, we expand upon the numerical results in Section 5 by including additional results for large graphs.

## I.1  Details behind Figure 1

For the random walk SGD (RWSGD) simulation in Figure 1, we consider the problem of minimizing a (scalar-valued) quadratic objective function

$$f(\theta) \triangleq \frac{1}{n}\sum_{i=1}^{n}F(\theta, i) = \frac{1}{2n}\sum_{i=1}^{n}(\theta - b(i))^2, \tag{56}$$

where $\theta,\ b(i) \in \mathbb{R}$ for $i = 1, 2, \cdots, n$ and $n$ is the number of nodes on the graph. The minimizer is given by $\theta^* \triangleq \arg\min_\theta f(\theta) = \frac{1}{n}\sum_{i=1}^{n}b(i)$. The RWSGD iteration for the objective function (56) is then given by

$$\theta_{t+1} = \theta_t - \gamma_{t+1}(\theta_t - b(X_{t+1})), \tag{57}$$

where we choose $\gamma_t = 1/t^{0.9}$ and $\{X_t\}_{t\geq 0}$ is the stochastic input, e.g., MHRW, Modified-MHRW, and FMMC.

In Figure 1, we simulate the SGD algorithm on two graphs; one is an 8-node graph $\mathcal{G}_1$ and the other is a 5-node graph $\mathcal{G}_2$. The two graphs are arbitrarily constructed while ensuring connectivity. See Figure 4 for resulting topologies.

Now, we are ready to introduce the construction of three Markov chains on two graphs in Figure 4.

**MHRW:** Metropolis-Hasting algorithm [27] shows that the transition matrix of MHRW is constructed in the following manner:

$$P(i, j) = \begin{cases} \min\left\{\frac{1}{d_i}, \frac{1}{d_i}\right\}, & j \in N(i), \\ 1 - \sum_{j\in N(i)}P(i, j), & j = i, \end{cases} \tag{58}$$

where $d_i$ is the degree of node $i$ and $N(i)$ is the set of node $i$'s neighbors.

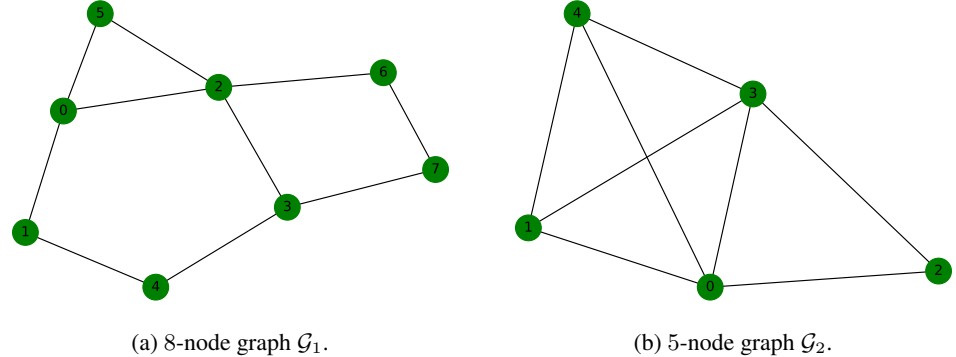

(a) 8-node graph $\mathcal{G}_1$.

(b) 5-node graph $\mathcal{G}_2$.

Figure 4: Topology of two graphs.

**Modified-MHRW:** To construct a 'modified-MHRW', which is more *efficient* than the standard MHRW,[4] we employ the notion of 'Peskun ordering', originated from [33].

**Definition I.1** (Peskun ordering [33])**.** For two finite, ergodic, reversible Markov chains $\{X_t\}_{t\geq 0}, \{Y_t\}_{t\geq 0}$ on the state space $\mathcal{V}$ with transition matrices $\mathbf{P}_X, \mathbf{P}_Y$ having the same stationary distribution $\boldsymbol{\pi}$, it is said that $\mathbf{P}_Y$ dominates $\mathbf{P}_X$ off the diagonal, written as $\mathbf{P}_X \preceq \mathbf{P}_Y$ if $P_X(i,j) \leq P_Y(i,j)$ for all $i, j \in \mathcal{V}$ and $i \neq j$.

We have the following lemma that connects the Peskun ordering to the efficiency ordering.

**Lemma I.2** ([33] Theorem 2.1.1)**.** *If $\mathbf{P}_X \preceq \mathbf{P}_Y$, then $\sigma_X^2(g) \geq \sigma_Y^2(g)$ for any scalar-valued function $g$ with $\mathbb{E}_{\boldsymbol{\pi}}(g^2) < \infty$, that is, $\{Y_t\}_{t\geq 0}$ is more efficient than $\{X_t\}_{t\geq 0}$.*

We can manually construct a more efficient Markov chain by reducing the self-transition probability $P(i, i)$ of the MHRW and redistributing to off-diagonal entries, whenever possible, in a way that each row still sums to one and the resulting matrix is doubly-stochastic (i.e., the resulting Markov chain is reversible w.r.t the uniform distribution). In view of Lemma I.2, this modification improves the efficiency (smaller AV $\sigma^2$ compared to the standard MHRW).[5]

**FMMC:** FMMC is obtained by solving a semidefinite programming (proposed in problem (6) of [6]), which gives a Markov chain that minimizes the SLEM of the transition matrix over the entire class of reversible Markov chains w.r.t the uniform stationary distribution for a given graph topology. This is done numerically by using the CVXOPT package [12]. Later we will show in the simulation that FMMC indeed has the smallest SLEM compared to MHRW and Modified-MHRW.

In what follows, we index these three Markov chains with numbers in the subscript: MHRW (indexed by 1), Modified-MHRW (indexed by 2), and FMMC (indexed by 3). For graph $\mathcal{G}_1$, the transition

---

[4]*Efficiency ordering* of Markov chains is introduced in Definition 3.5. In short, a Markov chain $\{X_t\}_{t\geq 0}$ is more efficient than $\{Y_t\}_{t\geq 0}$ if the asymptotic variances (AV) satisfy $\sigma_X^2(g) \leq \sigma_Y^2(g)$ for any scalar-valued function $g$ with $\mathbb{E}_{\boldsymbol{\pi}}(g^2) < \infty$, where $\sigma_X^2(g)$ is defined in (4).

[5]Note that there can be many ways to modify the standard MHRW that make the Markov chain more efficient. The pursuit of the 'optimal' modification w.r.t the efficiency is out of the scope of this paper.

matrices of MHRW $\mathbf{P}_1^{\mathcal{G}_1}$, Modified-MHRW $\mathbf{P}_2^{\mathcal{G}_1}$, and FMMC $\mathbf{P}_3^{\mathcal{G}_1}$ are given by

$$\mathbf{P}_1^{\mathcal{G}_1} = \begin{bmatrix} 1/12 & 1/3 & 1/4 & 0 & 0 & 1/3 & 0 & 0 \\ 1/3 & 1/6 & 0 & 0 & 1/2 & 0 & 0 & 0 \\ 1/4 & 0 & 0 & 1/4 & 0 & 1/4 & 1/4 & 0 \\ 0 & 0 & 1/4 & 1/12 & 1/3 & 0 & 0 & 1/3 \\ 0 & 1/2 & 0 & 1/3 & 1/6 & 0 & 0 & 0 \\ 1/3 & 0 & 1/4 & 0 & 0 & 5/12 & 0 & 0 \\ 0 & 0 & 1/4 & 0 & 0 & 0 & 1/4 & 1/2 \\ 0 & 0 & 0 & 1/3 & 0 & 0 & 1/2 & 1/6 \end{bmatrix},$$

$$\mathbf{P}_2^{\mathcal{G}_1} = \begin{bmatrix} 0 & 0.35 & 0.25 & 0 & 0 & 0.4 & 0 & 0 \\ 0.35 & 0.02 & 0 & 0 & 0.63 & 0 & 0 & 0 \\ 0.25 & 0 & 0 & 0.25 & 0 & 0.25 & 0.25 & 0 \\ 0 & 0 & 0.25 & 0 & 0.37 & 0 & 0 & 0.38 \\ 0 & 0.63 & 0 & 0.37 & 0 & 0 & 0 & 0 \\ 0.4 & 0 & 0.25 & 0 & 0 & 0.35 & 0 & 0 \\ 0 & 0 & 0.25 & 0 & 0 & 0 & 0.13 & 0.62 \\ 0 & 0 & 0 & 0.38 & 0 & 0 & 0.62 & 0 \end{bmatrix}, \tag{59}$$

$$\mathbf{P}_3^{\mathcal{G}_1} = \begin{bmatrix} 0.13 & 0.42 & 0.17 & 0 & 0 & 0.28 & 0 & 0 \\ 0.42 & 0.1 & 0 & 0 & 0.48 & 0 & 0 & 0 \\ 0.17 & 0 & 0 & 0.06 & 0 & 0.32 & 0.45 & 0 \\ 0 & 0 & 0.06 & 0.14 & 0.46 & 0 & 0 & 0.34 \\ 0 & 0.48 & 0 & 0.46 & 0.06 & 0 & 0 & 0 \\ 0.28 & 0 & 0.32 & 0 & 0 & 0.4 & 0 & 0 \\ 0 & 0 & 0.45 & 0 & 0 & 0 & 0.09 & 0.46 \\ 0 & 0 & 0 & 0.34 & 0 & 0 & 0.46 & 0.2 \end{bmatrix}.$$

For graph $\mathcal{G}_2$, the transition matrices of MHRW $\mathbf{P}_1^{\mathcal{G}_2}$, Modified-MHRW $\mathbf{P}_2^{\mathcal{G}_2}$, and FMMC $\mathbf{P}_3^{\mathcal{G}_2}$ are given by

$$\mathbf{P}_1^{\mathcal{G}_2} = \begin{bmatrix} 0 & 1/4 & 1/4 & 1/4 & 1/4 \\ 1/4 & 1/6 & 0 & 1/4 & 1/3 \\ 1/4 & 0 & 1/2 & 1/4 & 0 \\ 1/4 & 1/4 & 1/4 & 0 & 1/4 \\ 1/4 & 1/3 & 0 & 1/4 & 1/6 \end{bmatrix},$$

$$\mathbf{P}_2^{\mathcal{G}_2} = \begin{bmatrix} 0 & 0.25 & 0.25 & 0.25 & 0.25 \\ 0.25 & 0 & 0 & 0.25 & 0.5 \\ 0.25 & 0 & 0.5 & 0.25 & 0 \\ 0.25 & 0.25 & 0.25 & 0 & 0.25 \\ 0.25 & 0.5 & 0 & 0.25 & 0 \end{bmatrix}, \tag{60}$$

$$\mathbf{P}_3^{\mathcal{G}_2} = \begin{bmatrix} 0.09 & 0.25 & 0.33 & 0.08 & 0.25 \\ 0.25 & 0.25 & 0 & 0.25 & 0.25 \\ 0.33 & 0 & 0.34 & 0.33 & 0 \\ 0.08 & 0.25 & 0.33 & 0.09 & 0.25 \\ 0.25 & 0.25 & 0 & 0.25 & 0.25 \end{bmatrix}.$$

In both (59) and (60), observe that Modified-MHRW and MHRW follow the Peskun ordering, i.e., $\mathbf{P}_1^{\mathcal{G}_1} \preceq \mathbf{P}_2^{\mathcal{G}_1}$ and $\mathbf{P}_1^{\mathcal{G}_2} \preceq \mathbf{P}_2^{\mathcal{G}_2}$, such that Modified-MHRW is more efficient than MHRW according to Lemma I.2. In addition, the SLEMs of these matrices are given in Table 1, where FMMC has the smallest SLEM in both graphs compared to MHRW and Modified-MHRW. Interestingly, Modified-MHRW has larger SLEM than MHRW in graph $\mathcal{G}_1$, which means Modified-MHRW can mix slower than MHRW to the stationary distribution.

In Figure 5, we show the simulation result of each Markov chain in the RWSGD algorithm with iteration (57) w.r.t MSE $\mathbb{E}\|\theta_t - \theta^*\|_2^2$ in graph $\mathcal{G}_1$ and $\mathcal{G}_2$.[6] In both graphs, Modified-MHRW (green curve) performs better than MHRW (red curve) and FMMC (blue curve) with smallest MSE while it has the largest SLEM shown in Table 1. This implies that the order of SLEM does not reflect the order of MSE in the RWSGD algorithm.

---

[6]The reason we plot the same curves in each graph will be explained in the next paragraph.

| | $\mathcal{G}_1$ | $\mathcal{G}_2$ |
|---|---|---|
| MHRW ($\beta_1$) | 0.761 | 0.500 |
| Modified-MHRW ($\beta_2$) | 0.868 | 0.500 |
| FMMC ($\beta_3$) | 0.712 | 0.408 |

Table 1: SLEMs of the transition matrices in (59) and (60).

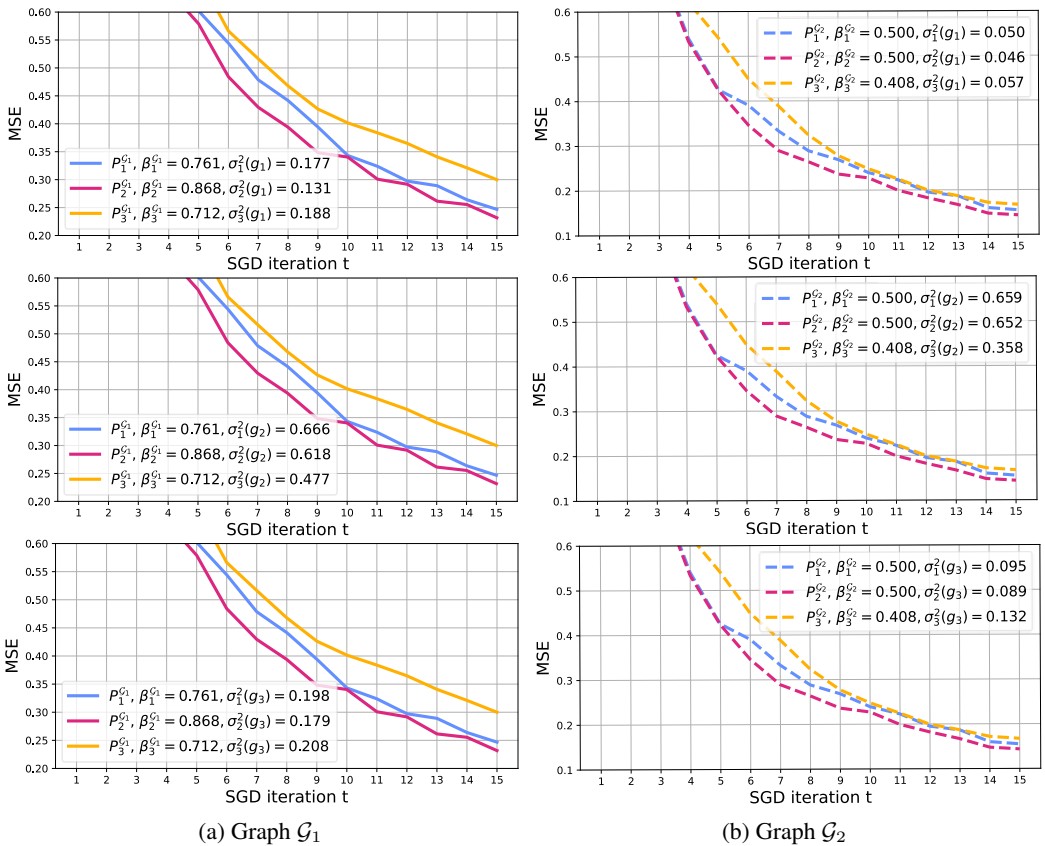

(a) Graph $\mathcal{G}_1$        (b) Graph $\mathcal{G}_2$

Figure 5: MSE $\mathbb{E}\|\theta_t - \theta^*\|_2^2$ of three Markov chains in the SGD algorithm with iteration (57).

In Figure 5, we repeat the plot in each graph three times with three different values of AVs inside the legend, the reason being that we want to see if the performance of each Markov chain is related to the AV $\sigma^2(g)$ and its test function $g$, other than SLEM solely. In the top row of Figure 5, as well as in Figure 1, we choose the test function $g_1(i) = \nabla F(\theta^*, i)$ for $i = 1, 2, \cdots, n$, where $\nabla F(\theta, i)$ is the gradient of the local function $F(\theta, i)$ (56) w.r.t $\theta$. In the middle row of Figure 5, the test function is $g_2(i) = d_i$, which estimates the average degree of the graph. In the bottom row of Figure 5, the test function is $g_3(i) = \mathbb{1}_{\{i=1\}}$, which estimates the probability of visiting node 1. We include the AVs of all three test functions $g_1, g_2, g_3$ in the legend of Figure 5, e.g., $\sigma_3^2(g_1)$ is the AV of the test function $g_1$ for FMMC.[7] We observe that $\sigma_1^2(g_1) < \sigma_3^2(g_1)$ and $\sigma_1^2(g_3) < \sigma_3^2(g_3)$ while $\sigma_1^2(g_2) > \sigma_3^2(g_2)$ in both graphs. This means MHRW and FMMC are *not* efficiency ordered, which is possible because efficiency ordering is a partial order such that not every two Markov chains can be ordered. On the other hand, in both graphs, $\sigma_1^2(g_k) > \sigma_2^2(g_k)$ for $k = 1, 2, 3$, which is consistent with the fact that the constructed Modified-MHRW is more efficient than MHRW. Regarding the MSE, we find that Modified-MHRW performs better than MHRW in both graphs, which is in line with the efficiency ordering. This leads us to conjecture that two efficiency ordered Markov chains might also have their performance in the RWSGD algorithm ordered in the same way.

---

[7]The AV of the test function for each Markov chain is calculated by running a stochastic simulation for a long time and directly computing according to the definition in (4).

**Remark I.3.** Section 1.1 in [35] numerically compares the performance of a reversible Markov chain and its non-reversible counterpart in the RWSGD algorithm w.r.t SLEM, and shows that the non-reversible counterpart with smaller SLEM performs better. The main theorem therein is also applicable to the comparison of two reversible Markov chains. However, as shown in Figure 5, we provide examples to show that a reversible Markov chain with smaller SLEM does not necessarily lead to smaller MSE in the RWSGD algorithm. Note that our results do not contradict the simulation results in Section 1.1 of [35], since it is possible for a Markov chain to have both smaller AV and smaller SLEM; which could be the case for the non-reversible counterpart in [35], although they didn't specify the AV in their simulation. Moreover, their main theorem is an upper bound to the error terms considered, which means that an SLEM-based ordering does not guarantee a performance ordering of the error terms themselves, as also exemplified in Figure 5. All of these together imply that SLEM alone cannot be the sole indicator of performance of the Markov chains as input sequences for RWSGD algorithms. □

## I.2 Numerical Results on Large Graphs

We first specify the process of dataset generation for the sum-of-nonconvex functions $\hat{f}(\theta)$ in (14). We generate random vectors $\mathbf{a}_1, \cdots \mathbf{a}_n, \mathbf{b} \in \mathbb{R}^{10}$ uniformly from $[0, 1]$ and ensure the invertibility of $\sum_{i=1}^{n} \mathbf{a}_i \mathbf{a}_i^T$. Then, we randomly select half of the matrices in $\{\mathbf{D}_i\}_{i \in [n]}$ and assign $+1.1$ to their $j$-th diagonal; other matrices are assigned $-1.1$ to $j$-th diagonal. We repeat the above process for all diagonal values $j = 1, 2, \cdots, 10$. This process guarantees $\sum_{i=1}^{n} \mathbf{D}_i = \mathbf{0}$.

We perform additional simulations on graph 'AS-733' [22] with 6474 nodes, and graph 'wikiVote' [21] with 889 nodes with the same objective functions $\tilde{f}(\theta)$ and $\hat{f}(\theta)$ in (14). The simulation results are given in Figure 6 and 7. We plot the curves of NBRW and SRW in the insets of Figures 6a, 6b, 7a, and 7b, with the same x,y axes but at linear scale, to better observe the difference in their performance. For both objective functions, NBRW has smaller MSE than SRW and both random and single shuffling perform better than uniform sampling, e.g., Figure 6a and 7a.[8] This demonstrate that NBRW and SRW are efficiency-ordered, which also holds for random/single shuffling and uniform sampling. Note that since we simulate on large graphs, for the logistic regression problem, the SGD algorithm with NBRW and SRW is yet to enter the asymptotic regime even in the $100,000$-th iteration, which can be explained by the blue and green increasing curves in the inset of Figure 6b and 7b. On the other hand, the curve of uniform sampling becomes flat and the curves of single/random shuffling are starting to go down in Figure 6b, 6d and 7b, 7d, implying that they have entered the asymptotic regime. These results are consistent to the observations in Figure 2, which support our theory.

## I.3 Additional Simulations on Non-convex Objective Function and SGD Variants

Regarding the SGD variants other than the vanilla SGD, central limit theorem (CLT) is less well studied in the literature. To list a few, [20] studied variance reduced SGD (SVRG) and obtained the CLT for constant step size. [2] analyzed Adam and their follow-up [3] extended the CLT for a general SGD algorithm, which includes Stochastic Heavy Ball (SHB), Nesterov accelerated SGD (NaSGD) and Adam. [24] established the CLT for momentum SGD (mSGD) and NaSGD under more general conditions on the step size. However, all of these recent works focus only on the Martingale difference noise

$$\mathbb{E}[\delta_{t+1}|\mathcal{F}_t \triangleq \sigma(\theta_0, X_0, X_1, \cdots, X_t)] = \mathbb{E}[\nabla f(\theta_t) - \nabla F(\theta_t, X_{t+1})|\mathcal{F}_t] = 0,$$

which is equivalent to saying that the input $\{X_t\}_{t \geq 0}$ is independently sampled from some identical distribution for each time $t$ (*i.i.d* input sequence). Meanwhile, for Markovian inputs,

$$\mathbb{E}[\delta_{t+1}|\mathcal{F}_t] = \sum_{i \in [n]} \pi_i \nabla F(\theta_t, i) - \sum_{i \in [n]} P(X_t, i) \nabla F(\theta_t, i) \neq 0$$

because $\pi_i \neq P(X_t, i)$ in general (unless $\{X_t\}_{t \geq 0}$ is an *i.i.d* sequence). It remains an open problem to obtain the CLT for these SGD variants with general Markovian inputs, which would be a prerequisite

---

[8]The curves of NBRW and SRW in Figure 6a and 7a appear flat because they are plotted in the same figure with uniform sampling and single/random shuffling, which have much smaller MSE. We plot the comparison between NBRW and SRW separately in the inset.

for our efficiency ordering. Indeed, one of our future works is to theoretically prove the CLT results for SGD variants with Markovian inputs and to carry over our efficiency ordering of different stochastic inputs.

Next, we simulate two SGD variants, i.e., Nesterov accelerated SGD (NaSGD) and ADAM, on graph "AS-733" (as used in Appendix I.2) with two pair of stochastic inputs, i.e., NBRW versus SRW and shuffling methods versus *i.i.d* input sequence, with respect to both convex objective function and non-convex objective function. We choose the convex objective function $\hat{f}(\theta)$ from (14) such that

$$\hat{f}(\theta) = \frac{1}{n} \sum_{i=1}^{n} \theta^T (\mathbf{a}_i \mathbf{a}_i^T + \mathbf{D}_i)\theta + \mathbf{b}^T \theta, \tag{61}$$

where $\sum_{i=1}^{n} \mathbf{a}_i \mathbf{a}_i^T$ is invertible and $\sum_{i=1}^{n} \mathbf{D}_i = \mathbf{0}$. We can see $\nabla^2 \hat{f}(\theta) = \frac{2}{n} \sum_{i=1}^{n} \mathbf{a}_i \mathbf{a}_i^T$ is a positive semi-definite matrix and $\hat{f}(\theta)$ is convex. Then, we modify matrices $\{\mathbf{D}_i\}_{i \in [n]}$ such that the first element on the main diagonal of each matrix $\mathbf{D}_i$ is subtracted by 0.1, and we denote the new matrices as $\{\mathbf{M}_i\}_{i \in [n]}$. We define a new function $\hat{g}(\theta)$ such that

$$\hat{g}(\theta) = \frac{1}{n} \sum_{i=1}^{n} \theta^T (\mathbf{a}_i \mathbf{a}_i^T + \mathbf{M}_i)\theta + \mathbf{b}^T \theta. \tag{62}$$

We numerically compute $\nabla^2 \hat{g}(\theta) = \frac{2}{n} \sum_{i=1}^{n} (\mathbf{a}_i \mathbf{a}_i^T + \mathbf{M}_i)$ and ensure it has at least one negative eigenvalue such that the objective function $\hat{g}(\theta)$ is non-convex. For Nesterov accelerated SGD, we employ the following iteration from [24]:

$$\begin{aligned} \theta_{t+1} &= u_t - \gamma_{t+1}\nabla G(u_t, X_{t+1}), \\ u_{t+1} &= \theta_{t+1} + \beta_{t+1}(\theta_{t+1} - \theta_t), \end{aligned} \tag{63}$$

where $\gamma_t = 1/0.9^t$ and $\beta_{t+1} \equiv \beta = 0.5$ in our settings. For ADAM, we use the following iteration from [18]:

$$\begin{aligned} g_{t+1} &= \nabla G(\theta_t, X_{t+1}), \\ m_{t+1} &= \alpha_1 m_t + (1 - \alpha_1)g_{t+1}, \\ v_{t+1} &= \alpha_2 v_t + (1 - \alpha_2)g_{t+1}^2, \\ m' &= m_{t+1}/(1 - \alpha_1^t), \\ v' &= v_{t+1}/(1 - \alpha_2^t), \\ \theta_{t+1} &= \theta_t - \gamma_t m'/(\sqrt{v'} + \epsilon), \end{aligned} \tag{64}$$

where $\gamma_t = 1/0.9^t$, $\alpha_1 = 0.9, \alpha_2 = 0.999$, $\epsilon = 10^{-8}$, $g_{t+1}^2$ is the element-wise square for the vector $g_{t+1}$ and $\sqrt{v'}$ is the element-wise square root for the vector $v'$. In both (63) and (64), function $G(\theta, i) = \theta^T (\mathbf{a}_i \mathbf{a}_i^T + \mathbf{D}_i)\theta + \mathbf{b}^T \theta$ for convex objective function $\hat{f}(\theta)$ and $G(\theta, i) = \theta^T (\mathbf{a}_i \mathbf{a}_i^T + \mathbf{M}_i)\theta + \mathbf{b}^T \theta$ for non-convex objective function $\hat{g}(\theta)$.

The insets of Figure 8 are to enlarge the curves of NBRW and SRW in ADAM algorithm for the iteration $t \in [40000, 50000]$ to make them more distinguishable. In Figure 8, we show that for both convex and non-convex objective functions, the curves of NBRW are always below those of SRW in vanilla SGD, NaSGD and ADAM, respectively. This not only supports our Theorem 3.6 on vanilla SGD and both convex and non-convex objective functions, but also suggest that the efficiency ordering is still valid for other SGD variants. In Figure 9, we also empirically test the performance of shuffling methods and uniform sampling on vanilla SGD, NaSGD and ADAM with non-convex objective function $\hat{g}(\theta)$. In all three SGD iterations, we show that shuffling methods are better than uniform sampling, although the gap between shuffling methods and uniform sampling is small in NaSGD and ADAM in Figure 9b and 9c and the reason could be that these SGD variants implicitly include the "momentum" may decrease the effect of the correlation from the stochastic inputs. We also notice from Figure 8 that for a given stochastic input, NaSGD and ADAM are better than vanilla SGD, while in Figure 9 the result is reversed. Currently, we only know that those SGD variants NaSGD and ADAM work better than vanilla SGD in practice for *i.i.d* input sequence. It remains an open problem for SGD variants with general Markovian inputs, and thus, it's possible that Markovian inputs can influence the performance of NaSGD and ADAM, compared to vanilla SGD. In any case, Figure 8 and Figure 9 still validate our efficiency ordering.

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

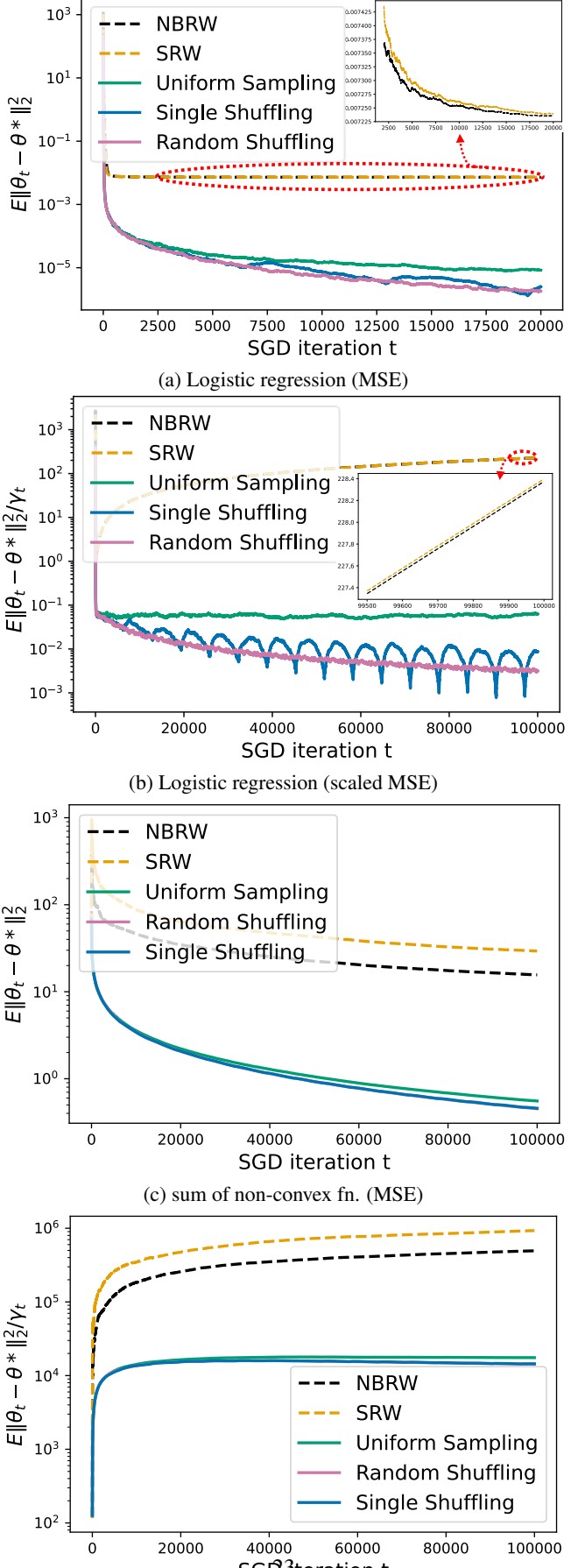

(a) Logistic regression (MSE)

(b) Logistic regression (scaled MSE)

(c) sum of non-convex fn. (MSE)

(d) sum of non-convex fn. (scaled MSE)

Figure 6: Performance comparison of different stochastic inputs on the graph 'AS-733'.

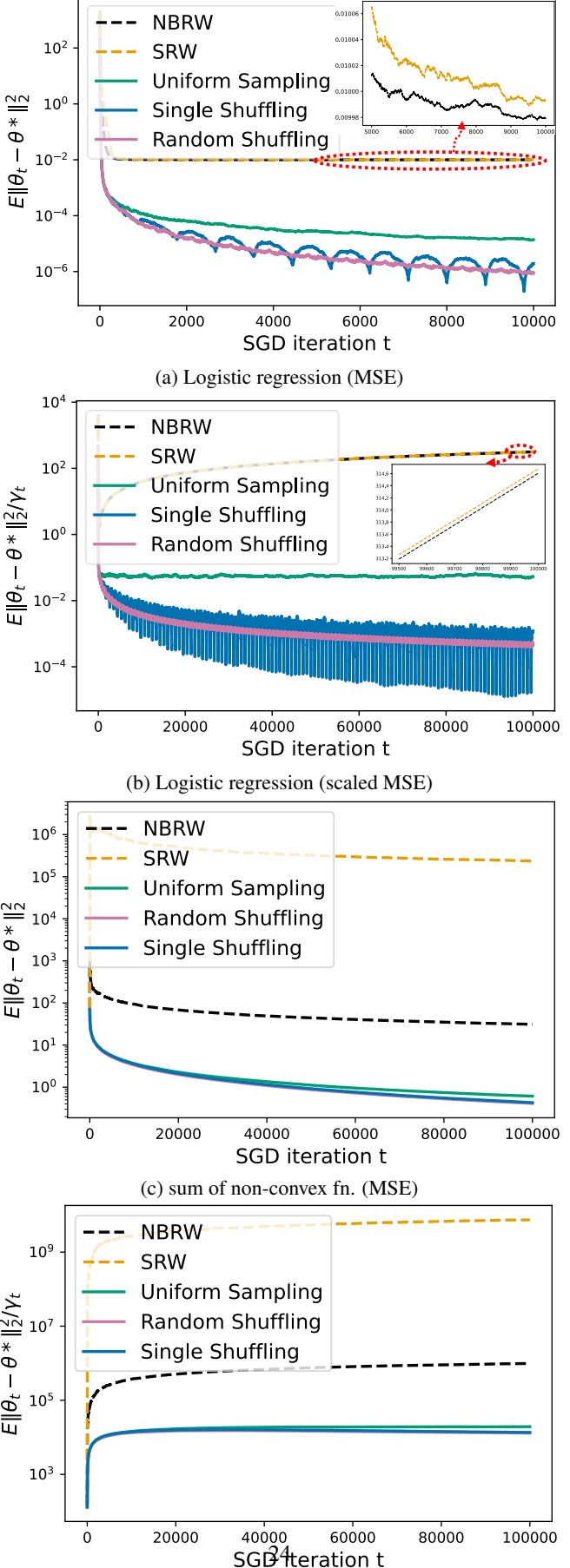

(a) Logistic regression (MSE)

(b) Logistic regression (scaled MSE)

(c) sum of non-convex fn. (MSE)

(d) sum of non-convex fn. (scaled MSE)

Figure 7: Performance comparison of different stochastic inputs on the graph 'wikiVote'.

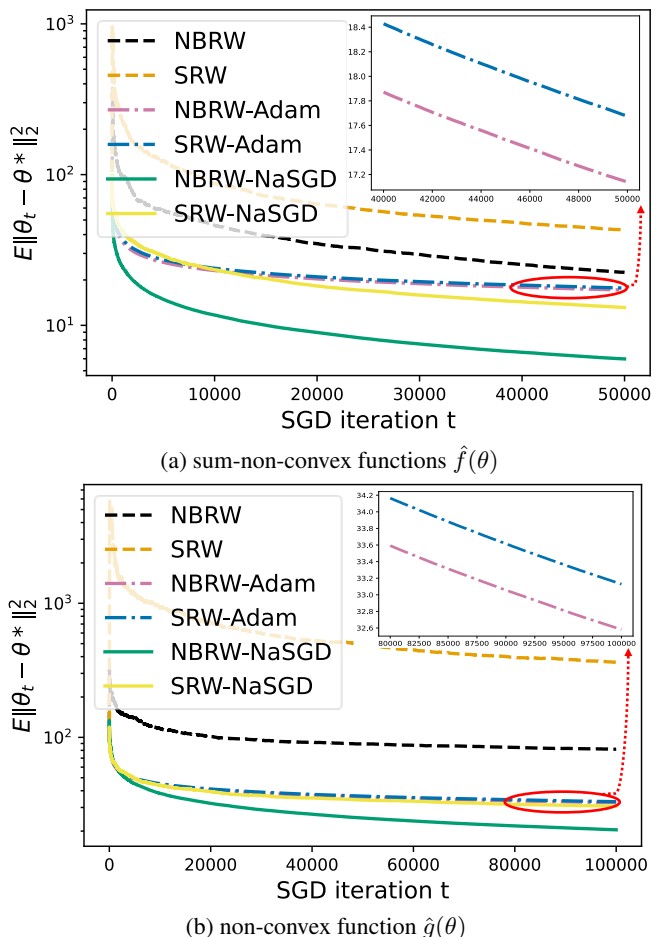

(a) sum-non-convex functions $\hat{f}(\theta)$

(b) non-convex function $\hat{g}(\theta)$

Figure 8: Performance comparison of NBRW and SRW in vanilla SGD, NaSGD and ADAM algorithms on the graph "AS-733".

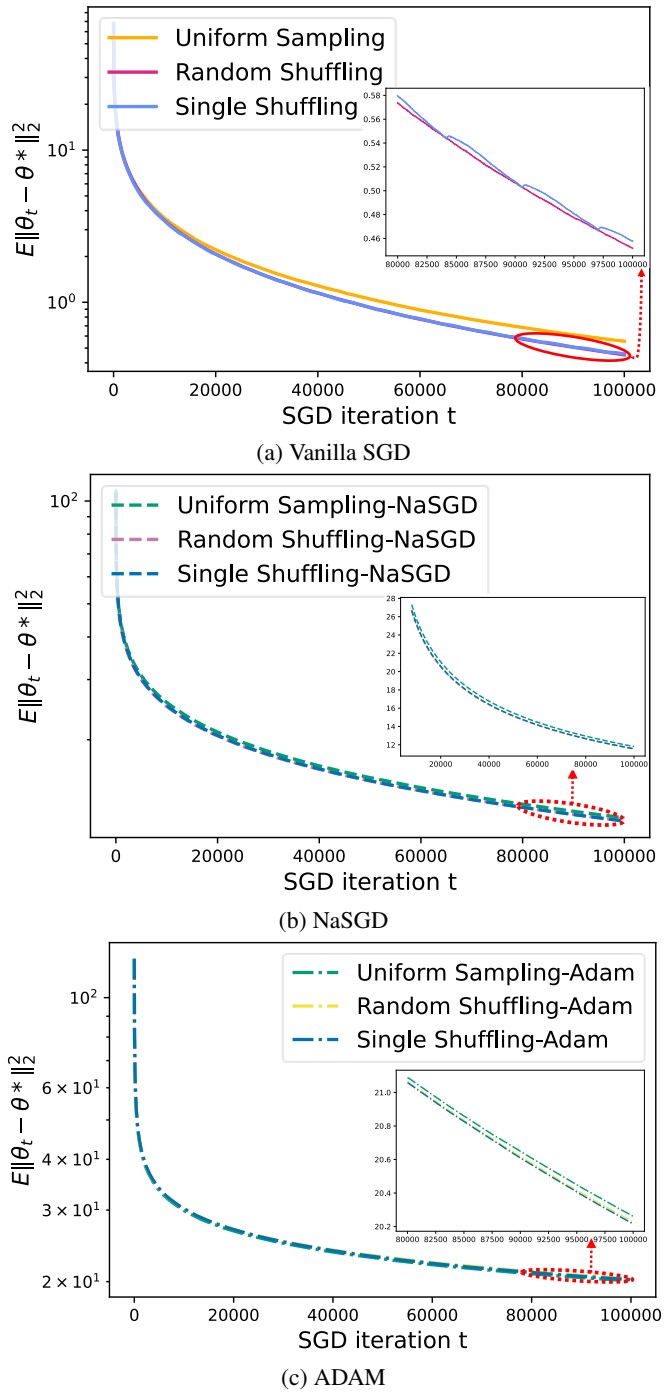

(a) Vanilla SGD

(b) NaSGD

(c) ADAM

Figure 9: Performance comparison of shuffling methods and uniform sampling with non-convex objective function $\hat{g}(\theta)$ on the graph "AS-733".