# OpenReview forum: "Efficiency Ordering of Stochastic Gradient Descent"
_NeurIPS.cc/2022/Conference — NeurIPS 2022 Accept_

### Official Review · Reviewer_3mgL · 2022-07-08

**Rating:** 7
**Confidence:** 4
**Soundness:** 4 excellent
**Presentation:** 3 good
**Contribution:** 3 good

**Summary:**

At each iteration, stochastic gradient descent (SGD) processes data sampled randomly from the training set. Denote this sequence of training data by $(X_t)_t$, which includes the case where $X_t$ are mini-batches. Empirical evidence by the authors suggest that the convergence of SGD is faster when the asymptotic variance (AV) of the chain $(X_t)_t$ is smaller, contrary to the commonly held belief that smaller SLEM (second largest eigenvalue modulus) lead to faster convergence of SGD. This motivates the authors to study the AV of data sampling approaches commonly employed in SGD, such as mini-batch shuffling, and define efficiency ordering on chains based on their AV (lower AV implies higher efficiency). They prove rigorously that certain non-Markovian sequences such as shuffling give better efficiency ordering than Markovian i.i.d. input sequences. They then demonstrate experimentally that the former indeed gives rise to faster converging SGD when applied to the CIFAR-10 dataset.

**Questions:**

- While the main body of the paper is well-written, the appendix I believe is not at the same quality and can be improved. For instance:
   - Line 69: "We can check the solution $\tilde{F}(\theta, z)$". What are we checking?
   - The proof of Proposition 4.1 (Appendix F) is confusing. What are the processes $X$ and $X'$? How are they related to the NBRW and SRW? While I agree with the computations in the proof, I still don't understand how it can be used to conclude that NBRW is more efficient than SRW.
   - In Appendix G, the line "regroup all the terms in each $k$-th epoch with length $n$" (line 226) is difficult to understand (I do understand now, but it took a while). It may help to supplement with a diagram.
   - What is $\hat{\pi}$ in the proof of Lemma H.1? How is it different from $\pi$?
   - I don't understand how the following line is true (Line 291-292):
> We also note that $P(s_t, s_{t+j}) = 0$ for $j \neq kn$ and $k \in \mathbb{N}$.

      Please could you explain more?
   - What do the embedded plots in figures 5a), 5b), 6a) and 6b) show?

- There are several misprints peppered throughout the appendix.
   - Line 72: I suggest phrasing "are right and left eigenvector of P" as "are the right and left eigenvectors of P respectively".
   - Line 94-95: "to ensure $\theta^*$ being local minimizer" should be "to ensure that $\theta^*$ is the local minimizer".
   - Line 226: I think "$\sum_{i = (k-1)n}^{kn-1} (g(X_i) - E_\pi (g))$" should read "$\sum_{i = (k-1)n}^{kn-1} (g(X_i) - E_\pi(g)) < \infty$".
   - Line 229: "because of bounded function $g$" should be "because of the boundedness of function $g$".
   - Line 247: "Especially, $u_m = \hat{\pi}$" should be "In particular, $u_m = \hat{\pi}$"
   - $s_6$ in line 268 should be $s_8$.
   - $a_j^{(t)}$ in line 290 should be $A_j^{(t)}$.
   - Appendix G: Lemma 4.2 is mislabelled as Proposition 4.2.

- While the results point to new methods for improving SGD, in practice, how easy is it to compute the asymptotic variance for a given chain? Also, is efficiency ordering still valid for other stochastic optimization methods such as Adam?


**Limitations:**

The work is mostly theoretical and as far as I can see, has no potential negative societal impact.

**Strengths And Weaknesses:**

Strengths:
- The results in the paper give new insights into SGD. In particular, it opens up a direction to look for more efficient SGD algorithms.
- The techniques involved in the proof are interesting and completely non-trivial. In particular, the idea of lifting non-Markovian sequences to an augmented space to make it Markovian was intriguing. The maths overall seem correct (although I do have some questions below) and fully rigorous.
- The main body of the paper is well-written and easy to follow.

Weakness:
- While the main body of the paper is well-written, the appendix, which contains all the proofs, can be improved. I defer some of my issues in the appendix to "Questions".
- Plots can be improved by:
   - Using larger label sizes (e.g. labels in Figure 2 are almost impossible to see without zooming in. Also better to increase label sizes in figure 1. Applies also to the figures in the appendix).
   - Improve colour-scheme by taking into consideration colorblindness. For instance, avoid red-green-blue combination (see e.g. https://davidmathlogic.com/colorblind/#%23D81B60-%231E88E5-%23FFC107-%23004D40 for more details).
- There are a few misprints/suggestions in the text of the main body that I spotted:
   - Line 140: $\mathbf{M}" \in \mathbb{R}^n$ should be $\mathbf{M} \in \mathbb{R}^n$ (note the quotation mark)
   - Line 268: "From the nature of same limiting distributions" can be phrased better. Maybe along the lines of "Since the limiting distributions are the same".
   - Line 369: "we random generate" should be "we randomly generate".

---

> ### Author Response · Authors · 2022-08-02
> **Response to Reviewer 3mgL (1/3)**
>
> We appreciate the reviewer for the detailed comments. We have improved the plots and fixed a few typos in the revision. Please find our replies to your questions below.
>
> >**(Comment 1): Line 69: "We can check the solution $\\tilde{F}(\\theta,z)$". What are we checking?**
>
> Answer: When saying ``check the solution $\\tilde{F}(\\theta,z)$'', what we meant was to verify that $\\tilde{F}(\\theta,z)$ is a Lipschitz-continuous function and satisfies equation (20) in assumption (B2). To make our intention clearer, we modify the sentence with the following phrase:
>
> ``Next, we can rewrite $\tilde{F}(\theta,z)$ in the closed form and show that it is Lipschitz continuous and satisfies (20) in assumption (B2).''
>
>
> >**(Comment 2): The proof of Proposition 4.1 (Appendix F) is confusing. What are the processes $X$ and $X'$? How are they related to the NBRW and SRW? While I agree with the computations in the proof, I still don't understand how it can be used to conclude that NBRW is more efficient than SRW.**
>
> Answer: In Appendix F we use $\\{X'\_t\\}\_{t\\geq 0}$ to represent the NBRW process while we use $\\{Y\_t\\}\_{t\\geq 0}$ for NBRW in Section 4, which we believe might have caused the confusion. To be consistent, we clarify that the process $\\{X\_t\\}\_{t\\geq 0}$, is SRW and $\\{Y\_t\\}\_{t\\geq 0}$ is the trajectory on the node space $\\mathcal{V} = [n]$ driven by NBRW throughout the supplementary material.
>
> We assume the reviewer wants to know how Theorem F.1 in Appendix F can be used to prove the efficiency ordering result. Since $\\{Y\_t\\}\_{t\\geq0}$ is non-Markovian process, we are unable to provide a CLT result directly for the non-augmented process $\\{Y\_t\\}\_{t\\geq0}$. However, we can do this for an augmentation given by $\\{Z\_t\\}\_{t \\geq 0}$, where $Z\_t \\triangleq (Y\_{t-1},Y\_t)$, which is Markovian. With this, we can show that its asymptotic covariance matrix coincides with the one for the non-augmented process $\\{Y\_t\\}\_{t\\geq0}$, that is, we can show $\\mathbf{\\Sigma}\_{Z}(\\nabla \\Phi(\\theta^*,\\cdot)) = \\mathbf{\\Sigma}\_Y(\\nabla G(\\theta^*,\\cdot))$ via equation (50), where $\\Phi(\\theta,Z\_t)\\triangleq G(\\theta,Y\_t)$ is an augmentation of the underlying function $G(\\theta,\\cdot)$. This then enables us to show that $\\mathbf{\\Sigma}\_{SRW} \\triangleq \\mathbf{\\Sigma}\_{X}(\\nabla G(\\theta^*,\\cdot))$ and $\\mathbf{\\Sigma}\_{NBRW} \\triangleq \\mathbf{\\Sigma}\_Y(\\nabla G(\\theta^*,\\cdot)) = \\mathbf{\\Sigma}\_{Z}(\\nabla \\Phi(\\theta^*,\\cdot))$ follow the ordering $\\mathbf{\\Sigma}\_{NBRW} \\leq\_L \\mathbf{\\Sigma}\_{SRW}$ by leveraging an existing result from [Lee2012] involving the efficiency ordering of asymptotic variances of NBRW and SRW and extending it to the Loewner ordering of asymptotic covariance matrices in Theorem 3.6(i), and as a result, by Theorem 3.6(ii) we have $\\mathbf{V}\_Z \\leq\_L \\mathbf{V}\_X$.
>
> >**(Comment 3): In Appendix G, the line "regroup all the terms in each k-th epoch with length n" (line 226) is difficult to understand (I do understand now, but it took a while). It may help to supplement with a diagram.**
>
> Answer: Thanks for the suggestion, we have added a diagram in Appendix G in the revision to illustrate the shuffling without replacement for better explanation.
>
> >**(Comment 4): What is $\\hat{\\pi}$ in the proof of Lemma H.1? How is it different from $\\pi$?**
>
> Answer: $\\hat{\\pi}$ is a typo of $\\pi$ and has been fixed in the revision.

---

> ### Author Response · Authors · 2022-08-02
> **Response to Reviewer 3mgL (2/3)**
>
> >**(Comment 5): I don't understand how the following line is true (Line 291-292):
> $$
>     P(s\_t,s\_{t+j})=0, \\text{for~} j\\neq kn, k\\in \\mathbb{N}.
> $$
> Please could you explain more?**
>
> Answer: What we mean by the expression $P(s\_t,s\_{t+j})=0$ is that the probability of self-transition to the current state in $j$ steps is zero. We admit that this notation can be confusing, so we now instead use the more common expression $P(s\_{t+j}=s|s\_t=s) = 0$.
> Recall the definition $s\_t \\triangleq \\{\\{A\_j^{(t)}\\}\_{j\\in[n]},c\_t\\}$, where the sequence $\\{A\_j^{(t)}\\}\_{j\\in[n]}\\triangleq \\{X\_{t-n+1},X\_{t-n+2},\\cdots,X\_t\\}$ is of length $n$, and records the history of past $n$ indices until time $t$. The integer $c\_t\\in \\{1,2,\\cdots,n\\}$ is the time spent in current epoch at time $t$. We improve the explanation of $P(s\_{t+j}=s|s\_t=s) = 0$ for $j\\neq kn, k\\in \\mathbb{N}$ in lines 291-292 in Appendix H.2 as follows.
>
> ``... for $k\\in\\mathbb{N}$. We also note that $P(s\_{t+j}=s|s\_t=s) = 0$ for $s\\in\\mathcal{S}$, $j\\neq kn$ and $k\\in\\mathbb{N}$. Since $\\{c\_t\\}$ by its definition is a periodic sequence $\\{1,2,\\cdots,n,1,2,\\cdots,n,1,2,\\cdots\\}$ with period of length $n$, we know that $c\_t=c\_{t+j}$ holds only when $j=kn$ for $k\\in\\mathbb{N}$. Then, for $j\\neq kn$, we have $c\_{t+j}\\neq c\_{t}$ such that $s\_{t+j} \\neq s\_t$, which leads to $P(s\_{t+j}=s|s\_t=s)=0$ for $j\\neq kn$ and $k\\in\\mathbb{N}$. Therefore, by definition ...''
>
> >**(Comment 6): What do the embedded plots in figures 5a), 5b), 6a) and 6b) show?**
>
> Answer: We have mentioned in footnote 6 in the supplementary material that the insets in Figure 5a) and 6b) are the comparison of the mean square error between NBRW and SRW in order to distinguish them more clearly. This also applies to Figure 5b) and 6b) to show the scaled mean square error and validate the CLT result. All the embedded plots indicate that NBRW is more efficient than SRW in the SGD iteration. To be more precise, we add a sentence in lines 419-420 in the supplementary material to explain the inset plots of Figure 5a), 5b), 6a) and 6b).
>
> ``We plot the curves of NBRW and SRW in the insets of Figures 5(a), 5(b), 6(a), and 6(b), with the same x,y axes but at linear scale, to better observe the difference in their performance.''

---

> > ### Comment · Reviewer_3mgL · 2022-08-10
> > **Thanks for clarifying + comments on plots**
> >
> > I thank the authors for clarifying my confusions. I have also checked that the plots have improved and was happy to see that additional experiments were included that addresses large graphs and empirical results on SGD variants. I would say that the embedded plots in Figures 6-9 still have tick labels that are impossible to see and would suggest the authors to improve the visibility, if not remove it altogether. It is also slightly disturbing that the subplots have different sizes e.g. compare the size of Fig. 9c vs 9b and 9a. Not important though, but would be nice if this could be fixed.

---

> ### Author Response · Authors · 2022-08-02
> **Response to Reviewer 3mgL (3/3)**
>
> >**(Comment 7): While the results point to new methods for improving SGD, in practice, how easy is it to compute the asymptotic variance for a given chain? Also, is efficiency ordering still valid for other stochastic optimization methods such as Adam?**
>
> Answer: We don't need to compute the covariance matrix in the CLT result to determine which Markov chain is more efficient as an input sequence for SGD. Instead, our starting point is to employ a more efficient chain as the stochastic input to improve the performance of the SGD iteration and show that the resulting SGD becomes more efficient with smaller covariance matrix, in the sense of Loewner ordering, when fed by a more efficient chain. The examples we give in the paper are NBRW versus SRW and shuffling methods versus uniform sampling. Additional details on efficient Markov chains are referred to [Sun2010], [Bremaud2013] Chapter 11, [Schwedes2021].
>
> To compute the asymptotic variance, we would like to refer the reviewer to Theorem 6.3.7 in [Bremaud2013], which gives the closed form of the asymptotic variance for any scalar-valued function. For any vector-valued function $\\mathbf{g}$, the closed form still holds for asymptotic covariance matrix $\\mathbf{\\Sigma}\_X(\\mathbf{g})$. However, in order to calculate $\\mathbf{\\Sigma}\_X$ in the CLT result (Lemma 3.1), we also need to know the optimal point $\\theta^*$, which is usually unknown beforehand in reality and makes the computation infeasible. By leveraging a more efficient Markov chain into the SGD iteration, we don't need to compute covariance matrix and we can theoretically guarantee that the resulting SGD is more efficient when driven by a more efficient chain with smaller covariance matrix, in the sense of Loewner ordering.
>
> Per your suggestion, we have empirically tested Nesterov accelerated SGD (NaSGD) and Adam for NBRW versus SRW and shuffling methods versus i.i.d input sequence, and found that the efficiency ordering is still valid for these SGD variants. The simulation results and the literature review on asymptotic analysis of SGD variants have been included in Appendix I.3 in the revision. In what follows, we will briefly introduce the latest works on SGD variants. To the best of our knowledge, the CLT result of other SGD variants for Markovian inputs, which would be a prerequisite for efficiency ordering, remains an open problem. Only recently some works start to analyze the asymptotic property of the SGD variants, e.g., [Lei2020] studied variance reduced SGD, [Li2022] studied momentum SGD (mSGD) and NaSGD, [Barakat2021] analyzed Adam Algorithm and [Barakat2021-2] extended the CLT for a general SGD algorithm. However, all these works focus only on the SGD iterates with Martingale difference noise term $\\delta_{t+1} = \\nabla f(\\theta_t) - \\nabla F(\\theta\_t,X\_{t+1})$ such that $\\mathbb{E}[\\delta\_{t+1}|\\mathcal{F}\_{t} \\triangleq \\sigma(\\theta\_0,X\_0,X\_1,\\cdots,X\_t)]=0$, which is equivalent to saying that $\\{X\_t\\}\_{t\\geq 0}$ is independently sampled from some identical distribution for each time $t$. Meanwhile, for Markovian inputs, $\\mathbb{E}[\\delta\_{t+1}|\\mathcal{F}\_{t}]= \\sum\_{i\\in[n]}\\pi_i \\nabla F(\\theta\_t,i) - \\sum\_{i\\in[n]}P(X\_{t},i)\\nabla F(\\theta\_t,i) \\neq 0$ because $\\pi_i \\neq P(X\_{t},i)$ in general (unless $\\{X\_t\\}\_{t\\geq 0}$ is an i.i.d sequence). Indeed, one of our future works is to theoretically prove the CLT results for SGD variants with Markovian inputs and to carry over our efficiency ordering of different stochastic inputs.
>
> [Sun2010] Yi Sun, Jürgen Schmidhuber, and Faustino Gomez. ``Improving the asymptotic performance of Markov chain Monte-Carlo by inserting vortices.'' Advances in Neural Information Processing Systems 23, 2010.
>
> [Bremaud2013] Pierre Bremaud. ``Markov chains: Gibbs fields, Monte Carlo simulation, and queues.'' Vol. 31. Springer Science \& Business Media, 2013.
>
> [Schwedes2021] Tobias Schwedes, and Ben Calderhead. ``Rao-blackwellised parallel MCMC.'' International Conference on Artificial Intelligence and Statistics, 2021.
>
> [Lei2020] Jinlong Lei, and Uday V. Shanbhag. ``Variance-Reduced Accelerated First-order Methods: Central Limit Theorems and Confidence Statements.'' arXiv:2006.07769, 2020.
>
> [Li2022] Tiejun Li, Tiannan Xiao, and Guoguo Yang, ``Revisiting the central limit theorems for the SGD-type methods'', arXiv:2207.11755, 2022.
>
> [Barakat2021] Anas Barakat, and Pascal Bianchi. ``Convergence and dynamical behavior of the ADAM algorithm for nonconvex stochastic optimization.'' SIAM Journal on Optimization 31.1 (2021): 244-274.
>
> [Barakat2021-2] Anas Barakat, Pascal Bianchi, Walid Hachem, and Sholom Schechtman. ``Stochastic optimization with momentum: convergence, fluctuations, and traps avoidance.'' Electronic Journal of Statistics 15.2 (2021): 3892-3947.

---

### Official Review · Reviewer_9xiZ · 2022-07-11

**Rating:** 7
**Confidence:** 4
**Soundness:** 4 excellent
**Presentation:** 3 good
**Contribution:** 4 excellent

**Summary:**

This paper studies the stochastic gradient descent algorithm as a stochastic sequence generated by a random walk on an arbitrary graph. It explores various Markov Chain Monte Carlo (MCMC) sampling techniques and proposes "efficiency ordering" of Markov chains. This is used to demonstrate that more efficient MCMC sampling can also lead to smaller covariances for SGD algorithms in the limit.

**Questions:**

- The discussion on the mini-batch gradient descent can perhaps be relegated to the Appendix which would both free up space for more appropriate and thorough descriptions on empirical analysis of the efficiency ordering and ensure that variants of SGD are handled / discussed more efficiently. For e.g. if one discusses mini-batch gradient descent, why not other variants including momentum acceleration methods and other known techniques?

- The main paper does not discuss the effect of the underlying graph structure (which affects SLEM) and its impact on efficiency ordering of SGD

- The paper does not have a conclusion or future work section. It lacks a discussion on potential drawbacks of the technique.

Minor comments
- 1. Asymptotic covariance matrix - definition of \hat{mu}_t (g) -> sum_k=1 to t g(X_k) NOT X_t?


**Limitations:**

Societal impact of the work is not discussed in this paper. Furthermore, limitations of the proposed technique need further thought.

**Strengths And Weaknesses:**

Strengths
- Paper is well written and presents both theoretical and empirical studies of the effect of MCMC sampling on SGD algorithms
- It introduces the novel concept of "efficiency ordering" of Markov chains and its. theoretical underpinnings
-Comparison of non-Markov random walks to their Markovian counterpart.

Weakness
- The main paper lacks sufficient details on the simulation and figures are too small for readability
-The overall readability of the paper needs to be improved

---

> ### Author Response · Authors · 2022-08-02
> **Response to Reviewer 9xiZ (1/3)**
>
> Dear reviewer, thanks for the comment! Please find our replies to your concerns below.
>
> >**(Comment 1): The discussion on the mini-batch gradient descent can perhaps be relegated to the Appendix which would both free up space for more appropriate and thorough descriptions on empirical analysis of the efficiency ordering and ensure that variants of SGD are handled / discussed more efficiently. For e.g. if one discusses mini-batch gradient descent, why not other variants including momentum acceleration methods and other known techniques?**
>
> Answer: We agree with reviewer's comments. Due to the page limit, the detailed descriptions of simulation setup cannot be put into the main body and are deferred to Appendix I.2 and I.3 in the revision. We have moved the discussion on mini-batch gradient descent to Appendix H.3 and briefly mentioned the SGD variants in the newly added Remark 4.4 as follows (all the citations therein are based on the reference in the revision).
>
> ``Remark 4.4. Mini-batch gradient descent is another popular gradient descent variant and is widely used in the machine learning tools [2,22,55] to accelerate the learning process when compared to SGD. In Appendix H.3 [1], we show how our framework can be applied to study min-batch based SGD algorithms, and include the asymptotic analysis on mini-batch gradient descent with shuffling.
> Besides mini-batch gradient descent, there are other SGD variants, e.g., momentum SGD, Nesterov accelerated SGD and ADAM, that have been studied in the SGD literature for non-asymptotic analysis [6,39,59]. However, asymptotic analysis on the SGD variants is not well studied in the literature, with only very recent results on the CLT for i.i.d input sequences [8,9,42,45]. Asymptotic analysis and CLT for SGD variants with more general Markovian input sequences, which would be a prerequisite for Theorem 3.6, remains an open problem. We defer the discussion on related works to Appendix I.3 [1], where we also empirically test the SGD variants and find that the efficiency ordering result still holds for these SGD variants - opening up the possibility for further theoretical analysis.''
>
> As mentioned in Remark 4.4, detailed discussion on the related works is included in Appendix I.3 in the Supplementary Material (we also include them below for the reference), where we also perform additional simulation results showing that ``our efficiency ordering holds for SGD variants with Markovian inputs''.
>
> Regarding the SGD variants other than the vanilla SGD, central limit theorem (CLT) is less well studied in the literature. To list a few, [Lei2020] studied variance reduced SGD (SVRG) and obtained the CLT for constant step size. [Barakat2021] analyzed Adam and their follow-up [Barakat2021-2] extended the CLT for a general SGD algorithm, which includes Stochastic Heavy Ball (SHB), Nesterov accelerated SGD (NaSGD) and Adam. [Li2022] established the CLT for momentum SGD (mSGD) and NaSGD under more general conditions on the step size. However, all of these recent works focus only on the Martingale difference noise
> $\\mathbb{E}[\\delta\_{t+1}|\\mathcal{F}\_{t} \\triangleq \\sigma(\\theta\_0,X\_0,X\_1,\\cdots,X\_t)] = \\mathbb{E}[\\nabla f(\\theta\_t) - \\nabla F(\\theta\_t,X\_{t+1})|\\mathcal{F}\_{t}] = 0$,
> which is equivalent to saying that the input $\\{X\_t\\}\_{t\\geq 0}$ is independently sampled from some identical distribution for each time $t$ (i.i.d input sequence). It remains an open problem to obtain the CLT for these SGD variants with general Markovian inputs, which would be a prerequisite for our efficiency ordering. Indeed, one of our future works is to theoretically prove the CLT results for SGD variants with Markovian inputs and to carry over our efficiency ordering of different stochastic inputs.
>
> [Lei2020] Jinlong Lei, and Uday V. Shanbhag. ``Variance-Reduced Accelerated First-order Methods: Central Limit Theorems and Confidence Statements.'' arXiv:2006.07769, 2020.
>
> [Barakat2021] Anas Barakat, and Pascal Bianchi. ``Convergence and dynamical behavior of the ADAM algorithm for nonconvex stochastic optimization.'' SIAM Journal on Optimization 31.1 (2021): 244-274.
>
> [Barakat2021-2] Anas Barakat, Pascal Bianchi, Walid Hachem, and Sholom Schechtman. ``Stochastic optimization with momentum: convergence, fluctuations, and traps avoidance.'' Electronic Journal of Statistics 15.2 (2021): 3892-3947.
>
> [Li2022] Tiejun Li, Tiannan Xiao, and Guoguo Yang, ``Revisiting the central limit theorems for the SGD-type methods'', arXiv:2207.11755, 2022.

---

> ### Author Response · Authors · 2022-08-02
> **Response to Reviewer 9xiZ (2/3)**
>
> >**(Comment 2): The main paper does not discuss the effect of the underlying graph structure (which affects SLEM) and its impact on efficiency ordering of SGD.**
>
> Answer: It is possible that changing the underlying graph structure may lead to the efficiency ordering of SGD, e.g., SRW on Graph $\\mathcal{G}\_1$ and SRW on Graph $\\mathcal{G}\_2$ have their covariance matrices (in the CLT of the SGD iteration) Loewner-ordered for some graphs $\\mathcal{G}\_1,\\mathcal{G}\_2$. This is, however, not our focus in this paper. In fact, in order for two Markov chains (SRW on graph $\\mathcal{G}\_1$ versus SRW on graph $\\mathcal{G}\_2$) to be efficiency-ordered, their stationary distribution must be the same in the first place (and then their covariance matrices are Loewner-ordered). If we were to consider two different graphs (with different SLEMs) in this framework, the underlying graph structures mush be all preserving the same stationary distribution (e.g., identical degree sequence for SRW), and clearly this would limit the scope of the ordering-based study.
>
> Instead, what we focus in this paper is, for any given graph, and for a given SGD with a Markovian input, we argue that it's generally possible to construct a more efficient chain (on the same state space with the same stationary distribution), e.g., NBRW versus SRW and shuffling methods versus i.i.d input sequence, and investigate how this translates into the resulting iterates out of SGD. For instance, we showed in Proposition 4.1 that for any general undirected graph, NBRW is more efficient than SRW in SGD, implying that their efficiency ordering holds regardless of the change in the underlying graph, provided that we use NBRW (instead of SRW) as an input to the SGD iteration on the same graph. Designing a more efficient chain with smaller asymptotic variance has been popular especially in the MCMC literature, such as [Sun2010], [Bremaud2013] Chapter 11, [Schwedes2021]. With respect to SLEM, denote $1=\\lambda\_1>\\lambda\_2\\geq \\cdots\\geq\\lambda\_n>-1$ as the eigenvalues of the transition matrix for a given Markov chain, SLEM is defined as $\\max\\{|\\lambda\_2|,|\\lambda\_n|\\}$, while the asymptotic variance contains all the eigenvalues of the transition matrix, as shown in equation (6.34) of [Bremaud2013]. These two metrics are related but don't reflect the same behavior of the given chain, e.g., smaller SLEM tends to mix faster to the stationary distribution while smaller asymptotic variance tends to provide fewer samples to achieve a certain accuracy (as discussed in [Lee2012] Section 2.2). Moreover, for some stochastic processes, their SLEMs are not comparable on general graphs, as also explained in lines 289-292 of Section 4.
>
> [Sun2010] Yi Sun, Jürgen Schmidhuber, and Faustino Gomez. ``Improving the asymptotic performance of Markov chain Monte-Carlo by inserting vortices.'' Advances in Neural Information Processing Systems 23, 2010.
>
> [Bremaud2013] Pierre Bremaud. ``Markov chains: Gibbs fields, Monte Carlo simulation, and queues.'' Vol. 31. Springer Science \& Business Media, 2013.
>
> [Schwedes2021] Tobias Schwedes, and Ben Calderhead. ``Rao-blackwellised parallel MCMC.'' International Conference on Artificial Intelligence and Statistics, 2021.
>
> [Lee2012] Chul-Ho Lee, Xin Xu, and Do Young Eun. ``Beyond random walk and metropolis-hastings samplers: why you should not backtrack for unbiased graph sampling.'' ACM SIGMETRICS Performance evaluation review, 40(1):319–330, 2012.

---

> ### Author Response · Authors · 2022-08-02
> **Response to Reviewer 9xiZ (3/3)**
>
> >**(Comment 3): The paper does not have a conclusion or future work section. It lacks a discussion on potential drawbacks of the technique.**
>
> Answer: Due to the page limit, we didn't include the conclusion section in our original submission. Per this comment, we have now relegated the discussion on mini-batch gradient descent to Appendix H.3 so that we have more space to summarize our work and discuss the future work. We have added a conclusion section as follows in the revision.
>
> ``We have introduced the notion of efficiency ordering of SGD algorithms, and shown that processes with smaller asymptotic covariance are always more efficient as input sequences for SGD algorithms. Furthermore, we numerically observe that this sense of efficiency ordering is also seen SGD variants such as Nesterov accelerated SGD and Adam. Since the asymptotic analysis of such SGD variants is not well-established under general stochastic inputs, establishing theoretical results on efficiency ordering remain an open problem.''
>
> >**(Comment 4): Asymptotic covariance matrix - definition of $\\hat{\\mu}\_t (g)$ -> $\\sum\_{k=1}^{t} g(X\_k)$ NOT $X\_t$?**
>
> Answer: The definition of $\\hat{\\mu}\_t (\\mathbf{g})$ should be $\\hat{\\mu}\_t(\\mathbf{g}) \\triangleq \\frac{1}{t}\\sum\_{k = 1}^{t} \\mathbf{g}(X\_k)$ and has been fixed.

---

### Official Review · Reviewer_jRx9 · 2022-07-11

**Rating:** 6
**Confidence:** 3
**Soundness:** 3 good
**Presentation:** 3 good
**Contribution:** 3 good

**Summary:**

The authors introduce the notion of efficiency ordering to classify stochastic algorithms.
They claim that some non-Markovian dynamics are more efficient that the Markovian ones for SGD.


**Questions:**

The authors should first provide evidence that their proposal is indeed an improvement for stochastic algorithms on very large graphs and non-convex functions to optimize.


**Limitations:**

Do not apply.

**Strengths And Weaknesses:**

The paper is unclear, and the claims seem to overstate the results in general.
The figures present data which are far from convincing.
In particular, the numerical experiment made on the Dolphins graph of 62 nodes is not meaningful.
Given the generality of the statement made by the authors, I would have expected much higher accuracy numerical experiments run on much larger graphs.

---

> ### Author Response · Authors · 2022-08-02
> **Response to Reviewer jRx9**
>
> >**(Comment 1): The authors should first provide evidence that their proposal is indeed an improvement for stochastic algorithms on very large graphs and non-convex functions to optimize.**
>
> Answer: We appreciate the reviewer's constructive comment that can improve our simulation results in order to better support our theory. We accommodate the comment by adding more simulations on graph ``AS-733'' (with $6474$ nodes, $13895$ edges) with a non-convex objective function in Appendix I.3 in the revised supplementary material. Before explaining the more detailed setup of the newly added simulation, we would like to emphasize that our theoretical result on efficiency ordering of SGD noise/input sequences, that is, the performance ordering of two input sequences given up front, is guaranteed to hold for any arbitrary and possibly large graph, and any non-convex function that satisfies assumptions (A1)-(A5) in Section 3. Therefore, any simulation shown is purely for demonstration purposes and is not the primary evidence of our results, which are guaranteed to hold via mathematical proofs.
>
> For our newly added simulations, the non-convex objective function is given by
> $$
> \\hat{g}(\\theta) =\\frac{1}{n}\\sum_{i=1}^n \\theta^T(\\mathbf{a}\_i\\mathbf{a}\_i^T + \\mathbf{M}\_i)\\theta + \\mathbf{b}^T\\theta.
> $$
> Since we construct vectors $\\{\\mathbf{a}\_i\\}\_{i\\in[n]}$ and symmetric matrices  $\\{\\mathbf{M}\_i\\}\_{i\\in[n]}$ in a way (details are given in Appendix I.3) that $\\nabla^2 \\hat{g}(\\theta) = \\frac{2}{n}\\sum\_{i\\in[n]}(\\mathbf{a}\_i\\mathbf{a}\_i^T + \\mathbf{M}\_i)$ is not positive semi-definite, it ensures the non-convexity of $\\hat{g}(\\theta)$. The new simulation results in Appendix I.3 show that our efficiency ordering still holds for large graph, non-convex function. Moreover, per other reviewers' comments, we have empirically tested other SGD variants, e.g., Nesterov accelerated SGD and Adam, with Markovian inputs and found that our efficiency ordering still holds for two stochastic inputs, e.g., NBRW versus SRW and shuffling methods versus i.i.d input sequence. We have modified line 375 in Section 5 to illustrate our additional simulation results as follows.

---

> > ### Comment · Reviewer_jRx9 · 2022-08-03
> > **New data are convincing**
> >
> > I appreciate the effort of the authors in adding numerical data making their work more robust.
> > In light of the new results (which are clear and support their analytical findings) and the comments by the other referees, I am happy to improve my evaluation.

---

### Official Review · Reviewer_1nnX · 2022-08-14

**Rating:** 5
**Confidence:** 3
**Soundness:** 3 good
**Presentation:** 3 good
**Contribution:** 2 fair

**Summary:**

This paper proposes an asymptotic analysis of SGD when the noise sequence is not necessarily iid but can be a Markov chain such as a random walk over an arbitrary graph. The main contribution of this work is to connect the concept of efficiency ordering (which was previously introduced in the literature to compare Markov chains with the same stationary distribution and hence compare MCMC algorithms) with an ordering of covariance matrices inherited from Central Limit Theorems controlling rescaled SGD iterates driven by Markov chain noise sequences. As a consequence, SGD algorithms can achieve smaller covariance errors when using more efficient noise sequences for MCMC sampling. As applications of this result, the paper considers decentralized optimization over a graph using high-order random walks for SGD on the one hand and (mini-batch) gradient descent with shuffling on the other hand, confirming the superiority of these mechanisms as previously noticed in the literature. The paper also notably shows how some non-Markovian processes (for which non-asymptotic analysis are not available in the literature) can outperform Markovian input sequences for SGD in terms of efficiency ordering. Numerical experiments illustrate the theoretical findings.

**Questions:**

1. Regarding the main contribution (Theorem 3.6), in general, is showing that the AV is small easier than controlling the asymptotic covariance from the CLT? The applications (Props. 4.1 to 4.4) seem to exploit previously known efficiency ordering from the MCMC literature and transfer them to SGD using Theorem 3.6 whereas these last results (from the MCMC literature) may not be easy to establish in general.

2. The example of Fig. 1 is interesting and motivates looking at the AV as a metric for ordering instead of SLEM. However, I am wondering if one can also find an example where the converse holds, i.e. where SLEM may be more relevant if this makes sense. I understand that the point in the paper is to motivate looking at AV but commenting on the converse would also be interesting to complete the picture for the reader regarding the comparison between the asymptotic approach and its non-asymptotic counterpart.

3. In the contributions, it is stated that the analysis covers non-Markovian stochastic processes beyond Markovian sequences. The example in p. 7 is a second-order Markov chain (which is non-Markov) and it is analyzed via an augmented process which is a Markov chain. Could one also use non-asymptotic analysis based on the mixing time for this same augmented process? This type of approach does not seem to be out of reach for this specific kind of non-Markovian process compared to the asymptotic approach pursued in the paper.

Typos:
- l. 188: (A2), what is a? It should rather be \alpha defining the step size in (A1) I guess.
- l. 364-365: what are f and g? Is it \tilde{f} and \hat{f} instead?

**Limitations:**

Limitations are not explicitly discussed by the paper (as acknowledged by the authors in the checklist in Question1. b). Some comments and questions regarding limitations of the work are discussed in the section above.

**Strengths And Weaknesses:**

— STRENGTHS:

1. The paper states that the covariance matrices associated to two random walks following an efficiency ordering are Loewner ordered even for non-Markovian stochastic processes. This is an advantage of the asymptotic approach compared to its non-asymptotic counterpart which cannot handle the non-Markovian setting for which the mixing time is irrelevant beyond the Markovian noise setting (see a clarification question regarding this point in the next section below). Moreover, the result implies that better MCMC samplers could be used to enhance SGD thanks to the connection between AV for MCMC and covariance matrices for SGD algorithms from CLT. The paper hence establishes an interesting connection between the MCMC literature and the asymptotic covariance associated to (rescaled) SGD iterates (using the same noise sequence).

2. The comparison between shuffling and iid input sequence and the use of shuffling in mini-batch gradient descent are interesting questions even in practice and prior work has observed and established some advantages of these techniques. The present work compares them using an alternative asymptotic point of view compared to the more common non-asymptotic (finite-time bounds) in the literature relying on mixing times.

3. The paper also extends prior art regarding applications. For instance, it shows that a high-order random walk (NBRW) which is non-Markov is more efficient than SRW for SGD on general graphs beyond d-regular graphs for which this was previously established using mixing times. It also confirms some known advantages of shuffling over iid sampling for SGD and addresses a class of functions (verifying (A2)) that is not covered by prior work.

4. Clarity: the paper is well-written and the contributions are clearly stated. The results are sound and the paper is technically solid (I only quickly went through appendices A, C (checked the proof of the main result) and D though).

— WEAKNESSES:

1.  Unlike the asymptotic approach, a non-asymptotic analysis captures the dependence on constants of the problem such as the data and the graph topology in the distributed optimization setting.

2. Among other experimental facts, two main elements are presented as motivation to consider asymptotic variance as a tool for the analysis. First, the paper mentions the connexion of AV with the MSE. Nevertheless, this connexion between finite-time MSE and the covariance matrix does not seem to be straightforward beyond the case of quadratic objective functions (or linear stochastic approximation) which is well understood in the literature with existing thorough non-asymptotic analysis (for e.g. [18, 45] as mentioned in the paper). Second, the paper also mentions confidence intervals: although a smaller covariance matrix indeed reduces the confidence interval length, this covariance matrix depends on the Hessian at the minimizer which is unknown. Hence, it is hard to exploit the asymptotic analysis in practice, although this may not be the main purpose of the paper regarding this confidence interval remark.

3. The asymptotic analysis seems to provide little additional actionable information compared to the non-asymptotic approach. In l. 112-117, the paper mentions that the asymptotic variance provides more information than non-asymptotic MSE bounds, but is this information usable in practice? Can we leverage this additional information to improve the algorithms easily beyond the case where we already have a more efficient sequence (in the sense of the MC ordering) at hand? Also, for instance, in the paper, if the asymptotic analysis addresses a different class of functions (satisfying (A2)) which does not include PL functions (as previously addressed in the literature), it only confirms previously known facts regarding shuffling for SGD for which more informative finite-time bounds exist as stated by the paper (l. 323).

4. Using Theorem 3.6 may not be easy in general, one needs to know an efficiency ordering of two sequences to transfer it to SGD. Is the problem of determining the efficiency ordering between two sequences easier than comparing the covariance matrices inherited from the CLT corresponding to SGD with the two different noise sequences? (see question in the next section)

5. Technical significance: The main contribution of the paper as highlighted by the paper is Theorem 3.6. From the technical point of view, this result follows from the definition of the asymptotic variance of the Markov chain in one direction and from the well-known explicit integral form of the covariance matrix together with the stability of the Loewner ordering (Lemma D.2) for the converse (Appendix, p.6). Compared to the proof of this main result, the applications require more technical work. As acknowledged by the authors (especially in the appendix), CLTs are known in the literature, even for general stochastic approximation schemes driven by Markovian noise, of which SGD with Markov noise is a particular case. The paper relies on this prior work for the CLT.

---

### Author Response · Authors · 2022-08-02
**Response to All Reviewers**

Dear reviewers,

Thank you so much for your comments and for the time and effort you put into reading, understanding and evaluating our paper! In particular, we highly appreciate your suggestions and questions to make the writing clear (Reviewer 9xiZ, 3mgL), and we should simulate for large graphs and non-convex functions (Reviewer jRx9). We carefully considered all questions, concerns and comments provided by reviewers and addressed all of them appropriately. We provided detailed responses to each review separately and we believe that our responses address all of the reviewers' concerns. We have uploaded the revised paper and supplementary material to reflect our responses and included the original version in the zip file for your reference. All the indices from theorems and equations we mentioned in the response are based on the original paper and supplementary material. If additional details, explanations, or clarifications are needed, we will be happy to provide them.

---

### Meta-Review · Area_Chair_7ugA · 2022-08-23

**Recommendation:** Accept
**Confidence:** Certain

**Metareview:**

The paper proposes an asymptotic analysis of SGD when the noise governed by a Markov chain such as a random walk over an arbitrary graph. The paper connects efficiency ordering with an ordering of covariance matrices inherited from the Central Limit Theorems.
One of the results of the work is that SGD algorithms can achieve smaller covariance errors by using a more efficient noise sequences from MCMC sampling. The paper considers decentralized optimization over a graph using high-order random walks for SGD and mini-batch gradient descent with shuffling on the other hand, as applications. The reviewers raised questions on experimental results on larger graphs for their applications, and the authors have preformed and committed to including these new experiments. As a result, all the reviewers are in favour of accepting, and are now confident in their scores.

**Award:**

No

---

### Decision · Program_Chairs · 2022-09-14

Accept